# Cross species systems biology discovers glial *DDR2, STOM,* and *KANK2* as therapeutic targets in progressive supranuclear palsy

Yuhao Min [1,2,3], Xue Wang [4], Özkan İş [1], Tulsi A. Patel[1], Junli Gao[1], Joseph S. Reddy [4], Zachary S. Quicksall [4], Thuy Nguyen [1], Shu Lin[1], Frederick Q. Tutor-New[1], Jessica L. Chalk[1], Adriana O. Mitchell[1], Julia E. Crook[2], Peter T. Nelson[5,6], Linda J. Van Eldik [5,7], Todd E. Golde[8], Minerva M. Carrasquillo [1], Dennis W. Dickson [1], Ke Zhang[1], Mariet Allen[1] & Nilüfer Ertekin-Taner [1,9] ✉

Progressive supranuclear palsy (PSP) is a neurodegenerative parkinsonian disorder characterized by cell-type-specific tau lesions in neurons and glia. Prior work uncovered transcriptome changes in human PSP brains, although their cell-specificity is unknown. Further, systematic data integration and experimental validation platforms to prioritize brain transcriptional perturbations as therapeutic targets in PSP are currently lacking. In this study, we combine bulk tissue (*n* = 408) and single nucleus RNAseq (*n* = 34) data from PSP and control brains with transcriptome data from a mouse tauopathy and experimental validations in *Drosophila* tau models for systematic discovery of high-confidence expression changes in PSP with therapeutic potential. We discover, replicate, and annotate thousands of differentially expressed genes in PSP, many of which reside in glia-enriched co-expression modules and cells. We prioritize *DDR2, STOM,* and *KANK2* as promising therapeutic targets in PSP with striking cross-species validations. We share our findings and data via our interactive application tool *PSP RNAseq Atlas* (https://rtools.mayo.edu/PSP_RNAseq_Atlas/). Our findings reveal robust glial transcriptome changes in PSP, provide a cross-species systems biology approach, and a tool for therapeutic target discoveries in PSP with potential application in other neurodegenerative diseases.

Progressive supranuclear palsy (PSP) is a neurodegenerative disorder with a relatively early age of onset and rapid progression to death[1]. PSP is a primary tauopathy characterized by the overexpression of 4-repeat tau isoform in both neuronal and glial cells, leading to cell-specific tau

lesions including neurofibrillary tangles (NFT) in the neurons, tufted astrocytes (TA), coiled bodies (CB) in oligodendrocytes, and tau threads (TauTh) in white matter. Multiple genetic risk factors have been associated with PSP, including variants in or proximal to

[1]Department of Neuroscience, Mayo Clinic, Jacksonville, FL, USA. [2]Center for Clinical and Translational Science, Mayo Clinic, Rochester, MN, USA. [3]Mayo Clinic Graduate School of Biomedical Sciences, Mayo Clinic, Jacksonville, FL, USA. [4]Department of Quantitative Health Sciences, Mayo Clinic, Jacksonville, FL, USA. [5]Sanders-Brown Center on Aging, University of Kentucky, Lexington, KY, USA. [6]Department of Pathology & Laboratory Medicine, University of Kentucky, Lexington, KY, USA. [7]Department of Neuroscience, University of Kentucky, Lexington, KY, USA. [8]Department of Pharmacology and Chemical Biology, Department of Neurology, Emory Center for Neurodegenerative Disease, Emory University, Atlanta, GA, USA. [9]Department of Neurology, Mayo Clinic, Jacksonville, FL, USA. ✉e-mail: taner.nilufer@mayo.edu

microtubule associated protein tau (*MAPT*), myelin associated oligodendrocyte basic protein (*MOBP*) and RUNX family transcription factor 2 (*RUNX2*)[2,3]. Some of these loci are also associated with brain gene expression levels, suggesting that genetic variants may confer PSP risk via transcriptional regulation[4,5]. We previously showed that PSP brains have vast transcriptional perturbations[6,7]. PSP brain gene expression networks have distinct association patterns with the different cell-specific tau lesions, highlighting cell-specificity of these transcriptional changes in PSP[7,8].

While these findings nominate brain perturbations in gene expression and transcriptional networks as potential culprits in PSP pathophysiology, they are primarily based on microarray expression measures of bulk brain tissue from human cohorts or do not systematically investigate PSP brain RNAseq changes in independent cohorts. Additionally, complementary data from single cell/single nucleus RNAseq (sc/snRNAseq) is necessary to assess cell-type-specific brain gene expression changes in PSP. Further, although investigating these transcriptome changes in model systems across different species can increase confidence in the findings and enable experimental validations, such studies are lacking in neurodegenerative diseases with few exceptions[9,10]. Consequently, the exact disease mechanism(s) and key molecular players that underlie PSP pathogenesis remain unclear, creating a barrier to nominating effective targets for disease-modifying treatment.

To enable the translational discovery of potential therapeutic targets with mechanistic insights, the myriad brain transcriptional changes in PSP must be narrowed down by systematic prioritization platforms that integrate multiple data modalities and experimental validations. In this study we integrated bulk brain RNAseq data across two large, independent human cohorts, with complementary human brain snRNAseq and tau mouse model data to discover and prioritize high-confidence cell-specific gene expression and network perturbations in PSP followed by in vivo validations in a *Drosophila* tau model. We built an interactive web application *PSP RNAseq Atlas* to broadly serve the research community as a facile tool to access and utilize these rich, complex datasets (https://rtools.mayo.edu/PSP_RNAseq_Atlas/). Our study applies robust systems biology approaches to multi-omics data across species to nominate high-priority genes with therapeutic potential in tauopathies, specifically in PSP.

## Results

### PSP brains have vast and replicable transcriptome perturbations in glia-enriched genes

To identify perturbed genes in PSP, we first collected and analyzed the bulk gene expression profile from the superior temporal gyrus of temporal cortex (TCX) tissue for 281 neuropathologically confirmed PSP cases and 127 controls that lack significant pathology. Due to the relatively low degree of gross tau pathology in TCX[11,12], we hypothesized that this region would be less susceptible to confounding factors associated with downstream consequences of the disease, such as neuronal loss[6], making the expression changes less likely to be secondary to these confounds. TCX is also a more easily accessible region, where there is typically more sample availability and less susceptibility to variations in acquisition of gray matter. Expression data was collected as part of two independent studies (studies 1 and 2, Table S1). All PSP cases have detailed measures of cell-specific tau (CB, NFT, TA, TauTh) and overall degree of neuropathology quantified from multiple brain regions as previously described[4,8].

After quality control (QC), a total of 22,560 unique genes were detected in both studies, the majority (67%) of which are protein-coding (Table S2, Figure S1). To identify the gene(s) associated with PSP diagnosis, we compared the brain gene expression levels between PSP cases and controls using multiple linear regression adjusting for relevant covariates in each dataset separately and combined the results using an inverse-variance meta-analysis model. Using a similar

approach, we also analyzed the association of brain gene expression levels with the severity of tau neuropathologies measured within the 281 PSP cases. Compared to controls, 2,528 genes were differentially expressed (DEGs) in PSP brains at an FDR-adjusted p-value of 0.05 (Fig. 1a), suggesting extensive transcriptional dysregulations in PSP brains at the bulk tissue level, even in a brain region relatively spared from gross tau pathology. Importantly, a secondary model that adjusted for cell proportions (Figure S2) showed congruent expression dysregulation, indicating that the expression perturbations are not confounded by neuropathology-induced differences in cell populations. Neuropathology association among the PSP cases indicated the greatest number of associations with NFT (134 genes), while 8 gene levels were associated with TauTh (Fig. 1b). Using less stringent significance thresholds (unadjusted *p* value < 0.05), associations were also observed between gene expression levels with other tau neuropathologic lesions in PSP (Fig. 1a). A complete list of all expression associations is included in Supplementary Data 1.

We next focused on the top gene expression changes in PSP brains compared to controls. We defined top expression changes as a PSP vs control DEG with the bottom 5% FDR and top 5% |logFC| (Fig. 1b). We observed biologically congruent gene expression associations with both diagnosis and neuropathology such that genes that are higher in PSP than controls are also positively correlated with tau neuropathologies and those that are down in PSP have negative neuropathology correlations (Fig. 1b, S3). Among all DEGs (FDR < 0.05), there is also a striking and statistically significant cell-type specificity as many of the PSP up-regulated genes are also marker genes[13] for astrocytes (*p* < 0.001, 229 genes) or endothelia (*p* = 0.013, 63 genes). In contrast, many of the PSP down-regulated genes are oligodendrocyte-specific (*p* < 0.001, 77 genes) (Figs. 1b, c, S4).

Our review of the DEGs revealed genes previously implicated in PSP or Alzheimer's disease (AD) (Fig. 1d, e). One of the top perturbed genes is the antisense RNA *KANSL1-AS1* which is significantly lower in PSP brains (logFC = −1.00, FDR = 8.790E-5, Figs. 1d, e, S5, Table S3). Both *KANSL1-AS1* and its sense gene *KANSL1* are located within the PSP risk genome-wide association study (GWAS) locus near tau-encoding *MAPT*[2,4]. Another top perturbed gene, astrocyte-enriched *YAP1*, which is significantly higher in PSP brains, was identified as a regulatory network hub gene that is higher in AD brains[14]. We previously showed the astrocyte-enriched *CLU*, an AD risk GWAS gene, to be higher in both AD and PSP brains[6], whereas the oligodendrocyte-enriched *MOBP*, a PSP risk GWAS gene, is lower in both diseases[7]. Our current DEG results confirm prior work and demonstrate remarkable replication between the two brain cohorts (Fig. 1d). Collectively, these findings highlight robust transcriptome perturbations in PSP brains in glia-enriched genes.

### Glial cell-enriched gene co-expression network modules are associated with PSP

To identify and characterize groups of co-expressed genes associated with PSP we performed harmonized weighted gene co-expression network analysis (WGCNA)[15] across both studies. A total of 16 modules were detected using the consensus network construction algorithm after harmonized gene assignments (Fig. 2, Supplementary Data 2). These consensus network modules were robust with high agreement between the gene expression clusters and module assignments (Figure S6a) and high preservation of modules between the two studies (Figure S6b). We correlated the module eigengenes (ME), a summary measure of all gene expression values in the module, with PSP diagnosis and each neuropathology phenotype (Fig. 2a). We identified 6 modules that are significantly (Bonferroni adjusted *p* < 0.05) associated with PSP diagnosis. Among them, 3 modules have higher (M4, M6, and M12) and 3 modules have lower (M3, M10, M11) expression in PSP cases. These associations were consistent across both studies (Figure S7).

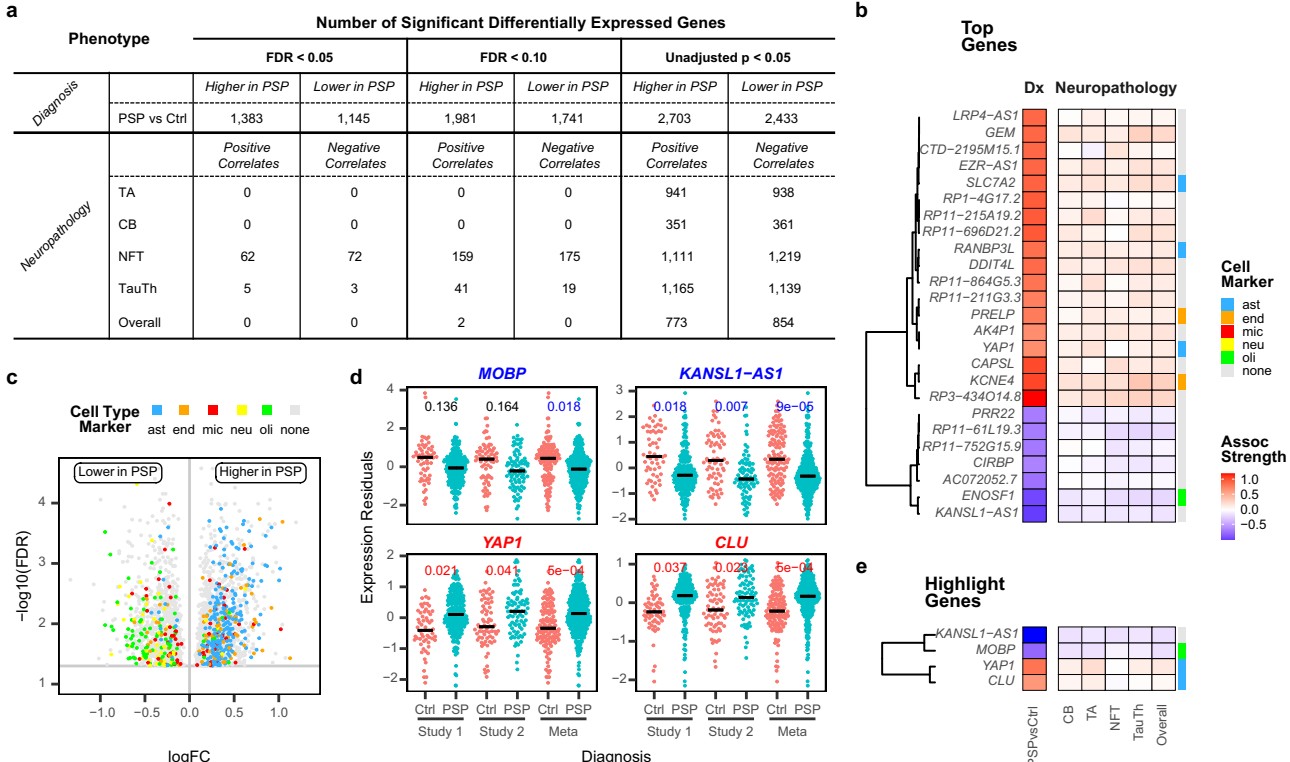

**Fig. 1 | Bulk RNAseq analysis identifies vast and replicable differentially expressed genes in PSP brains. a** Number of differentially expressed genes with respect to different phenotypes at different threshold. **b** Heatmap of logFC (for PSPvsCtrl) and association beta (for neuropathology) of the top genes defined as lowest 5% FDR and top 5% |logFC|. Top genes that are also marker genes for astrocyte, endothelia, and oligodendrocyte are color-coded. **c** Volcano plot for all DEGs with FDR < 0.05 comparing gene expression levels between PSP and control. The color of the dots represents cell type enrichment of the DEGs: blue = astrocyte, orange = endothelia, red = microglia, yellow = neuron, green = oligodendrocyte, gray = no cell type specificity. **d** Distribution of the expression levels of *MOBP*,

*KANSL1-AS1*, *YAP1*, and *CLU* between PSP and control in each study cohort. We highlighted these genes given their known association with PSP risk or other neurodegenerative diseases. Statistics: differential expression analysis using linear regression adjusting for covariates. FDR-adjusted p values are presented. Unadjusted p values are provided in Supplementary Data 1 and source data. Study 1: *N* = 257 PSP and Control individuals; Study 2: *N* = 151 PSP and Control individuals; Meta: *N* = 408 PSP and Control individuals. Also see Supplementary Table 1. **e** Heatmap for the 4 genes highlighted in **d** also showed consistent expression changes with PSP diagnosis and neuropathology. Source data are provided as a Source Data file.

We hypothesized that these modules represent gene expression perturbations in distinct cell types and biological pathways (Fig. 2b–d). Indeed, three of the PSP-associated modules were enriched for glial genes. We found an enrichment of oligodendrocyte genes (Bonferroni-adjusted $p < 2.22E-16$) in the down-regulated M3, consistent with our prior work[8]. M6, which is up in PSP, is enriched for endothelial (Bonferroni-adjusted $p < 2.22E-16$) and microglial (Bonferroni-adjusted $p = 5.29E-4$) marker genes, whereas M4, also up in PSP, is an astrocyte-enriched module (Bonferroni-adjusted $p < 2.22E-16$). Gene ontology (GO) terms enriched in co-expression modules were broadly consistent with their cell types (Fig. 2d, S8, Supplementary Data 3). Microglial/endothelial M6 is enriched for immunity- and vascular-related GO terms, while astrocytic M4 has metabolic, and oligodendroglial M3 has RNA processing/splicing GO term enrichment. These findings suggest that cell-type specific transcriptional changes in PSP may be indicators of disrupted biological processes including immune activation, angiogenesis, cell energetics and RNA metabolism.

**Single-nucleus RNAseq captures glial expression changes in PSP brains**
We sought to further replicate the glial gene expression changes from bulk brain RNAseq detected in our two independent studies by conducting snRNAseq experiments in brain tissue samples from the TCX of PSP and control patients (Table S4). After QC (Figure S9), we obtained 26,241 nuclei from 34 brain samples comprising 18 PSP and 16 controls. To ensure cell populations can be reliably identified across each sample, we integrated the snRNAseq data, treating each sample

as a batch variable. Nuclei were clustered based on the integrated PCA embedding using the Louvain algorithm implemented in Seurat[16], which yielded 28 nuclei clusters. (Fig. 3a). There was no statistically significant enrichment of nuclei from either sex or diagnosis in any cluster (Figures S10, 11). Although a few samples were statistically over-represented in 8 clusters, 7 of these clusters were small, and each contributed to <1% of the total nuclei in the snRNAseq dataset, indicating homogeneity for most of the nuclear clusters with respect to sex, diagnosis, and samples.

We annotated each cluster based on the overlap of their highly overexpressed cluster-marker genes and a list of (see Methods) well-known cell-type marker genes curated from the literature (Fig. 3b) or via published databases (Figure S12, 13)[13,17], which yielded consistent annotations. All major brain cell types were identified (Fig. 3c). All clusters could be annotated according to their cell type except three small clusters (CL23, CL24, CL26), which constitute <1% of total nuclei.

Upon comparison of the expression levels between PSP and control nuclei in each cluster, we identified significant DEGs at FDR < 0.05 for genes expressed in >10% of the nuclei in the cluster analyzed. We detected significantly up- or down-regulated DEGs in both neuronal and glial clusters (Table S5, Supplementary Data 4). To provide additional context and annotations for the genes from the three glial-marker enriched WGCNA co-expression modules, M3, M4 and M6, that are replicably associated with PSP in bulk RNAseq from two studies, expression scores, calculated based on the average expression levels of these modules' genes for each nucleus in the

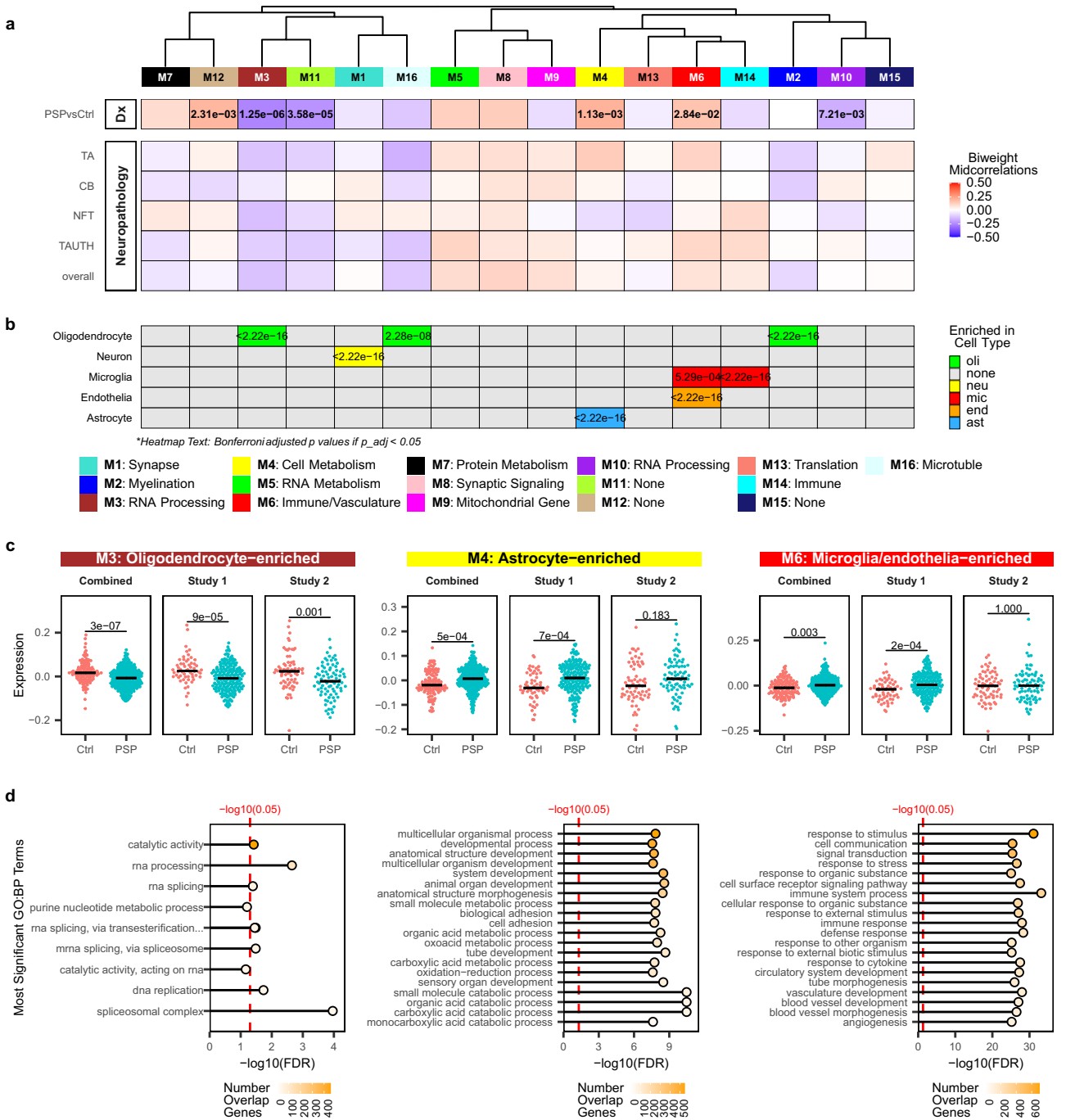

**Fig. 2 | Glial cell-enriched gene co-expression network modules are associated with PSP. a** The associations between the module eigengenes (ME) and PSP diagnosis or quantitative tau neuropathology were assessed. **b** Enrichment of cell type specific genes was detected in seven modules, three of which (M3=oligodendrocyte, M4=astrocyte, and M6=microglia/endothelia) were significantly associated with PSP. Statistics: one-sided Fisher's Exact Test. Unadjusted *p* values are provided in Supplementary Data 2 and source data. **c** Associations were robust and consistent across both studies and significant in the consensus network modules M3, M4, M6. Statistics: two-sided t-test comparing PSP and control module eigengenes. Bonferroni-adjusted p values are presented. Study 1: *N* = 257 PSP and Control individuals; Study 2: *N* = 151 PSP and Control individuals; Combined: *N* = 408 PSP and Control individuals. Also see Supplementary Table 1. **d** Top enriched gene ontology biological process terms in the PSP-associated glial cell-enriched modules. Source data are provided as a Source Data file.

snRNAseq, were analyzed across clusters for each cell type (Fig. 3d). For completeness, all clusters were included in the analysis, even for those that accounted for less than 1% of the total nuclei. Module genes have high expressions in the nuclei corresponding to the enriched cell type of each module. Oligodendrocyte-enriched module M3, astrocyte-enriched M4 and microglia/endothelia-enriched M6 genes have the highest expression in oligodendrocyte, astrocyte, and endothelia snRNAseq clusters, respectively, thus validating module cell-type annotations. There was expression of these module genes in other cellular clusters, underscoring that they are enriched in but not exclusive to specific cell types.

We subsequently evaluated the overlap between the snRNAseq DEGs in each nuclear cluster with genes from the three modules (Fig. 3e, S14). Oligodendrocyte-enriched M3 genes have the most significant overlap (*p* = 2.74E-10) with DEGs of the oligodendrocyte cluster CL0. Further, 86.76% of these overlapping snRNAseq genes were

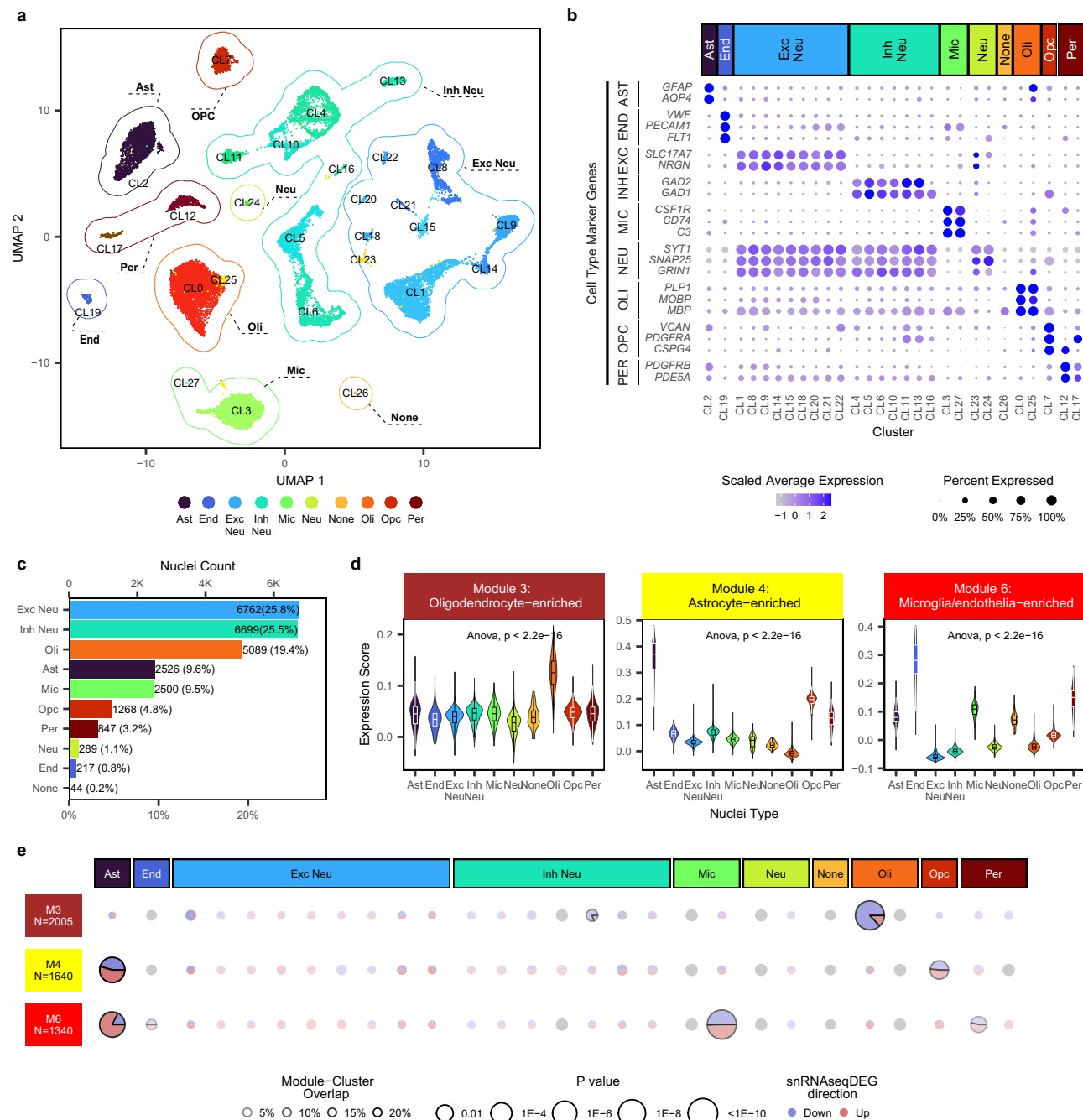

**Fig. 3 | Single-nucleus RNAseq captures glial expression changes in PSP brains.** **a** Nuclei in UMAP space colored by cluster number and grouped by its cell type. **b** Average expression and percent nuclei expressed of the cell type marker genes used for cluster type assignment in each snRNAseq cluster. **c** Count and proportions of each nuclei type. **d** Expression levels of module genes from the three cell-type-enriched, PSP-associated expression modules (M3, M4, M6) in the snRNAseq clusters for each cell type. Boxplots indicating the distribution of the expression scores and the first quartile, the median, and the third quartile. The upper whisker indicates the maximum value no further than 1.5 times the inter-quartile range from the third quartile. The lower whisker indicates the minimum value no further than 1.5 times the inter-quartile range from the first quartile. The expression was derived using 26,241 nuclei from 34 PSP and control samples. Statistics: One-way Analysis of Variance. Exact *p* values are provided in source data. **e** The number of overlapping genes between snRNAseq DEGs in each cluster and the three modules. *P* value: Fisher's exact test for the enrichment of snRNAseq DEGs in the expression module. Pie chart demonstrates proportion of up- and down-regulated genes. The radii of the pie charts reflect the significance of the overlap, where significant overlaps (*p* < 0.05) have solid outline. Transparency of pie chart reflects the size of the overlap between module genes and cluster DEGs (more opaque=higher number of overlapping genes). Statistics: one-sided Fisher's Exact Tests. Source data are provided as a Source Data file.

down in PSP, consistent with the negative association between M3 eigengene and PSP (Fig. 2a). Astrocyte-enriched M4 genes overlap mostly with astrocyte CL2 DEGs (*p* = 4.48E-49). These overlapping genes have a slightly higher proportion of up DEGs (54.52%), consistent with the positive association between M4 eigengene and PSP (Fig. 2a).

Microglia/endothelia-enriched M6 genes significantly overlap with microglia CL3 DEGs (*p* = 2.34E-10), pericytes CL12 DEGs (p = 3.90E-4) and astrocytes CL2 DEGs (*p* = 3.35E-20). Their direction of change is mostly up in astrocytes (82.36% up) and mixed in the other clusters (microglia: 49.57% up, pericytes: 53.85% up).

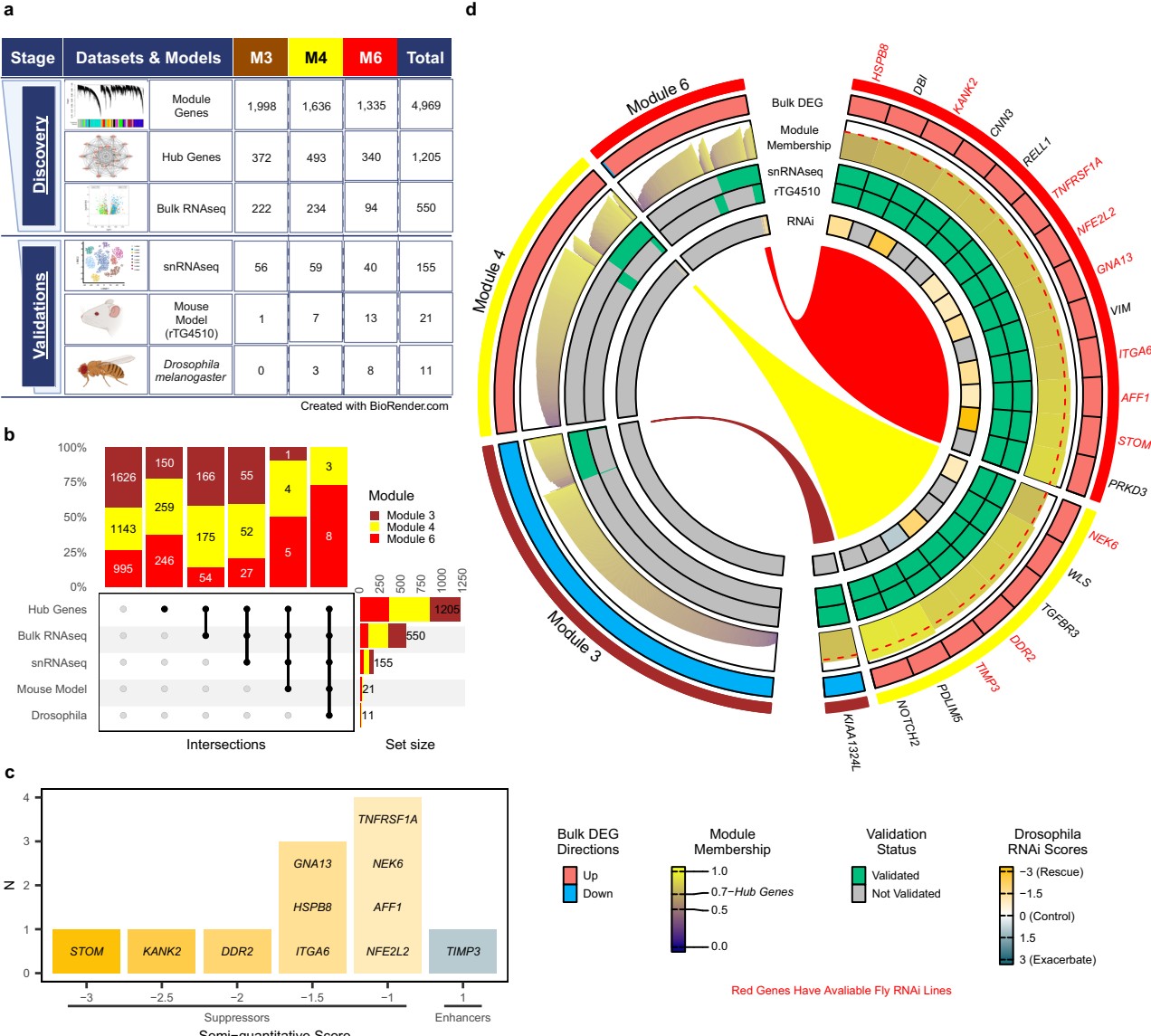

**Fig. 4 | Prioritization and experimental validation strategy for PSP brain transcriptome changes. a** Prioritization approach using bulk brain human RNA-seq, snRNAseq and rTg4510 tau mouse brain transcriptome data which led to 21 high confidence glial perturbed genes in PSP. Of these, 11 had available *Drosophila* ortholog genes with RNAi for in vivo screening. **b** Upset Plot showing the overlap of different validation methods and the number of genes selected at each filtering stage from each module. **c** Summary results of the *Drosophila* screen of the 11 genes in using the GMR-Gal4 system. **d** Circos plot visualization of genes from the three glial cell-enriched, PSP-associated gene expression modules. Modules M3, M4, M6 (outer-most = first ring) and their bulk DEG directions of PSP associations (second ring, red = up and blue = down-regulated modules), module membership (third ring, module membership scores are color-coded), validation in snRNAseq (fourth ring, validated = green), tau mouse model rTg4510 brain RNAseq (fifth ring, validated=green), and *Drosophila* tau model RNAi in vivo screening (sixth ring, *Drosophila* eye neurodegeneration morphology scores are color-coded) are shown. Red text indicates the 11 high confidence PSP glial perturbed hub genes that have available *Drosophila* orthologs and were screened experimentally in vivo. Source data are provided as a Source Data file.

In summary, snRNAseq data corroborates the cell-enrichment annotations for the PSP-associated bulk RNAseq modules M3, M4, and M6. Directionality of gene expression changes for these modules' genes is highly consistent between bulk RNAseq and snRNAseq oligodendrocyte cluster, although other glial clusters have DEGs that change in both directions, suggesting that single nucleus data may capture more subtle gene expression changes that may be missed in bulk data.

**Cross-species prioritization and screening of glial gene expression changes in PSP**

Our bulk RNAseq data analyses yielded expression perturbations, highlighting glial cell-enriched co-expression modules (M3, M4, M6) comprising 4,969 genes. The subset of these module genes that are also significant snRNAseq DEGs in the same glial cell clusters still constitute a large number for experimental validations (Figure S14). We, therefore, applied a systematic data-driven prioritization approach to further narrow down and select genes using the human brain transcriptome data from this study and a published tau mouse model[18].

Our prioritization approach is schematized in Fig. 4a. Among the 4969 genes from modules M3, M4 and M6, we focused on those that are central and highly connected within each module, defined as hub genes with module membership (MM) > 0.7. There are 550 hub genes that are also DEGs (FDR < 0.05) based on the meta-analysis of two bulk brain RNAseq studies. We further filtered these genes by selecting those that are also significant DEGs in the snRNAseq data cluster corresponding to the bulk module cell type and that have a concordant

direction of change between bulk and snRNAseq. For oligodendrocyte-enriched M3 and astrocyte-enriched M4 hub genes, we filtered for snRNAseq DEGs that are down in PSP in oligodendrocyte and up in astrocyte clusters, respectively, resulting in 56 and 59 genes, respectively (Fig. 4a). For the microglia/endothelia-enriched M6 hub genes, given the enrichment of snRNAseq DEGs from this module genes within these cell types (Fig. 3e), we selected those that are up-regulated snRNAseq in microglia, endothelia, pericytes or astrocytes, resulting in 40 genes. In total, there are 155 such genes which reflect robust glial gene expression changes in PSP brains.

To determine the gene expression changes that are preserved in a mouse model of tauopathy, we utilized brain transcriptome data from a mouse model that overexpresses a mutant form of human tau encoding *MAPT* and develops tau neuropathology by 4 months of age[18]. We analyzed brain gene expression data from 24 transgenic (rTg4510) mice and their wild-type control littermates sacrificed at 4.5 or 6 months of age (Table S6), available on the AD Knowledge Portal[19].

Among the 155 genes with robust glial expression changes in PSP brains, 21 were also significant DEGs perturbed in the same direction as humans in either 4.5- or 6-month rTg4510 mouse brains (Fig. 4a, b). The rTg4510-validated genes are significantly over-represented in the astrocyte-enriched module M4 ($p = 1.97E-3$) and microglia/endothelia-enriched module M6 ($p = 4.63E-18$), whereas only one DEG from the oligodendrocyte-enriched M3 was validated in the mouse data. The cell-type-specific overrepresentation of validated genes suggests that the rTg4510 mouse model may best recapitulate tauopathy-related glial expression changes in astrocytes, endothelia, and microglia but not oligodendrocytes. The 21 genes that passed through the human and mouse model filters represent high-confidence PSP glial DEGs with cross-species validation.

To determine whether experimentally perturbing levels of high-confidence PSP glial genes would impact tau-related neurodegeneration, we carried out a knockdown screening experiment with the 21 genes using a *Drosophila* model of tauopathy with GMR-Gal4 driver[20]. Of the 21 high-confidence PSP glial DEGs, 11 have a *Drosophila* ortholog and available RNAi stocks (Tables S7, 8). Using a semi-quantitative scoring system, where negative scores reflect suppression and positive scores reflect enhancement of the neurodegenerative eye morphology, we determined that RNAi suppression of 10 of the 11 tested genes suppressed the neurodegenerative eye morphology (Fig. 4c). The prioritization and gene information is summarized in Fig. 4d.

### Validation of top tau toxicity suppressing PSP glial perturbed genes

Three of the 11 screened genes, *DDR2, KANK2, and STOM*, have the strongest suppressive effect on the *Drosophila* tau eye phenotype (Score < −1.5) when down-regulated with RNAi (Fig. 4c, d). *DDR2* is a hub gene in the astrocyte-enriched M4, whereas *KANK2* and *STOM* are hubs in the microglia/endothelia-enriched M6 (Fig. 5, Figures S15, 16). We first examine the gene expression perturbation of these three genes in our human brain transcriptome data. In alignment with the prioritization paradigm, all three genes have higher expression levels (FDR < 0.05) in PSP brains based on bulk RNAseq (Fig. 5a). The up-regulation in bulk tissue is also consistent across both studies, suggesting the results are not driven by a particular cohort. The three genes are up-regulated even after adjusting for differences in cell proportions (Table S7), indicating that the expression perturbations are not driven by this but are likely associated with disease pathogenesis. In snRNAseq (Fig. 5b), all three genes are significantly upregulated in astrocytes. Pseudobulk DEG analysis using the snRNAseq datasets also supported the up-regulation of the three genes in astrocytes (Table S8). Interestingly, the subclustering analysis (Figure S17) indicated *DDR2* and *KANK2* are up-regulated in specific populations of astrocytes including reactive astrocytes, whereas the up-regulation of *STOM* is universal to all astrocytes. Our transcriptome data demonstrate that *STOM, KANK2*, and *DDR2* are up-regulated in glial cells in PSP.

Consistent with the results that *DDR2, KANK2*, and *STOM* are all upregulated in human PSP and rTg4510 mouse brain transcriptome, our initial tau *Drosophila* screening experiment (Fig. 4c) demonstrated that RNAi inhibition of these genes suppressed tau-mediated cell toxicity in this model. To further validate and confirm these *Drosophila* screening results, we repeated the RNAi experiments for these three genes, and quantified the severity of tau-mediated toxicity using two blinded independent evaluators (Fig. 5c, d). As expected, the expression of tau in *Drosophila* under the GMR-Gal4 driver led to significant eye degeneration compared to wild type (WT) flies (two-sided Wilcoxon rank sum test, p = 6.16E-5). Importantly, expression of RNAi against *DDR2, KANK2*, or *STOM* in tau *Drosophila* significantly reduced the tau-mediated cell toxicity in fly eyes. These results suggest that these three genes can be targeted in PSP or other tauopathies as a potential therapeutic avenue to ameliorate tau-mediated neurodegeneration.

We built an interactive web application tool *PSP RNAseq Atlas* (https://rtools.mayo.edu/PSP_RNAseq_Atlas/), which houses human bulk and snRNASeq transcriptome, mouse and *Drosophila* results. Our application can quickly report the key statistics for queried genes and is built to facilitate the dissemination of our results to the broader research community.

## Discussion

To translate the growing amount of multi-omics data into high-confidence candidate therapeutic targets with mechanistic implications, systematic prioritization platforms that integrate large-scale, multi-modal data and experimental validations are required. In this study, we applied a cross-species systems biology approach to large-scale human brain transcriptome data at bulk tissue and single nucleus resolutions, a tau mouse model[18] and experimental validations in a *Drosophila* tau model[20], which revealed robust glial transcriptome perturbations and nominated therapeutic targets in PSP.

Using bulk brain transcriptome, we discovered 2528 differentially expressed genes (DEGs) that are robustly detected across two studies of 408 donors, comprised of PSP cases, and controls without neurodegenerative diseases. PSP up-regulated DEGs are enriched for astrocyte- or endothelia-specific genes, whereas down-regulated genes are enriched for oligodendrocyte markers. This human bulk brain transcriptome is organized into 16 co-expression network modules, three of which have both robust PSP association and cell-type enrichment. Consistent with the DEG findings, the PSP down-regulated module M3 is enriched for oligodendrocyte genes, whereas the up-regulated M4 and M6 are astrocyte and microglia/endothelia enriched modules, respectively. In our prioritization paradigm, we first focused on 550 bulk DEGs that are also highly connected hub genes in these glial modules, based on the rationale that these centrally linked genes with perturbed expressions may represent targetable disease pathway molecules.

With brain snRNAseq of PSP and control donors, we further filtered down these 550 hub bulk DEGs by selecting the subset of 155 that are snRNAseq DEGs with concordant expression changes in the cell cluster corresponding to the bulk module cell type (Figure S18). Of these 155 human bulk brain and snRNAseq glial DEGs, 21 had consistent changes in the rTg4510[18] tau mouse model brain RNAseq. Eleven of these had *Drosophila* orthologues and when tested in an RNAi-based screening of *Drosophila* tau model[20], they significantly influenced tau-related neurodegeneration. We prioritize *STOM*, and *KANK2* from the microglia/endothelia enriched (M6) and *DDR2* from the astrocyte-enriched (M4) modules, as potential therapeutic targets with both multi-omics and model system validation in our study.

*STOM* encodes stomatin, which belongs to an evolutionarily conserved superfamily of proteins, is oligomeric, localizes to

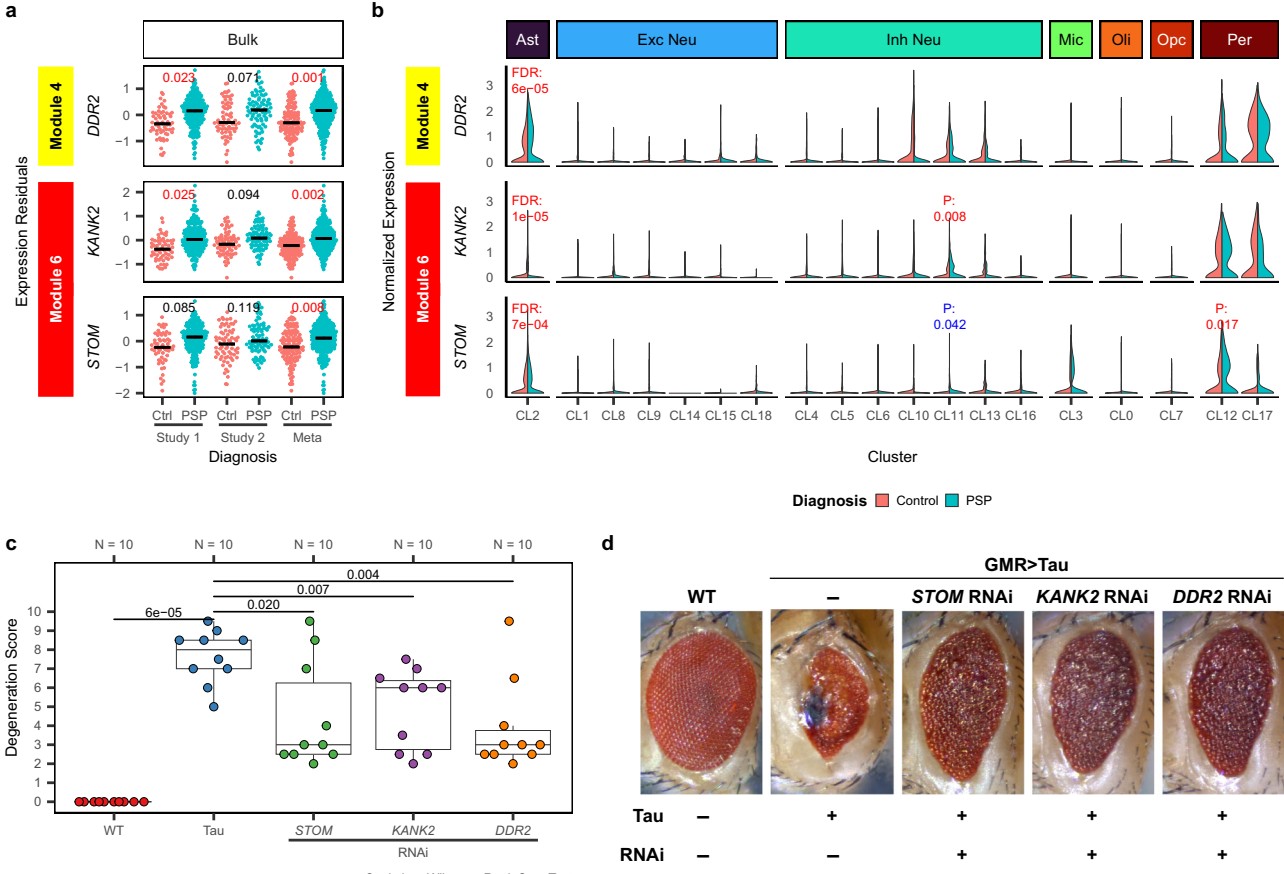

**Fig. 5 | Functional validation of the top tau-mediated toxicity suppressing PSP genes. a** Expression levels of the top in vivo validated genes in bulk RNAseq dataset. Horizontal bar indicates median expression levels. Statistics: differential expression analysis using linear regression adjusting for covariates. FDR-adjusted *p* values are presented. Unadjusted *p* values are provided in Supplementary Data 1 and source data. Study 1: *N* = 257 PSP and Control individuals; Study 2: *N* = 151 PSP and Control individuals; Meta: N = 408 PSP and Control individuals. Also see Supplementary Table 1. Red text indicates up-regulation in PSP. Blue text indicates down-regulation in PSP. **b** Expression levels of the top in vivo validated genes in the snRNAseq dataset. Statistics: Zero-inflated generalized linear regression comparing PSP versus Control nuclei adjusting for sex and age, implemented by the MAST package. FDR-adjusted (FDR) or unadjusted (*P*) *p* values as indicated. Red FDR/p

indicates up-regulation in PSP. Blue FDR/p indicates down-regulation in PSP. Expression changes with unadjusted *p* values greater than 0.05 were not annotated for clarity. **c** Statistical tests (two-sided Wilcoxon rank sum tests) showed a significant reduction in eye degeneration (*n* = 10 biologically independent flies per genotype were assessed). Boxplots indicating the distribution of the fly eye degeneration scores, the median, and the third quartile. The upper whisker indicates the maximum value no further than 1.5 times the inter-quartile range from the third quartile. The lower whisker indicates the minimum value no further than 1.5 times the inter-quartile range from the first quartile. **d** Representative *Drosophila* eye pictures indicate a robust rescue of tau-mediated toxicity when inhibiting the expression of the fly ortholog of *STOM, KANK2*, or *DDR2* with RNAi. Source data are provided as a Source Data file.

membrane lipid rafts, functions as a membrane scaffold through actin binding and regulates activities of ion channels and receptors[21]. Stomatin has anti-cancer properties[22], is upregulated with hypoxia in the rat brain[23] and has known chemical ligands, highlighting its potential as a druggable target[24]. *KANK2* is a primarily mesenchymal member of an evolutionarily conserved gene family[25,26], interacts with actin and serves in multiple roles acting as a scaffold within the cortical microtubule stabilizing complexes[25] including cell migration[27].

The emergence of two microtubule-associated membrane scaffold proteins, *STOM* and *KANK2*, as top prioritized potential therapeutic targets for PSP, a primary tauopathy characterized by microtubule-associated protein tau neuropathology, is noteworthy. Both genes are enriched in biological processes involved in the immune response (Figs. 2, 3) within module M6, which has greatest overlap of bulk DEGs with astrocyte, microglial, and endothelial snRNAseq DEGs. Collectively, these results suggest that these proteins may be involved in the pathogenesis of PSP by promoting glial tau neuropathology via disrupting microtubule structure and function.

*DDR2*, which encodes discoidin domain receptor, is a collagen-activated receptor tyrosine kinase that regulates multiple functions

including cell migration, proliferation and tissue modelling[28] through MAP kinase pathway, leading to the activation of transcriptional factor *RUNX2*[29], which itself is a PSP risk gene[30]. Besides its functions in bone modelling[29] and cancer[28], *DDR2* has also been implicated in neurodegeneration[31]. Consistent with our findings, it was shown that DDR2 levels are upregulated in the astrocytes from *MAPT* mutant organoids[32]. *DDR2* knockdown in vitro and in vivo led to increased clearance of neurodegenerative proteins including tau, reduced cell loss and attenuated immune response including *TREM2* positive microgliosis[31]. *DDR2* was previously shown to oppose the osteoclastogenic effect of *TREM2* by sequestering its signalling partner PlexinA1[33]. Intriguingly, *DDR2* shares signalling partners with the neurodegenerative disease risk gene, *TREM2*[34], and both genes have functions in bone metabolism and neurodegeneration. Our findings add *DDR2* to the growing list of neurodegenerative disease genes also involved in bone modelling.

Potent small molecule inhibitor of DDRs, nilotinib replicated the experimental outcomes of DDR knockout models[31]. Nilotinib was shown to be safe in randomized clinical trials for Parkinson's disease[35–37] and Alzheimer's disease[38], with a reduction of

cerebrospinal fluid (CSF) tau reported in some studies[35,38]. Our findings suggest that nilotinib or other small molecules inhibiting *DDR2* may also be viable therapeutic candidates in PSP. We provide a list of small molecule inhibitors retrieved from the Drug Gene Interaction database (DGIdb) in Supplementary Data 5.

Our collective findings demonstrate glial expression changes in PSP in a brain region relatively spared from gross neuropathological changes in this condition[11,12], suggesting these changes are less likely to be driven by neuropathology including neuronal loss or gliosis. We detected a smaller number of genes significantly associated with neuropathology than with PSP diagnosis. This is likely because the analysis was carried out only in PSP cases, which has lower power both owing to smaller sample size and smaller effect size. Furthermore, the neuropathology phenotypes represent specific cellular tau pathology in PSP, which may be associated with less transcriptional perturbations than the binary overall disease phenotype. Nevertheless, our cross-species multimodal systems biology approach prioritized three genes as potential therapeutic targets in PSP. Additionally, the internal replication of the PSP bulk transcriptome changes in two independent studies with confirmation in snRNAseq provides the field with hundreds of high-confidence genes well-annotated for their module membership, cell-type enrichment, and biological processes.

Our findings also replicate prior transcriptome changes reported in PSP including down-regulation of oligodendrocyte-enriched myelination gene networks[7] and up-regulation of immune networks with astrocytic tau pathology[8]. Some of the most perturbed genes in PSP brains have been implicated in the risk of PSP and shown to have regulatory changes, namely oligodendrocytic *MOBP*[2,4,7], *KANSL-AS1*[2,4,10] and astrocyte-enriched *YAP1*[14]. Another astrocyte-enriched AD risk gene, *CLU*, is also significantly higher in PSP brains and AD[6], underscoring common transcriptome changes in these two neurodegenerative diseases[6] that may point to shared disease mechanisms. Our findings and web tool are therefore expected to be useful across different neurodegenerative disorders.

Despite these strengths and insights, there are limitations to our study. We acknowledge that single-nucleus RNAseq captures only the transcriptional activity within the nucleus and might potentially miss biological changes relevant to PSP, which may be detectable by single cell RNAseq (scRNAseq). Nevertheless, we were still able to validate in snRNAseq the bulk RNAseq expression changes in the corresponding nuclei type that led to the discovery of *DDR2*, *KANK2*, and *STOM* as potential therapeutic targets for PSP. Moreover, snRNAseq allowed the use of frozen brain tissue, which made it possible to include as many PSP samples as possible. There are no PSP-specific mouse models. We used a single mouse model of tauopathy rTg4510[18] that despite age-dependent brain deposition of human 4-repeat tau, neuronal, and synaptic loss, is not strictly a model of PSP. Further, factors other than human tau overexpression may be contributing to these mouse phenotypes[39]. However, this model was still valuable for the purpose of screening human DEGs with concordant changes in a mouse model of tauopathy. We note that although many of the astrocytic, microglial, and endothelia genes from modules M4 or M6 were validated in rTg4510, this was not the case for oligodendrocytic DEGs. This suggests that the rTg4510 tauopathy model may be capturing some but not all aspects of the glial perturbations in PSP. Our available data will enable future studies for conducting systematic cross-species comparisons to identify models that recapitulate different aspects of human brain molecular perturbations. This will be useful in selecting appropriate models for experimental validations and pre-clinical therapeutic trials. Moreover, carrying out cell-type-specific overexpression or knockdown of the target genes in mouse models and assessing the clinical impact will be informative to validate the therapeutic target prior to clinical applications.

In our study, we elected to use a well-established *Drosophila* model of tau-mediated neurodegeneration[20,40] for experimental validations. Although this model has the advantage of facilitating a relatively rapid and cost-effective medium-throughput experimental screen, it is not a one-to-one translation to human brain disease. Additionally, unlike the *repo*-Gal4 driver[41], the *Drosophila* GMR-Gal4 driver is not glia specific as it drives the expression of tau in all cell types. Nevertheless one of the top prioritized genes *DDR2*, has congruent findings in *Drosophila* when suppressed with RNAi as in other in vitro and in vivo models[31] and human clinical trials[35,38]. This provides proof-of-principle for the utility of the fly as a facile screening tool to prioritize findings for further validations by more cumbersome though also more complex models.

In summary, our systems biology approach that integrates multimodal omics and phenotype data across species with experimental validations discovered *DDR2, STOM* and *KANK2* as potential therapeutic targets in PSP. Our findings demonstrate robust transcriptome changes in glia that may underlie the striking glial tau-pathology in this disease. Our findings and data we make available in the interactive web application *PSP RNAseq Atlas* (https://rtools.mayo.edu/PSP_RNAseq_Atlas/) highlight the complex pathophysiology of PSP. Our web application, data and results are also expected to serve as a resource for the research community to apply their own paradigms for therapeutic target or biomarker prioritizations in PSP or other neurodegenerative diseases.

## Methods
### Sample information
This study was approved by the Mayo Clinic Institutional Review Board (IRB). The use of samples from the University of Kentucky and Banner Sun Health Research Institute has been approved by the appropriate review boards. Additional data used in this study from the AD Knowledge Portal (https://adknowledgeportal.synapse.org) were accessed under the data usage agreement (https://adknowledgeportal.synapse.org/DataAccess/DataUseCertificates). All personally identifiable information has been removed. Written informed consent was obtained from all participants, their qualified caregivers or next of kin.

We have collected RNAseq from bulk brain tissue of 408 frozen, post-mortem, temporal cortex (superior temporal gyrus) tissue samples consisting of 127 control and 281 PSP patients from two independent study cohorts (Table S1). All PSP cases were obtained from the Mayo Clinic Brain Bank and received a neuropathologic diagnosis of PSP[11,12] by a single neuropathologist (DWD). All PSP samples also underwent neuropathological evaluation for the overall and cell-specific tau lesions (TA, CB, NFT, TauTh). The semi-quantitative tau pathology scores were transformed and normalized as previously described[4,8]. Amongst brain donors with PSP, there was only one patient with a *MAPT* mutation, namely the A152T variant which is considered a rare risk factor for tauopathies and not a fully penetrant pathogenic mutation[42,43].

Control samples were obtained from Banner Health Research Institute (Banner), Mayo Clinic Brain Bank (Mayo), and University of Kentucky (UKy). Controls were defined as those brain samples with Braak[44] NFT stage of 3.0 or less; CERAD[45] neuritic and cortical plaque densities of 0 (none) or 1 (sparse) and lacked any of the following pathologic diagnosis: AD, Parkinson's disease (PD), DLB, VaD, PSP, motor neuron disease (MND), CBD, Pick's disease (PiD), Huntington's disease (HD), FTLD, hippocampal sclerosis (HipScl) or dementia lacking distinctive histology (DLDH), as previously described[46].

Study 1 comprises 199 PSP and 58 control samples (43 UKy, 15 Mayo), whereas study 2 has 82 PSP and 69 control samples (44 Banner, 25 Mayo). RNAseq and DEG from study 2 were previously reported[6,46]. Within each study cohort, there was well-balanced sex distribution between PSP and controls. However, there were significant differences in donor age at death, RNA integrity number (RIN) and brain bank source, all of which were adjusted for in the analyses.

For the snRNAseq, all samples were obtained from the Mayo Brain Bank. The PSP cases ($N = 18$) received a neuropathologic diagnosis of PSP[11,12] by a single neuropathologist (DWD). The control samples ($N = 16$) are defined as those brain samples that lack any of the following pathologic diagnosis: AD, Parkinson's disease (PD), DLB, VaD, PSP, motor neuron disease (MND), CBD, Pick's disease (PiD), Huntington's disease (HD), FTLD, hippocampal sclerosis (HipScl) or dementia lacking distinctive histology (DLDH), as previously described[4,8]. Samples for snRNAseq were selected to ensure that the PSP and control donors had comparable sex and age at death. Additionally, we prioritized samples with available bulk tissue RNAseq and/or whole genome sequencing data and for those that had sufficient amount of available frozen tissue. Lastly, for the PSP samples, we selected samples that represented the quantitative neuropathology score distribution of the larger pool of available PSP samples.

## RNAseq of the bulk human brain

Generation of raw bulk RNAseq data in the study 2 cohort was previously described[6,46]. Briefly, libraries were prepared from total RNA using the TruSeq RNA Sample Prep Kit (Illumina, San Diego, CA) and sequenced on Illumina HiSeq 2000 instruments. For study 1, cDNA libraries were prepared using 200 ng of total RNA according to the manufacturer's instructions for the TruSeq RNA Sample Prep Kit v2 (Illumina, San Diego, CA). The concentration and size distribution of the completed libraries were determined using an Agilent Bioanalyzer DNA 1000 chip (Santa Clara, CA) and Qubit fluorometry (Invitrogen, Carlsbad, CA). Libraries were sequenced at six samples per lane, following Illumina's standard protocol using the HiSeq 3000/4000 PE Cluster Kit. The flow cells were sequenced as 100 ×2 paired-end reads on an Illumina HiSeq 4000 using the HiSeq 3000/4000 sequencing kit and HCS v3.3.20 collection software. Base-calling was performed using Illumina's RTA version 2.5.2.

## Single nucleus RNAseq of human brains

Frozen temporal cortex tissue samples were obtained from the Mayo Clinic Brain Bank. Total RNA from ~20 mg collected tissue was isolated to evaluate the quality of tissue. RNA integrity number (RIN) was determined via Agilent 2100 Bioanalyzer using RNA Pico Chip assay, and tissues that have RIN > 6.0 were utilized in nuclei isolation and single nucleus RNA sequencing (snRNAseq).

For each participant, 100 mg tissue sample was used for nuclei isolation using a modified protocol[47,48]. Samples were homogenized with 25 strokes of loose and tight pestle sequentially using dounce homogenizer in homogenization buffer (0.25 M sucrose, 25 mM KCl, 5 mM MgCl2, 20 mM tricine-KOH, pH 7.8, 1 mM DTT, 0.15 mM spermine, 0.5 mM spermidine, protease inhibitors, 5 μ g/mL actinomycin, 5 u/ μL recombinant RNAse inhibitor, and 0.04% BSA). IGEPAL (5%, Sigma, I8896) solution was added following stroke with the tight pestle to a final concentration of 0.32%. After 10 additional strokes, the tissue homogenate was filtered using a 30 μm cell strainer. Debris was pelleted by centrifugation at $500\,g$ for 5 minutes and washed with Wash and Storage Buffer (WSB, 1XPBS with 2%BSA and 5 u/μL recombinant RNAse inhibitor (Takara Bio, 2313 A)). The nuclei-containing supernatant was filtered again with a 30 μm cell strainer, followed by centrifugation at $500\,g$ for 10 minutes. After re-suspending the pallet in 700 μl cold PBS with 5 U/μl RNAse inhibitors, 300 μl debris removal solution (Miltenyi Biotech) was added, and the solution was gently mixed. Another 1 mL WSB was carefully overlaid on top of the nuclei solution. The supernatant was removed after centrifugation at $3000\,g$ for 10 minutes. The nuclei were washed once with WSB and pelleted after centrifugation for 10 minutes at $1000\,g$.

Isolated nuclei were sorted using fluorescence-activated nuclei sorting (FANS). Human Nuclear Antigen [235-1] (ab191181, Abcam) antibody was applied to the nuclei at 1:50 and incubated for 1 hour on ice. Concurrently, mouse IgG1, kappa monoclonal [15-6E10A7] isotype

was included as controls (ab170190, Abcam, 1:50). Goat anti-mouse Alexa488 secondary antibodies (ab150113, Abcam, 1:200) were incubated with the nuclei for 30 minutes on ice. The stained nuclei were reconstituted in WSB and sorted using BD FACSAria II sorter using the 70-micron nozzle with 70 psi sheath pressure and 1.5 ND filter. Examples of the gating strategy are shown in Figure S19. After quantifying the sorted nuclei using a hemocytometer in 0.04% trypan blue, a total of 1000 estimated nuclei at 700 nuclei/μl were loaded on the 10x Chromium microchip. Single cell gel beads-in-emulsion (GEMs) were generated by running Single Cell Instrument (10X Genomics) on the sorted nuclei. Chromium Single Cell 3' Gel Bead and Library Kit v3.1 (10X Genomics, No. 120237) and the Single Index Kit T Set A (10X Genomics, No. 1000213) were used to prepare the single nucleus RNAseq libraries according to the manufacturer's instructions. Qualities of libraries were checked using Agilent High Sensitivity DNA Kit (Agilent Technologies, 5067-1504) via Agilent 2100 Bioanalyzer. DNA libraries were sequenced at the Mayo Clinic Genome Analysis Core (GAC) using the Illumina NovaSeq 6000 sequencer.

## Bulk human brain RNAseq analyses and differential gene expression

Alignment and processing of the bulk RNAseq data in study 2 were previously described[46]. For study 1, raw paired-end reads were processed through MAP-RSeq pipeline v2.0[49]. MAP-RSeq removes reads of low base-calling Phred scores, aligns remaining reads to reference genome hg19 using TopHat aligner v2.0[50], and counts reads in genes and exons using subread. It obtains QC measures from pre- and post-alignment reads using the RSeQC toolkit and fastQC[51]. Subsequently, we identify samples for exclusion defined as those samples with high RNA degradation, low mappability, discordance between recorded sex and estimated sex, or based on principal component analysis (PCA) such that samples whose PC1 or PC2 are outside mean $+/- 3{*}SD$.

After excluding samples that fail QC, raw RNA read counts from remaining samples were normalized using R package CQN, which gives library size, gene length, and GC content adjusted expression values in the log2 scale. Based on the bimodal expression distribution, genes with low CQN values (CQN < 2 for study 1, CQN < −1 for study 2) were filtered out (Figure S1). Only genes that are expressed in both cohorts are used in the following analysis. Batch effect due to the source of the samples (the brain bank from which the sample was obtained) was corrected using the combat function from R package sva[52]. The associations between the batch-corrected, normalized, bulk gene expression and different clinicopathological traits (PSP diagnosis, TA, CB, NFT, TauTh, and overall pathology) were assessed for study 1 and study 2 separately using a multiple linear model adjusting for technical and biological covariates including sex, age at death, RIN, and sequencing flowcell. Gene expression levels were treated as a continuous dependent variable, while the traits of interests were treated as independent variables, coded as binary (PSP = 1, control = 0) or continuous (quantitative pathology scores) variables. For the association of gene expression levels with quantitative pathology scores, the tests were performed within PSP cases only, as controls do not have pathology scores. The association effect sizes (regression beta coefficients) in studies 1 and 2 were combined using an inverse-variance weighting meta-analysis, implemented in R package 'meta'[53]. A fixed-effect model was used for gene-trait associations when Higgin's and Thompson's $I^2$ heterogeneity value[54] was less than 0.3, while a random-effect model was used otherwise. Lastly, we adjusted for multiple testing using the Benjamini-Hochberg false discovery rate[55] (FDR).

Cell proportions were estimated based on the expression values of the top 100 BRETIGEA cell type marker genes[13] for astrocyte, endothelia, microglia, neuron, and oligodendrocyte using the Digital Sorting Algorithm (DSA)[56]. The association between PSP vs Control diagnosis and gene expression was assessed in a comprehensive model that adjusted for key covariates (sex, age at death, RIN, sequencing

flowcell) and the estimated cell proportions in each study. Meta-analysis was carried out using the same strategy outlined for the main model. Multiple testing was adjusted using FDR.

## Bulk human brain RNAseq co-expression network analyses

Gene expression networks were constructed using the R package WGCNA[15]. Before the network analysis, residuals from the batch-corrected, normalized gene expression data after adjusting for technical and biological covariates (sex, age, RIN, flowcell number) were generated for study 1 and study 2 separately. We calculated the adjacency matrices $A_{ij}$ using the pairwise biweighted midcorrelation (bicor), whose element $a_{ij} = bicor(g_i, g_j)$, for study 1 and study 2 separately. Topological overlap matrices (TOMs) were calculated based on their adjacency matrices as a signed network with a soft power threshold of 12 for studies 1 and 2 separately. Subsequently, after applying quantile scaling, we calculated the consensus TOM as the component-wise minimum of the TOMs from study 1 and study 2. The modules were identified using hierarchical clustering and dynamic treecut algorithm provided in the WGCNA package from consensus TOM.

From these initial modules, we identified the final modules as follows. We calculated the module eigengenes (ME) as the signed first principal component using gene expression of initial modules. The module memberships (MM) were calculated as the biweighted midcorrelation of the ME and gene expression. Closely related modules, defined as biweighted midcorrelation of ME higher than 0.8, were merged. We further examined and reassigned genes in each module based on their MM. Specifically, genes with negative MM were moved to the background module (module 0). In contrast, genes in the initial background modules were reassigned to the module with maximum MM if the maximum MM was greater than 0.5. We recalculated the ME and MM values using the final module definitions.

A t-distributed Stochastic Neighbor Embedding (t-SNE) plot (Figure S6) was made for final modules using gene expression of the combined cohort of study 1 and 2 through R package Rtsne as described in other studies[57]. In addition, statistics were calculated regarding the preservedness of final modules in studies 1 and 2 separately (Figure S6). We assessed the module preservations using the WGCNA::modulePreservation() function[58], and calculated preservation Z statistics with 100 permutations.

Biweighted midcorrelation between the ME and traits (PSP diagnosis, TA, CB, NFT, TauTh, and overall pathology) were calculated for each module. The association significance p values were adjusted to the number of modules using Bonferroni correction. Based on the module assignment, we tested if there were any significant enrichment of the top 100 BRETIGEA[13] cell type marker genes in any of the modules using a one-sided version of Fisher's exact test and a Bonferroni adjusted p-value cut-off of 0.05. Additionally, we calculated the gene ontology (GO) terms enriched in the modules using the R package anRichment[15], which calculates the statistical significance of the over-representation of GO terms based on hypergeometric distribution with an FDR-adjusted p-value of 0.05. Modules were manually annotated with the top GO terms from the list. (Figures S7, 8).

## Human Brain Single Nucleus RNAseq (snRNAseq) Validations

Raw snRNAseq data were processed and aligned using Cell Ranger version 4.0 (10X Genomics). Raw reads were mapped to the human reference genome hg38 using the STAR aligner. We obtained an average of 924 nuclei (standard deviation: 469, $N = 36$) per sample and performed quality control for each individual (Figure S9). First, nuclei with an extreme number of mapped UMIs (lower bound 1000, upper bound 98th percentile, or >=85,906.5, Figures S9a, b) or detected genes (lower bound 500, upper bound 98th percentile >=9,648.8, Figures S9c, d) were removed. Nuclei with more than 10% mitochondrial genes were also excluded (Figure S9e, f). At a sample level, PSP sample 12117 was removed from the analysis because it has an abnormal distribution in terms of the number of mapped UMIs and the number of detected genes (Figure S9g). PSP sample 12051 was removed due to the low number of nuclei ($N = 117$, Figure S9h). Genes that are only expressed (defined as having a count of greater than 0) in less than 6 nuclei are excluded (Figure S9i). The recorded sex for each individual was compared against that of the median expressions of 4 chromosome Y genes *RPS4Y1*, *EIF1AY*, *DDX3Y*, and *KDM5D*[59] to identify any potential mislabeled samples (Figure S9j).

Using Seurat R package[16], we merged UMI counts from each sample, performed library size normalization, and log transformation. We integrated the dataset using the Harmony package[60], treating each sample as its own batch. Nuclei were clustered using a Shared Nearest Neighbor (SNN) Graph implemented in Seurat[16] at a resolution of 0.4 with default parameters. We checked if there was significant enrichment (fold-enriched > 5, unadjusted p-value < 0.05) of nuclei from any diagnosis, sex, or sample in each cluster (Figures S10, 11).

To elucidate cluster cell type, we first identified cluster marker genes that are significantly (FDR < 0.05) over-expressed (logFC > 0.5) and universally expressed (percent nuclei > 70%) in each cluster compared to other clusters. We annotated each cluster based on the overlap of their highly overexpressed cluster-marker genes and known cell-type marker genes[61–66] (Fig. 3b): *GFAP* and *AQP4* for astrocyte, *VWF*, *PECAM1*, *FLT1* for endothelia, *NRGN* and *SLC17A7* for excitatory neurons, *GAD1* and *GAD2* for inhibitory neuron, *C3*, *CD74*, and *CSF1R* for microglia, *GRIN1*, *SNAP25*, and *SYT1* for neurons, *MBP*, *MOBP*, and *PLP1* for oligodendrocyte, *CSPG4*, *PDGFRA*, and *VCAN* for oligodendrocyte progenitor cells (opc), and *PDE5A* and *PDGFRB* for pericyte. Additional cluster cell type assignment was performed with published databases (Figure S12, 13)[13,17], which yielded consistent annotations.

Differentially expressed genes (DEG) between PSP and control nuclei were identified for each cluster through a hurdle model implemented in MAST R package[67], with adjustment of sex and age. We required DEG to be detected (UMI > 0) in at least 20% of the nuclei in a cluster. Multiple testing was adjusted for using FDR < 0.05. We performed pathways enrichment analysis using FUMA GWAS web service (v1.4.0)[68] using all 22,431 expressed snRNAseq genes as background. Multiple testing was adjusted for using the Benjamini-Hochberg method. MigSigDB v7.0 is used. A minimal overlap of 2 genes was required.

Pseudobulk DEG comparing the expression between PSP and control participants was performed for each cell type separately (Supplementary Data 4, Table S9). Briefly, the pseudo-count matrix was obtained by adding the read count from all the nuclei of the same individual. The pseudo count between PSP and control individuals was compared using a negative binomial generalized linear model implemented in R package edgeR[69], with adjustment of sex and age. P-values were corrected for multiple testing using FDR.

For each nucleus in the snRNAseq analysis, its expression scores, which reflect the expression levels of a selected set of genes, were calculated as the average expression levels of the module genes based on the module definition from the 16 WGCNA modules using the AddModuleScore() function implemented in Seurat[16]. We assessed the overlap between module genes and PSP vs Control DEGs in all clusters. Significance was calculated using a one-sided Fisher's exact test. Only single-nucleus DEGs that are also detected in bulk RNAseq are used in the analysis.

We utilized the snRNAseq data to validate and filter the bulk RNASeq DEGs based on our prioritization approach (Fig. 4a). We focused on the significant bulk DEGs (FDR < 0.05 in the meta-analysis of study 1 + 2) from the PSP-associated co-expression modules that are also enriched in brain cell types (i.e., modules 3, 4 and 6). There were 4969 such DEGs, 550 of which were also highly-connected network hub genes with module membership (MM) > 0.7. We filtered these 550 significant PSP DEG hub genes by selecting those that are also significant DEGs in the snRNAseq data cluster corresponding to the

bulk module cell type and that have a concordant direction of change between bulk and snRNAseq. Specifically, for M3 candidate genes, we selected those that are significantly down-regulated in any of the oligodendrocyte clusters (CL0 or CL26). For the M4 candidate genes, we selected those that are upregulated in the astrocyte cluster (CL2). Lastly, for the M6 candidate genes, we selected those that are upregulated in any of the microglia (CL3, CL27), endothelia (CL19), pericytes (CL12, CL17), or astrocytes (CL2) clusters.

To elucidate additional cell population, the astrocyte nuclei were subjected to subclustering analysis (Figure S17). Briefly, read counts of nuclei from astrocyte cluster CL2 were normalized, log-transformed, and harmonized using the same procedures for the full dataset. Clustering was carried out at a resolution of 0.3 with default parameters. Subcluster marker genes were identified using the Wilcoxon Rank Sum test. Differentially expressed genes between PSP and Control nuclei in each subcluster were identified with the same approach used for the full dataset.

### Validation with rTg4510 Tauopathy Mouse Model Brain RNAseq

Bulk brain RNAseq data of the tauopathy mouse model rTg4510[18] was retrieved from the AD Knowledge Portal (syn3157182). Briefly, the dataset consists of bulk RNAseq from forebrain samples of rTg4510 mice overexpressing human P301L tau (4R0N) and wild-type non-transgenic (nonTg) littermate control mice. Using the RNAseq data from the 4.5- and 6-month-old (Table S6) mice, we normalized the gene expression values using CQN, followed by QC to check for outliers or mismatched sex (Figures S20, 21). Read counts between rTg4510 and nonTg mice were compared in each dataset within each age group using a negative binomial generalized linear model implemented in R package edgeR[69], with adjustment of sex and RIN. DEGs were identified based on an FDR-adjusted p-value of 0.05. Mouse genes were mapped to their human orthologs using R package biomaRt[70] and ensemble version 105.

Using the mouse DEG information, we investigated the 155 human DEGs (Fig. 4a) that were significant and had congruent changes in both bulk and snRNAseq and were also hubs in the prioritized modules M3, M4, and M6. These human DEGs were considered validated in the mouse model if significant at an FDR-adjusted level of 0.05 in either 4.5- or 6-month rTg4510 mouse brains and had the same direction of change as in humans.

### Validation with Drosophila Tau Model Experiments

High confidence PSP glial DEGs with significant and congruent changes in human bulk and snRNAseq and rTG4510 mouse model brains were experimentally validated in a *Drosophila melanogaster* model expressing human wild-type Tau protein[20]. Briefly, *Drosophila* orthologs of the human gene are obtained by querying DIOPT[71] version 8. We obtained all available RNAi lines against the *Drosophila* homologues of the high confidence PSP glial DEGs (Fig. 4a) from the Bloomington Drosophila Stock Center (BDSC), Transgenic RNAi Project (TRiP) collection or requested from laboratories within the fly research community[72] (Tables S9, 10). We crossed the *GMR-GAL4/CyO; UAS-hTau/TM3*, to RNAi lines and selected progeny that co-expressed both human Tau and RNAi *(GMR-GAL4/+; UAS-hTau/UAS-RNAi, where the UAS-RNAi can be on any chromosome)*. We first compared the morphology of their eyes with the control flies expressing only the tau protein using a −4 to 4 semi-quantitative scale where 0 indicated no change compared to the tau-expressing control. A positive score indicates enhancement/exacerbation of the well-described eye neurodegeneration pathology, whereas a negative score indicates suppression/rescue of this pathology. A score of 4 indicated that the flies had no eyes, whereas a score of −4 indicated that the eyes were indistinguishable from that of the wild-type control. If the flies fail to eclose, we indicate the phenotype as being lethal as previously described[40].

To assess the impact of the three top tau-toxicity suppressor genes, we aged the progenies for 5 days at 25 °C before pictures of the fly left eyes were taken. The severity of the tau-induced eye degeneration was assessed blindly based on the following categories: loss of bristle (0-1), size (0-1), color (0-2), the presence of necrotic patterns (0-2), the collapse of the eye (0-2), and the loss of ommatidia (0-2), where a higher score indicates a more severe phenotype. Scores were provided by two independent evaluators separately, and the average scores were used for analysis.

### Data sharing and interactive web tool

We built a web application to enable the interactive exploration of our data called the *PSP RNAseq Atlas* (https://rtools.mayo.edu/PSP_RNAseq_Atlas/), which is freely available to the broad research community. The *PSP RNAseq Atlas* is searchable by gene name and provides results for human bulk RNAseq PSP DEG, neuropathology associations, snRNAseq, rTG4510 brain associations, *Drosophila* tau model results and any available therapies against the prioritized genes based on the Drug Gene Interaction Database (DGIdb) v4.0[73]. In addition, all human and mouse RNAseq data in this manuscript is available via the AD Knowledge Portal (https://adknowledgeportal.synapse.org). The AD Knowledge Portal is a platform for accessing data, analyses and tools generated by the Accelerating Medicines Partnership (AMP AD) Target Discovery Program and other National Institute on Aging (NIA)-supported programs to enable open-science practices and accelerate translational learning. Data is available for general research use according to the following requirements for data access and data attribution (https://adknowledgeportal.synapse.org/DataAccess/Instructions).

### Reporting summary

Further information on research design is available in the Nature Portfolio Reporting Summary linked to this article.

## Data availability

All human and mouse RNAseq data in this manuscript is available via the AD Knowledge Portal (https://adknowledgeportal.synapse.org). The AD Knowledge Portal is a platform for accessing data, analyses and tools generated by the Accelerating Medicines Partnership (AMP AD) Target Discovery Program and other National Institute on Aging (NIA)-supported programs to enable open-science practices and accelerate translational learning. Data is available for general research use according to the following requirements for data access and data attribution (https://adknowledgeportal.synapse.org/DataAccess/Instructions). An overview of all the data generated and used in this study can be found on the manuscript landing page (https://doi.org/10.7303/syn51361408). The bulk brain and single-nucleus RNAseq data generated in this study have been deposited in the AD Knowledge Portal under the Mayo RNAseq study (accession ID: syn5550404) and the Mayo Clinic Brain Molecular Signatures of Alzheimer's Disease (MC-BrAD) study (accession ID: syn51298412) [https://doi.org/10.7303/syn2580853]. The rTg4510 mouse RNAseq data used in this study are available in the AD Knowledge Portal under the Tau and APP mouse model (TAUAPPms) study (accession ID: syn3157182). All summary results for human bulk RNAseq PSP DEG, neuropathology associations, WGCNA analysis, snRNAseq, rTg4510 brain associations, *Drosophila* tau model results, and any available drugs against the prioritized genes based on the Drug Gene Interaction Database (DGIdb) v4.063 is available through our web application (https://rtools.mayo.edu/PSP_RNAseq_Atlas/) and the Supplementary Information. Source data are provided with this paper.

## Code availability

The analysis and visualization codes are available via the AD Knowledge Portal (https://doi.org/10.7303/syn51671671). The source code

powering the web application is also hosted on the AD Knowledge Portal (https://doi.org/10.7303/syn51674938).

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

## Acknowledgements

We would like to thank the patients and their families for their participation, without whom these studies would not have been possible. The results published here are in whole or in part based on data obtained from the AD Knowledge Portal (https://adknowledgeportal.org). The Mayo RNAseq study data was led by Dr. Nilüfer Ertekin-Taner, Mayo Clinic, Jacksonville, FL as part of the multi-PI U01 AG046139 (MPIs Golde, Ertekin-Taner, Younkin, Price) using samples from The Mayo Clinic Brain Bank. Data collection was supported through funding by NIA grants P50 AG016574, R01 AG032990, U01 AG046139, R01 AG018023, U01 AG006576, U01 AG006786, R01 AG025711, R01 AG017216, R01 AG003949, CurePSP Foundation, and support from Mayo Foundation. Study data included samples collected through the Sun Health Research Institute Brain and Body Donation Program of Sun City, Arizona, USA. The Brain and Body Donation Program is supported by the NINDS (U24 NS072026, National Brain and Tissue Resource for Parkinson's Disease and Related Disorders); the NIA (P30 AG19610, Arizona Alzheimer's Disease Core Center); the Arizona Department of Health Services (contract 211002, Arizona Alzheimer's Research Center); the Arizona Biomedical Research Commission (contracts 4001, 0011, 05-901, and 1001, to the Arizona Parkinson's Disease Consortium); and the Michael J. Fox Foundation for Parkinson's Research. Control brain samples from the University of Kentucky were from the UK-Alzheimer's Disease Research Center (P30 AG072946). Additional support for these studies was provided by the NINDS grant R01-NS080820 (NET), NIA grant R01-AG061796 (NET), NIA grant U19-AG074879 (NET), Alzheimer's Association Zenith Fellows Award (NET), and NCATS grant TL1 TR002380 (YM). We thank the Mayo Clinic Genome Analysis Core (GAC), Co-Directors, Julie M. Cunningham, PhD and Eric Wieben, PhD, and supervisor Julie Lau, for their collaboration in collection of omics data. We thank Drs. Jada Lewis, Karen Duff, David Westaway and David Borchelt for generating these lines of transgenic mice and providing us access to them. We thank Dr. Joshua Shulman, Dr. Tom Lee, and Yarong (Linda) Li from the Baylor College of Medicine for sharing additional *Drosophila* stocks and providing technical and intellectual contributions to the experiments. We thank Jason P. Sinnwell from Mayo Clinic Quantitative Health Sciences and Andrea P. Laack from Mayo Clinic IT for assisting with the deployment of our web application, PSP RNAseq Atlas. We thank Matthew A. Bockol from Mayo Clinic IT and Abby Linden from Sage Bionetworks for assisting with the deposition of the human RNAseq data.

## Author contributions

M.A. and N.E.T. conceived the study. Y.M., X.W., M.A., and N.E.T. wrote the manuscript; Y.M., X.W., J.S.R. and Z.Q. performed quality control and data analysis. J.E.C. consulted on the statistical methods. M.A., M.M.C.,

Ö.İ., T.A.P., T.N., K.G.M., S.L., F.Q.T. performed samples selection and preparation. Y.M., J.L.C., J.G., and K.Z. performed Drosophila validation. D.W.D. provided tissue from the Mayo Clinic Brain Bank and neuro-pathologically characterized Mayo brain samples. P.N. and L.V.E. provided tissue from the University of Kentucky ADRC Biospecimen Repository. T.E.G. provided the mouse data. A.O.M., M.A., and Y.M. performed data uploading, meta data generation. Y.M. performed the analysis, generated the tables and figures, and developed the web application. X.W., M.A., N.E.T., J.G., K.Z., M.M.C., J.S.R., Ö.İ., and T.A.P. provided advice on interpreting the results. N.E.T. oversaw the study and provided direction, funding, and resources. All authors read and approved the final manuscript.

## Competing interests

Mayo Clinic has filed a provisional patent application related to the content of the manuscript. These competing interests do not affect the design, conduct, or reporting of this research. The remaining co-authors declare no competing interests.
