## [Peer Review File · Nature Communications]

Cross Species Systems Biology Discovers Glial DDR2, STOM, and KANK2 as Therapeutic Targets in Progressive Supranuclear PalsyREVIEWER COMMENTS

Reviewer #1 (Remarks to the Author):

The manuscript submitted by Min et al. applied a cross-species systems biology approach to human brain transcriptome data at bulk tissue and single nucleus resolutions, a tau mouse model, and experimental validations in a *Drosophila* tau model. Authors utilized bulk RNA-seq data and single nucleus RNA-seq data from human PSP and control brain samples to uncover the transcriptome changes that lead to the progression of the PSP and formation of tau lesions. Then authors prioritized genes from glial-associated modules and validated them in a tau mouse model and a *drosophila* tau model. Initially the authors identified 2,513 differentially expressed genes and then narrowed down to 550 hub genes, 155 snRNA seq DEGs (human brain), 21 DEGs in rTg451018 tau mouse model and finally to 11 *drosophila* genes. The study also claimed to identify a few therapeutic targets (STOM, KANK2 and DDR2) based on their RNAi screening of *Drosophila* tau model.

Major commen

1) The major concern is low number of nuclei obtained in snRNA-seq data. In total the study profiled 2526 astrocytes and 2500 microglia from 34 samples, which means, on average, 70 to 74 astrocytes/microglia per sample. Considering the sparsity of single-cell technology, I am not confident with the conclusion drawn from low number of nuclei obtained in snRNA-seq data. In addition, it would help if authors could provide snRNA-seq UMAPs by samples and by library-prep batch.

2) As shown in the supplemental table S4, there are 16 controls and 18 PSP human brain samples, it is unclear if all control samples are from Mayo or if some are from UKY. And additionally, which brain banks are the brain from under the source 'Neither' in table S4? It would be better if authors could provide more detailed metadata for snRNA-seq, as well as UMAPs by age and by brain banks.

3) It is unclear how was the association test performed for neuropathology in Figure 1B? The authors could provide a summary table (as excel file and not as a figure) of all identified DEGs in supplementary files, especially DEGs related to neuropathology. Figure 1A shows a table of the number of upregulated and downregulated PSP genes from bulk RNA sequencing respected to both the neuropathology and diagnosis. Still, more detailed information was not included in this paper. There is only a table in the supplementary of PSP DEGs within 1Mbp of PSP GWAS loci (Table S3). A similar summary table for Figure 1C highlighting cell type markers as DEGs would be much appreciated.

4) It would be great if the authors could explain how the enrichment analysis (eg how is the p-value obtained) was performed for each cell type in Figure 2B. Also can the authors present the WGCNA dendrogram and examples of genes network (network plot of genes) of the M3, M4, and M6. But more importantly, I wonder why M6, a mixed microglia-endothelia module, is selected for the representation of the module, but the rest emphasis of the following paper is on cell type-specific changes in glial cells. The authors' Figure 3D expression score (module score) in snRNA-seq data is much higher in endothelial

cells. Furthermore, in Figure 3E, genes from bulk RNA-seq M6 module have a higher overlap percent with Astrocytes clusters and a more significant p-value. So, is M6 a microglia-endothelia module or astrocyte module or mixed glial module?

5) The authors used the percent of overlapped genes between PSP-associated modules from bulk RNA-seq and snRNA-seq DEGs to support the idea of cell-type enrichment M3, M4, and M6 modules. But one could argue that M4 and M6 at least are not preserved in snRNA-seq. Half or more of the overlapped DEGs from M4 and M6 have an opposite direction from the same module in bulk RNA-seq to snRNA-seq. These module genes are not “co-expressed” in snRNA-seq data. I would suggest projecting modules from bulk RNA-seq to snRNA-seq or running WGCNA on snRNA-seq again and checking what is preserved.

6) The authors evaluated the rough eye phenotype score (-4 to +4) as a parameter to identify the effect of RNAi treatments which seems conceivable. However, Tau transgenic flies die prematurely, and I wonder if the treatments helped to improve the lifespan of the flies. Also, evaluation of neurodegeneration/ Vacuolar pathology (as neurodegeneration in *Drosophila* is commonly accompanied by vacuoles) in a tau transgenic and treated flies would be helpful to understand and compare those therapeutic targets.

7) The authors retrieved Bulk brain RNA seq data of the tauopathy mouse model ‘rTg451018’ from the AD Knowledge Portal. However, snRNA seq study/data in rTg451018 tau mouse model can give a better insight of cell specific (especially glial specific) changes. Also designing a knockdown/knockout study in rTg451018 tau mouse model would be helpful for evaluating the therapeutic efficacy of STOM, KANK2 and DDR2 to ameliorate the cognitive deficits. For example, astrocyte specific overexpression/down regulation of DDR2 might help to understand/validate the role of DDR2 as a therapeutic target in PSP.

8) The manuscript needs to be thoroughly checked. In line 170 a correction made by the authors is visible (track changed?).

Reviewer #2 (Remarks to the Author):

In this manuscript, Min Y et al. integrated bulk RNA-Seq data from 281 PSP samples and 127 controls from two brain banks. They also performed single nucleus RNAseq data from 34 samples from PSP and control brains. These integrations resulted in a list of candidate transcripts DE in PSP brains compared to controls. The authors then used publicly available transcriptome data from the forebrain of a mouse model of tauopathy (rTg4510 human P301L mutation). They validated the leading candidates in *Drosophila* expressing human wild-type tau.

Major

The samples are from three different brain banks. It is more concerning that the control samples are from two other brain banks (UKY) and Banner. It is unclear why, if the primary source of the PSP samples is from Mayo, there are two separate studies with statistically different demographic characteristics. There is no mention of the postmortem interval time in any cohorts or analyses.

It would be advisable that the authors include separate CQN histograms for the cases and controls of each study.

A separate analysis comparing DEG among controls from different brain banks are also advisable. A PCA analysis will help to support the proposed DEG in PSP vs. controls.

The rationale for using meta-analysis instead of harmonizing the data using batches and other important covariates for each study needs to be clarified. It is also clear that an independent validation cohort needs to be added to the current version of the manuscript.

It is unclear whether the "site" or source of the brain samples was included in the analysis since it is a significant variable, as reported in S1T.

Reporting the surrogate variable table while using the sva package will be advisable, which could give a precise view of how the batch effect and other latent sources of variation might have affected the data.

In the snRNA-Seq data, individual samples dominating the number/proportion of nuclei per cluster in FS8, were they included in the analyses? They are clearly different than the rest of the samples.

It is worrisome that CL27 (Microglia) is dominated by one sample 1933., CL25 (oligodendrocytes) is mostly from sample 11488, and CL24 (Neu) is from 1964. The enrichment of the DEG in modules from bulk RNA seq in glia needs to be clarified. The astrocyte and microglia clusters from snRNA-Seq data have an over-representation from individual samples. They should have been excluded from the analyses. Therefore, the analyses of the human data could be presented in a better and more transparent way.

Adding more details (metrics) on the QC process of the snRNA-Seq could improve the reproducibility of this study.

Minor

A better rationale for choosing the temporal cortex in PSP cases where there is not much tau pathology
How many samples failed QC of the bulk RNA Seq of study 1?

From the description of the methods, one might think that there is not only a batch effect but that the bulk RNA-Seq pipeline used in study one (the current one) is different from the one used for samples from study 2.

A methodological consideration and limitation of the snRNA-Seq are using FANS, which has been demonstrated to induce artifacts in microglia transcriptome, which is not discussed in this manuscript. Fig 3, panel D, Module 6 shows that endothelial cells and pericytes exhibit higher levels of expression of genes in that module.

How was the enrichment of cell type-specific genes performed with the modules' DEG?

From Table S4 is not possible to determine if the controls are from the same biobank or not. Again there is no mention of PMI, RIN, or additional variables.

Reviewer #3 (Remarks to the Author):

In this very interesting manuscript, Min and co-authors, present their work from cross-species systems biology study of PSP. In this project, the authors analyze a large scale human brain transcriptome data of bulk RNA-seq, and single nucleus of PSP and control donor, and follow up their results in the rTg4510 mouse model and performed RNAi experiments Drosophila. Overall, they present novel discoveries of robust glial transcriptomic perturbations and identified three potential therapeutic targets in PSP.

I have few questions/suggestions for the manuscript:

1. Results section “PSP brains have vast and replicable transcriptome perturbations in glia-enriched genes”:

a- Are the differences identified in the magnitude detected PSP vs CT (N=2,513) and tau tau neuropathology (N=74 or less) related to power issue?

b- Are there any genes differentially expressed in some of the compression?

c- Are these results driven from one of the two studies employed?

d- These results suggest that PSPS is associated with additional changes in the TCX not captured by tau neuropath variables. Are any of these changes observed in the TCX of other neuropath diseases, such as Alzheimer’s? Are there any additional neuropath variable, or inferred variable (such as proportion of oligos, or other cell-type) that can be approximated using cell deconvolution, that can be explaining these differences, as suggested in lines 106-114? Additional analyses of sources of unwanted variation (e.g. SVA) might reveal additional interesting and important insights that can conciliate the differences in these analyses.

e- As currently stated, it is not clear for this reviewer what was tested in the section “top PSP vs. control brain expression changes defined as lowest 5% FDR and top 5% $|\logFC|$ ”.

f- Data from the Cerebellum was also generated for some of these brain donors. How are the TCX changes compared to Cerebellum?

2. It would be important to communicate the readers the relational that guided the selection of the 18 PSP and 16 CT for single-cell experiments from the far larger number of brains with bulk. In addition, Is there any power analyses suggesting that single-cell from these would provide similar of more insights than larger cohorts

3. Sentence in lines 166-8 should be edited. It is not clear what contributed to <1% of the nuclei.

4. Regardless specific association to any of the 28 clusters, did the authors identify any of the changes in the cellular population structure identified using bulk RNA-seq in the single-cell data? I would suggest to perform additional analyses that can help determine this question, which I think is central to this work. Does pseudo-bulk analyses of single-cell data provides similar results to the bulk RNA-seq? Is it a power issue? More importantly, nowadays several authors isolate each of the cell-types from the data set (digital sorting) and recluster the data to identify specific changes. I think this approach can be highly informative for this data.

5. I would suggest the authors to try to refine/extend the gene networks identified using RNA-seq using the single-cell data. In addition, gene regulatory network analyses (beyond co-expression) can be highly informative for these key modules identified in bulk and followed up in single-cell,
6. Did the authors consider using additional data (e.g. Glasauer et al - <https://doi.org/10.1016/j.stemcr.2022.07.011>) to replicate their findings in oligodendrocytes not captured in the rTg4510 model?
7. Additional data, statistical test significance should be provided for drosophila RNAi experiments. Did the authors evaluate and capture any additional phenotypic (expected/unexpected) changes beyond eye morphology?
8. Are any of the compounds associated with the networks identified that can also ameliorate neurodegeneration in the drosophila models?
9. Are the cells in clusters (CL23, CL24, CL26) undifferentiated/precursor cells? Additional investigation should be done to determine if these are artifacts and discard these in that case.
10. Please provide the QC parameters and add some plots/information for the snRNAseq QC.
11. I suggest the authors to provide additional descriptions of the large variety of data and analyses shared on their website. As currently stated, readers might miss many of these great resources.
12. In this manuscript author mentioned the three potential targets (DDR2, STOM and KANK2) but they have focused on DDR2 inhibitor. It would be more informative if they mentioned the list of drugs/inhibitor that can be used against other targets.

Cross Species Systems Biology Discovers Glial *DDR2*, *STOM*, and *KANK2* as Therapeutic Targets in Progressive Supranuclear Palsy
NCOMMS-22-50578-T

Response to reviewer comments

R1-1.....	4
R1-2.....	8
R1-3.....	9
R1-4.....	9
R1-5.....	12
R1-6.....	14
R1-7.....	15
R1-8.....	15
R2-1.....	16
R2-2.....	17
R2-3.....	18
R2-4.....	18
R2-5.....	19
R2-6.....	20
R2-7.....	20
R2-8.....	20
R2-9.....	21
R2-10.....	21
R2-11.....	21
R2-12.....	22
R-Minor-2-1.....	23
R-Minor-2-2.....	23

R-Minor-2-3.....	23
R-Minor-2-4.....	24
R-Minor-2-5.....	24
R-Minor-2-6.....	24
R-Minor-2-7.....	25
R3-1	26
a. Are the differences identified in the magnitude detected PSP vs CT (N=2,513) and tau neuropathology (N=74 or less) related to power issue?	26
b. Are there any genes differentially expressed in some of the compression?.....	27
c. Are these results driven from one of the two studies employed?	28
d. These results suggest that PSPS is associated with additional changes in the TCX not captured by tau neuropath variables. Are any of these changes observed in the TCX of other neuropath diseases, such as Alzheimer’s? Are there any additional neuropath variable, or inferred variable (such as proportion of oligos, or other cell-type) that can be approximated using cell deconvolution, that can be explaining these differences, as suggested in lines 106-114? Additional analyses of sources of unwanted variation (e.g. SVA) might reveal additional interesting and important insights that can conciliate the differences in these analyses.	29
e. As currently stated, it is not clear for this reviewer what was tested in the section “top PSP vs. control brain expression changes defined as lowest 5% FDR and top 5% logFC “.	31
f. Data from the Cerebellum was also generated for some of these brain donors. How are the TCX changes compared to Cerebellum?	32
R3-2	32
R3-3	32
R3-4	33
R3-5	37
R3-6	37
R3-7	37
R3-8	39
R3-9	40
R3-10	40
R3-11	41
R3-12	42

We thank the Editors and the Reviewers for their careful review of our manuscript, their comments, input, and suggestions. We are grateful for all the positive comments from the Reviewers. Below, we provide the point-by-point responses to all comments. We believe that the modifications to address the comments of our reviewers have further enhanced the impact of our study.

Specific points provided by the reviewers are shown in blue font and any text edits we made to the manuscript in response to the reviews are shown in red underlined font in this document. In our submission, we provide two versions of our Main and Supplementary manuscript files; one in which all changes are tracked and another one in which all changes are accepted.

To facilitate our reviewers' and Editor's review of our responses to each of their 37 points, we include an index in the front page of this document, which provides hyperlinks to each of the responses. We also use hyperlinks within the body of this response document. We have responded to every critique including conducting new analyses and experiments. We include in our revised manuscript 2 new supplementary figures, new sections to main figures, edited versions of tables, as well as 5 new **Supplementary Data Tables** (each with multiple sub-tables) and new text, as detailed below. In addition, in this document we provide 15 **response figures** that encompass results we generated addressing our reviewers' comments. We believe that these changes further strengthened our manuscript and look forward to receiving the new comments of the Reviewers based on this modified resubmission and a favorable decision on our work.

Reviewer 1:

The first reviewer summarizes the manuscript as follows: “The manuscript submitted by Min et al. applied a cross-species systems biology approach to human brain transcriptome data at bulk tissue and single nucleus resolutions, a tau mouse model, and experimental validations in a Drosophila tau model. Authors utilized bulk RNA-seq data and single nucleus RNA-seq data from human PSP and control brain samples to uncover the transcriptome changes that lead to the progression of the PSP and formation of tau lesions. Then authors prioritized genes from glial-associated modules and validated them in a tau mouse model and a drosophila tau model. Initially the authors identified 2,513 differentially expressed genes and then narrowed down to 550 hub genes, 155 snRNA seq DEGs (human brain), 21 DEGs in rTg451018 tau mouse model and finally to 11 drosophila genes. The study also claimed to identify a few therapeutic targets (STOM, KANK2 and DDR2) based on their RNAi screening of Drosophila tau model.”

We thank Reviewer 1 for their summary and provide a point-by-point response to each of their comments as follows:

R1-1: The major concern is low number of nuclei obtained in snRNA-seq data. In total the study profiled 2526 astrocytes and 2500 microglia from 34 samples, which means, on average, 70 to 74 astrocytes/microglia per sample. Considering the sparsity of single-cell technology, I am not confident with the conclusion drawn from low number of nuclei obtained in snRNA-seq data. In addition, it would help if authors could provide snRNA-seq UMAPs by samples and by library-prep batch.

We thank the reviewer for their comments. We would like to point out that the transcriptional perturbation conclusions drawn in our manuscript arise not only from the snRNAseq data but also from the bulk brain transcriptome data from two independent cohorts which revealed replicable findings in a total of 281 PSP and 127 control brains, complementary findings in brain transcriptome of a mouse model of tauopathy and functional validations in a Drosophila tau model. As such, our conclusions are supported by one of the largest brain transcriptome cohorts in PSP and cross-species validations.

In regards to the snRNAseq data in our manuscript, we demonstrate that there are many differentially expressed genes between PSP and control nuclei that are significant after FDR-adjustment in oligodendrocyte cluster CL0, astrocyte cluster CL2, and microglia cluster CL3 (Table S5). While we acknowledge that there may be difficulties in detecting DEGs with snRNAseq, especially for cell types that are less abundant, we should emphasize that we utilize the brain snRNAseq data not as a discovery cohort but to “further replicate the glial gene expression changes from bulk brain RNAseq detected in our two independent studies” (Results section: **Single-nucleus RNAseq captures glial expression changes in PSP brains**). Indeed, our snRNAseq data “corroborates the cell-enrichment annotations for the PSP-associated bulk RNAseq modules M3, M4, and M6”. Further, snRNAseq data consistent with that in bulk transcriptome helped prioritize the high-confidence genes for cross-species validations (**Figure 4**).

With respect to the number of nuclei assessed in our study, we have comparable number of nuclei per sample with other snRNAseq studies. For example, Mathys et al¹ reported a total of 3392 astrocytic nuclei and 1920 microglial nuclei among 48 individuals in their DEG comparison (Alzheimer’s disease vs. control), averaging around 70 and 40 nuclei per individual. To our knowledge, the only other single-cell/nuclei transcriptome profiling of PSP brains is performed by Sharnma et al² in all male donors (3 PSP cases and 3 controls). Even though they profiled a higher number of nuclei (N = 45,559), the dataset is yet published or hosted. Therefore, our brain snRNAseq dataset represents a sizable PSP vs. control study with comparable numbers of nuclei to others.

We should emphasize that we have applied analytic methods to account for the inherent sparsity in snRNAseq data (**Methods Section: Human Brain Single Nucleus RNAseq (snRNAseq) Validations**). We adopted

a hurdle model implemented in the MAST package³ in our initial submission. Additionally, we removed genes with high sparsity, as only genes that are expressed in over 20% of the nuclei could be considered as DEG.

Additionally, to further address Reviewer 1's comment, we conducted DEG analyses using the pseudobulk approach⁴. We performed pseudobulk DEG (pbDEG) analysis at donor level by aggregating UMI counts from the same cell type in each individual. We calculated the gene logFC between PSP and control donors in each cell type using the edgeR package. We then correlated the logFC obtained from pbDEG and MAST DEG from clusters with more than 1% nuclei (**Response Figure 1, Panel A**). We showed that the correlations are largely positive. Additionally, the correlations are the highest when the pseudobulk cell type matches the cell type of the cluster. For example, the logFC calculated using MAST in CL2 (astrocyte cluster) has the highest correlation with the logFC calculated using astrocyte pseudobulk data. We then focused on the glial clusters with more than 1% of total nuclei. We showed in **Response Figure 1, Panel B** that the MAST-calculated logFC of CL2 genes has a high correlation with the pseudobulk logFC of astrocytes (Spearman rho = 0.78, $p < 2.2 \times 10^{-16}$). Similarly, we observed highly significant correlations of MAST-calculated logFC and pseudobulk logFC in microglia (Spearman rho = 0.68, $p < 2.2 \times 10^{-16}$), oligodendrocyte (Spearman rho = 0.8, $p < 2.2 \times 10^{-16}$), as well as OPC, and pericyte clusters (data not shown).

Lastly, we report pseudobulk DEG results for the top three genes, *DDR2*, *KANK2* and *STOM*. These genes showed consistent changes with the pseudobulk approach as we observed using MAST. Specifically, *DDR2* was up-regulated in astrocyte cluster CL2 using MAST and is also up-regulated in astrocyte pseudobulk data (logFC = 0.361), with a nominal significance level of 0.089. Glial gene *KANK2* and *STOM* are up-regulated in multiple glia cell types (microglia, astrocyte, oligodendrocyte and pericyte). The levels of significance are less (higher p values) given the smaller sample sizes using the pseudobulk approach, however the directionality and cell type assignments are consistent between these approaches. Taken together, the consistency of the snRNAseq results with bulk transcriptome, cross-species validation results and with the pseudobulk approach underscores the validity of our conclusions.

Response Figure 1

A

Correlations of PSP vs Ctrl DEG logFC between MAST in each cluster (x-axis) and pseudobulk in the corresponded cell type

B

C

GeneName	Cell	Cluster	MAST		Pseudobulk	
			logFC	pvalue	logFC	pvalue
DDR2	ast	CL2	0.139	6e-06	0.361	0.089
	mic	CL3	NA	NA	-0.070	0.919
	oli	CL0	NA	NA	0.111	0.763
	opc	CL7	NA	NA	-0.136	0.749
	per	CL12	0.018	0.774	-0.077	0.743
	CL17	-0.077	0.409			
KANK2	ast	CL2	0.047	1e-06	0.726	0.013
	mic	CL3	NA	NA	0.023	0.962
	oli	CL0	NA	NA	1.159	0.007
	opc	CL7	NA	NA	0.371	0.497
	per	CL12	0.033	0.855	-0.241	0.333
	CL17	0.036	0.090			
STOM	ast	CL2	0.087	1e-04	0.435	0.010
	mic	CL3	0.084	0.081	0.509	0.018
	oli	CL0	NA	NA	1.221	0.003
	opc	CL7	NA	NA	0.194	0.691
	per	CL12	0.081	0.017	0.538	0.033
		CL17	0.028	0.757		

Following the reviewer's suggestions, we also prepared the additional UMAPs with different variables. The first reviewer asked for UMAPs by samples and library-prep batch. We provided these and also UMAPs by diagnosis, sex, age and RIN, as well. These new UMAPs were added to the manuscript as **Figure S9** shown below. We included the cell type assignment UMAPs from manuscript **Figure 3A** for the reviewer's convenience. As can be seen in this new **Figure S9**, there is no systematic enrichment of particular clusters by the different variables.

Figure S9 (new figure added to the manuscript)

Figure 3A

R1-2: As shown in the supplemental table S4, there are 16 controls and 18 PSP human brain samples, it is unclear if all control samples are from Mayo or if some are from UKY. And additionally, which brain banks are the brain from under the source 'Neither' in table S4? It would be better if authors could provide more detailed metadata for snRNA-seq, as well as UMAPs by age and by brain banks.

We thank the reviewer for the feedback. We agree that the demographics table for the snRNAseq participants can be improved and now provide a **modified Table S4**. To clarify, all snRNAseq participants were from the Mayo brain bank, as described in line 417 of the original manuscript. We added the "Bulk RNAseq Overlap" category to report any sample used in both bulk RNAseq and snRNAseq. For example, 16 out of the 18 PSP samples were also used in the bulk RNAseq experiment, and all 16 were used in Study 1 of the bulk RNAseq experiment. Similarly, 7 out of the 16 control samples also had bulk RNAseq data. The remaining 11 samples (9 control, 2 PSP) were not used in either study 1 or study 2. We understand that the information might be confusing to the readers and not add too much value, so we removed it from the **modified Table S4**.

We also added more metadata to the table, including RIN. Additional UMAP figures by age and RIN (as well as other variables) were also generated in response to R1-1.

Modified Table S4

	Control, N = 16 ¹	PSP, N = 18 ¹	Overall, N = 34 ¹	p-value ²
RIN	7.85 (6.47, 8.22)	7.55 (6.90, 7.80)	7.70 (6.73, 8.17)	>0.9
Sex				0.5
Female	9 (56%)	8 (44%)	17 (50%)	
Male	7 (44%)	10 (56%)	17 (50%)	
Age	90 (87, 93)	86 (82, 91)	89 (83, 92)	0.10
Source				
Mayo	16 (100%)	18 (100%)	34 (100%)	

¹Median (IQR); n (%)

²Wilcoxon rank sum test; Pearson's Chi-squared test

R1-3: It is unclear how was the association test performed for neuropathology in Figure 1B? The authors could provide a summary table (as excel file and not as a figure) of all identified DEGs in supplementary files, especially DEGs related to neuropathology. Figure 1A shows a table of the number of upregulated and downregulated PSP genes from bulk RNA sequencing respected to both the neuropathology and diagnosis. Still, more detailed information was not included in this paper. There is only a table in the supplementary of PSP DEGs within 1Mbp of PSP GWAS loci (Table S3). A similar summary table for Figure 1C highlighting cell type markers as DEGs would be much appreciated.

We thank the reviewer for their comments. We have made the full list of all our results available via a new tool as described in our manuscript: “We share our findings and data via our new interactive application tool PSP RNAseq Atlas (https://rtools.mayo.edu/PSP_RNAseq_Atlas/)”. Nevertheless, we agree that additional supplementary tables might also be helpful for readers. Therefore, we provide additional supplementary data (**Supplementary Data 1**) that reports the association statistics (beta, p, q) in each study and the meta-analysis results for all phenotypes (PSP vs Ctrl, neuropathology). In this same table, we also annotated if a gene is a cell-type marker gene in the column cell_marker. We believe readers can easily apply filters to investigate their genes of interest. We also modified the original manuscript text to better reflect the testing being performed with respect to neuropathology.

R1-4: It would be great if the authors could explain how the enrichment analysis (eg how is the p-value obtained) was performed for each cell type in Figure 2B. Also can the authors present the WGCNA dendrogram and examples of genes network (network plot of genes) of the M3, M4, and M6. But more importantly, I wonder why M6, a mixed microglia-endothelia module, is selected for the representation of the module, but the rest emphasis of the following paper is on cell type-specific changes in glial cells. The authors’ Figure 3D expression score (module score) in snRNA-seq data is much higher in endothelial cells. Furthermore, in Figure 3E, genes from bulk RNA-seq M6 module have a higher overlap percent with Astrocytes clusters and a more significant p-value. So, is M6 a microglia-endothelia module or astrocyte module or mixed glial module?

We thank the reviewer for the comments that enables us to clarify these points. We used a one-sided Fisher’s exact test to assess the enrichment of cell marker genes in any particular WGCNA module (**see Response Figure 2** below). In this case, for each combination of cell type and WGCNA module, we build 2x2 contingency tables with the number of genes that are cell type marker genes and are in module (**a**), the number of genes that are in the module but not a cell type marker gene (**b**), the number of genes that are cell type marker gene but not in module (**c**), and the number of expressed genes that are not marker gene nor module gene (**d**). We used the following R command to perform the test and obtained enrichment p values: `fisher.exact(cbind(c(a,b),c(c,d)), alternative="greater")`.

Response Figure 2

Fisher's exact test (FET): Is the overlap of Set 1 and Set 2 statistically significant?
 2 x 2 Contingency table:

$$\begin{Bmatrix} |Set 1 \cap Set 2| & |Set 1 \setminus Set 2| \\ |Set 2 \setminus Set 1| & |background \setminus (Set 1 \cup Set 2)| \end{Bmatrix}$$

As requested, we provided the WGCNA dendrogram (**Response Figure 3**) and network plot highlighting the top 20 genes by module membership for M3, M4, and M6 (**Response Figure 4**). We also plotted the prioritized genes that we nominated as shown in **Figures 4 and 5**.

Response Figure 3

Response Figure 4

Top 20 genes by MM

Prioritized genes
(See figure 4A)

We apologize for the confusion in the **Figures 3D-E**. Please allow us to clarify as follows. In analyzing the bulk transcriptome (**Results section: Glial cell-enriched gene co-expression network modules are associated with PSP**), a module was selected for further analysis if a) its eigengene significantly correlates with PSP diagnosis and b) it has significant enrichment of cell type marker genes from any cell type. As noted in this section, “We found an enrichment of oligodendrocyte genes (Bonferroni-adjusted $p < 2.22E-16$) in the down-regulated M3, consistent with our prior work⁸. M6, which is up in PSP, is enriched for endothelial (Bonferroni-adjusted $p < 2.22E-16$) and microglial (Bonferroni-adjusted $p = 5.29E-4$) marker genes, whereas M4, also up in PSP, is an astrocyte-enriched module (Bonferroni-adjusted $p < 2.22E-16$).” We should add that M6 was also nominally (but not Bonferroni-adjusted) enriched in astrocyte marker genes.

We next investigated M3, M4, and M6 genes in the context of snRNAseq (**Figure 3, Results section: Single-nucleus RNAseq captures glial expression changes in PSP brains**). For this part, we included all cell types in the snRNAseq for completeness, even some rare cell types such as the endothelia that make up $<1\%$ of nuclei. In calculating the module scores, we found evidence from snRNAseq that M6 genes are also expressed in astrocytes and pericytes (**Figures 3D-E**). The module scores provide support that M6 genes are enriched with not only endothelial and microglial genes, but also astrocyte and pericyte genes. As we stated in the manuscript, the high module scores “underscore that they [cell marker genes] are enriched in but not exclusive to specific cell types”. We should acknowledge that because snRNAseq assesses gene expression at a higher resolution, it may capture aspects of the transcriptome (such as enrichment of a gene in a variety of cell types) that may have been masked in bulk RNAseq.

Taken together, we want to clarify that M6 is a mixed-glia module that is statistically enriched with microglial and endothelial genes in the bulk transcriptome. The naming of the modules is purely based on the bulk RNAseq data, and we used the snRNAseq data (both the expression score in 3D and the overlaps in 3E) as a means of annotation and validation. We made the following modification to our manuscript to better reflect our intention and for clarification (**Results section: Single-nucleus RNAseq captures glial expression changes in PSP brains**).

To provide additional context and annotations for the genes from the three glial-marker enriched WGCNA co-expression modules, M3, M4 and M6, that are replicably associated with PSP in bulk RNAseq from two studies, Expression scores, calculated based on the average expression levels of these modules’ genes for each nucleus in the snRNAseq, were analyzed across clusters for each cell type (Figure 3D). For completeness, all clusters were included in the analysis, even for those that accounted for less than 1% of the total nuclei.

For the microglia/endothelia-enriched M6 hub genes, given the high module scores and the enrichment of snRNAseq DEGs from this module genes within these cell types (**Figure 3D-E**), we selected those that are up-regulated snRNAseq in microglia, endothelia, pericytes or astrocytes, resulting in 40 genes.

R1-5: The authors used the percent of overlapped genes between PSP-associated modules from bulk RNA-seq and snRNA-seq DEGs to support the idea of cell-type enrichment M3, M4, and M6 modules. But one could argue that M4 and M6 at least are not preserved in snRNA-seq. Half or more of the overlapped DEGs from M4 and M6 have an opposite direction from the same module in bulk RNA-seq to snRNA-seq. These module genes are not “co-expressed” in snRNA-seq data. I would suggest projecting modules from bulk RNA-seq to snRNA-seq or running WGCNA on snRNA-seq again and checking what is preserved.

We appreciate the reviewer for the comment. We would like to clarify that, as we stated in R1-1 and R1-4, the use of the snRNAseq data together with the bulk RNAseq WGCNA (**Figure 3D-E**) results is for the validation of the cell marker gene enrichment and to provide complementary single nucleus expression information which

were used for the prioritization and subsequent validations (**Figure 4**). We agree that co-expression modules constructed from bulk expression data might not necessarily be preserved in snRNAseq data. We again want to acknowledge that because snRNAseq assesses gene expression at a much higher resolution, it can capture other aspects of the transcriptome, such as the up- regulation of some genes and down-regulation of others in the same cluster that might have been masked in bulk RNAseq that typically captures the most abundant gene expression directionality.

Nevertheless, we followed the first reviewer’s suggestions and projected the bulk WGCNA module definition onto the snRNAseq dataset. Specifically, we calculated the module preservedness (WGCNA function `modulePreservation()`) using pseudobulk RNAseq data in each cell type to avoid sparseness. Cell types that account for less than 1% of the total nuclei were included in the analysis for completeness. We found that the oligodendrocyte module M3 is well-preserved in oligodendrocytes. Likewise, the astrocyte module M4 is preserved in astrocyte nuclei. M6 which has enrichment of multiple cell type marker genes does not have high preservation (preservation score >10) in any cell type. These findings are aligned with **Figure 3D**.

Response Figure 5

Our modules were constructed from bulk data alone, therefore it is possible that some gene expression changes are not detected at bulk tissue levels but are detected at single-cell levels. In addition to the technical differences between bulk and single-cell/nucleus RNAseq, the granularity of the two methods is also different. For example, our group⁵ and others^{6,7} have shown unique microglial and astrocyte populations that would not have been detected at bulk tissue levels, respectively. Additionally, the enrichment of cell marker genes does not indicate that the module is exclusively specific to one cell type but not the other. We already highlight these nuances in our paper (**Results section: Single-nucleus RNAseq captures glial expression changes in PSP brains**): “Directionality of gene expression changes for these modules’ genes is highly consistent between bulk RNAseq

and snRNAseq oligodendrocyte cluster, although other glial clusters have DEGs that change in both directions, suggesting that single nucleus data may capture more subtle gene expression changes that may be missed in bulk data.”

Importantly and also as described in R1-1, the snRNAseq data was used to prioritize the genes from the bulk RNAseq modules M3, M4 and M6 for the cross-species validations (**Figure 4**). Indeed, of the top 3 genes highlighted in our study, two of them from M6, *KANK2* and *STOM* were further confirmed including the functional studies in the *Drosophila* model.

R1-6: The authors evaluated the rough eye phenotype score (-4 to +4) as a parameter to identify the effect of RNAi treatments which seems conceivable. However, Tau transgenic flies die prematurely, and I wonder if the treatments helped to improve the lifespan of the flies. Also, evaluation of neurodegeneration/ Vacuolar pathology (as neurodegeneration in *Drosophila* is commonly accompanied by vacuoles) in a tau transgenic and treated flies would be helpful to understand and compare those therapeutic targets.

We appreciate the reviewer for these suggestions. We agree that a more sophisticated evaluation of the fly phenotype is helpful in increasing our confidence in the results. We repeated the RNAi experiment for the top three genes (*DDR2*, *KANK2*, and *STOM*) and blindly scored the fly eyes again using a more quantitative system and two independent examiners. We described the results in R3-7. Briefly, our additional data provides further support to our initial findings in *Drosophila* where RNAi for all three genes led to a statistically significant reduction in tau-related toxicity (**Figure 5**).

We understand that as shown in the Wittmann et al paper⁸ the Elav-Gal4 driving expression of 4R tau can produce additional phenotypes, such as reduced lifespan and brain pathology in flies. However, we want to clarify that, unlike the GMR-Gal4 driver we used in our paper, which allows the expression of tau and RNAi in both neuronal and glial cells, the Elav-Gal4 is a neuronal-specific driver. Because we showed in our data that the three target genes are all up-regulated in glial cells, a system that drives expression only in the neuronal cells- but not the glial cells- (i.e. Elav-Gal4 driver) would not be an appropriate one to pursue functional validations for our prioritized genes.

Furthermore, significant pathology in Elav>TauWT flies usually does not develop until 25 days past eclosing and can survive for up to 60 days. For this reason, it would not be possible to complete the vacuolar pathology evaluation and longevity assay in this model within the 3-months timeline allocated for the responses to the reviews. In the future, our targets can be further investigated using a glial-specific driver, such as *repo-Gal4*⁹. To avoid confusion to the readers, we clarified our *Drosophila* models use GMR drivers and added relevant discussions:

In Results section: Cross-species prioritization and screening of glial gene expression changes in PSP.

To determine whether experimentally perturbing levels of high-confidence PSP glial genes would impact tau-related neurodegeneration, we used a *Drosophila* tau model with the GMR-Gal4 driver.

In Discussion:

Additionally, unlike the repo-Gal4 driver, the *Drosophila* GMR-Gal4 driver is not glial specific as it drives the expression of tau in all cell types.

R1-7: The authors retrieved Bulk brain RNA seq data of the tauopathy mouse model ‘rTg451018’ from the AD Knowledge Portal. However, snRNA seq study/data in rTg451018 tau mouse model can give a better insight of cell specific (especially glial specific) changes. Also designing a knockdown/knockout study in rTg451018 tau mouse model would be helpful for evaluating the therapeutic efficacy of STOM, KANK2 and DDR2 to ameliorate the cognitive deficits. For example, astrocyte specific overexpression/down regulation of DDR2 might help to understand/validate the role of DDR2 as a therapeutic target in PSP.

We appreciate the reviewer’s suggestions on the use of mouse models to investigate the functional outcomes of the genes and their implications in drug development. We agree that snRNAseq or scRNAseq studies in rTG4510 mice will be helpful in understanding the cell-specific changes. However, to our knowledge, there are no publicly available single-cell RNAseq datasets from rTG4510 mice. Lisi et al reported scRNAseq of rTG4510 mice¹⁰ as a pre-print, but they focused their analysis on neuronal populations and did not report DE analysis.

We also agree that a knockdown study with a PSP mouse model is certainly an important step in translating our findings into therapy. However, it may take years to generate such mouse models, which is certainly useful for the future but not feasible nor within the scope for the current study. Nevertheless, we added additional text discussing the usefulness of such mouse models in terms of translation. In the **Discussion**, we stated:

... This will be useful in selecting appropriate models for experimental validations and pre-clinical therapeutic trials. Moreover, carrying out cell-type-specific overexpression or knockdown of the target genes in mouse models and assessing the clinical impact will be informative to validate the therapeutic target prior to clinical applications.

R1-8: The manuscript needs to be thoroughly checked. In line 170 a correction made by the authors is visible (track changed?).

We thank the reviewer for carefully reading through our manuscript and spotting the styling inconsistency. We have made the modification to make the manuscript more publication-ready.

Reviewer 2:

The second reviewer summarizes the manuscript as follows: “In this manuscript, Min Y et al. integrated bulk RNA-Seq data from 281 PSP samples and 127 controls from two brain banks. They also performed single nucleus RNAseq data from 34 samples from PSP and control brains. These integrations resulted in a list of candidate transcripts DE in PSP brains compared to controls. The authors then used publicly available transcriptome data from the forebrain of a mouse model of tauopathy (rTg4510 human P301L mutation). They validated the leading candidates in *Drosophila* expressing human wild-type tau.”

We thank Reviewer 2 for their summary and provide a point-by-point response to each of their comments as follows:

R2-1: The samples are from three different brain banks. It is more concerning that the control samples are from two other brain banks (UKY) and Banner. It is unclear why, if the primary source of the PSP samples is from Mayo, there are two separate studies with statistically different demographic characteristics.

We thank the reviewer for the comments. We acknowledge that although all RNAseq experiments were performed at a single sequencing center at Mayo Clinic, the source of the specimen should be closely examined. In our original analysis, we applied batch correction methods to adjust for any potential variability due to the specimen source. Other potential confounders are also adjusted in our regression model. Further, we demonstrated the replicability of the DEG and WGCNA results using the bulk RNAseq data from two independent cohorts.

Nonetheless, to address our second reviewer’s comment further, we also performed the PSP vs Control DEG analysis with samples just from the Mayo Brain Bank and compared the findings to the analyses including samples from the other brain banks. As shown in the following **Response Figure 6**, adding samples from other sites did not alter the direction of DEGs, as indicated by the overwhelming positive correlation of logFC (**Response Figure 6A, top**). As one might expect, the p values change, likely due to the reduction in sample size in the Mayo-only analysis (**Response Figure 6A, bottom**). Further comparison of the three DEG categories (Up in PSP, Down in PSP, or N.S) in All vs Mayo-only analysis (**Response Figure 6B**) further highlighted the consistent patterns. Importantly, we chose to include control samples from outside of the Mayo Clinic Brain Bank primarily to increase the power of the analysis. We followed up with a power analysis of the linear model we used (5 independent variables) using an alpha of 0.05 and a power of 80%. We found that the minimal detectable effect size of using samples from all sites (0.0319) is 78% of that of using just Mayo samples (0.0407), demonstrating the increased power in using all samples.

In summary, our manuscript already demonstrates robust and replicable findings using two independent cohorts. We already applied the appropriate adjustments in our analyses. We further demonstrated that a single Brain Bank (i.e. Mayo-only) vs. all samples (all Brain Banks) yield highly consistent findings with respect to direction and size of effect for the DEGs (**Response Figure 6A-B**). We also show that inclusion of all samples leads to increase power, as expected, which justifies the inclusion of these samples (**Response Figure 6C**).

Response Figure 6

R2-2: There is no mention of the postmortem interval time in any cohorts or analyses.

We agree with the reviewer that postmortem interval (PMI) is a valuable measure of tissue quality, which might ultimately influence the sequencing results. Despite this, PMI is not always readily available, especially when the Brain Bank serves as a tissue repository for samples that are sent from other institutions, rather than being the single site that acquires all samples directly. This is the case for the Mayo Clinic Brain Bank that serves as the main repository for PSP brain samples for the CurePSP Foundation, where samples are provided from many different institutions.

However, the RNA integrity number (RIN) also reflects the quality of the sample and the degradation of RNA, especially in the context of RNA sequencing experiments. Our lab has been using RIN as a technical variable in prior transcriptome studies^{11,12}. In this manuscript, we included RIN as a covariate in bulk RNAseq DEG and WGCNA. We did not include RIN in the snRNAseq analysis because the RIN in PSP and control samples are not significantly different (Wilcoxon rank sum test, $p = 0.9724$).

We again emphasize that our bulk DEG and WGCNA findings from two independent cohorts are robustly replicable. Our findings in snRNAseq further prioritized genes that have consistent findings in this dataset, further demonstrating the validity of the transcriptome data.

R2-3: It would be advisable that the authors include separate CQN histograms for the cases and controls of each study.

We thank the reviewer for the suggestion, which we now incorporated into our revised manuscript. We agree that it would be informative to add separate CQN distributions for PSP and control subjects. We generated these figures and showed that the CQN distributions are indeed similar in both PSP and control brains. We depict the new **Figure S1** below.

R2-4: A separate analysis comparing DEG among controls from different brain banks are also advisable.

Our understanding of the suggestion is to compare the gene expression levels of control participants from different brain banks. We performed such DE analysis within controls by comparing the source-corrected CQN expression levels of non-Mayo controls (University of KY samples for study 1, Banner samples for study 2) to that of the controls from Mayo, coding Mayo controls as references. We implemented a similar linear regression model as that we used for the PSP vs Control DEGs, where we adjusted age, sex, RIN, and flowcell.

Results from study 1 and study 2 were combined using meta-analysis in the same fashion as we reported in the manuscript. We define significant DE as those with FDR less than 0.05.

As shown in the **Response Figure 7** below, none of the genes are significantly differentially expressed between the controls across brain banks. Therefore, the differences in expression between PSP and controls are unlikely to be confounded by the source of the controls. These findings along with those from R2-1 demonstrate that the DEG findings in our study are robust and not driven by potential Brain Bank-related differences.

Response Figure 7

R2-5: A PCA analysis will help to support the proposed DEG in PSP vs. controls.

We thank the reviewer for suggesting the PCA analysis. We calculated the first 50 principal components (PC) after centering and scaling the source-corrected CQN expression data using the PSP vs Control DEG in each study). For both studies, we observed separation of samples in the first PC by diagnosis, supporting the vast and replicable PSP vs. control DEG results in our manuscript.

Response Figure 8

R2-6: The rationale for using meta-analysis instead of harmonizing the data using batches and other important covariates for each study needs to be clarified. It is also clear that an independent validation cohort needs to be added to the current version of the manuscript.

We thank the reviewer for suggesting harmonizing the expression data as an alternative. We agree that a carefully planned harmonization of the expression is viable. However, the samples from the two bulk brain RNAseq studies were sequenced independently at different time point using different pipelines. We list the major differences in the RNAseq pipeline in the following table:

Parameter	Study 1	Study 2 ¹²
Library Preparation	TruSeq RNA Sample Prep Kit v2	TruSeq RNA Sample Prep Kit
Sequencer	Illumina HiSeq 4000	Illumina HiSeq 2000
RSeQC	Version 2.6.2	Version 2.3.2
R	R-3.2.0	R-2.13.0

Additionally, although MAPRseq was utilized to process the raw reads for both RNAseq datasets, there are nevertheless differences in the versions of the software used. We believe it will be very challenging to accurately account for all the differences between studies if we try to harmonize the data. Therefore, we chose a more conservative approach by performing meta-analysis of the DEG, similar to the study by Wan et al [PMID: 32668255], where they combined the Alzheimer’s Disease vs Control DEG across 7 datasets.

We also agree that validation is important in any study. It is precisely for this reason that we already have built-in replications and validations in our whole study paradigm (please see the **graphical abstract** and **Figure 4**). In fact, because the two bulk RNAseq cohorts in our study are entirely independent, the significant bulk RNAseq DEGs and modules are replicated in those two cohorts, thereby strengthening our conclusions. We provided further validations with snRNAseq, followed by a mouse tauopathy model and functional validations in a Drosophila tau model. We note that there are very few published RNAseq studies targeting PSP patient brains, let alone providing findings in two separate independent cohorts as we did in our study. Our study that already has two independent bulk RNAseq datasets, in addition to human snRNAseq, tauopathy mouse model, and Drosophila experimental data will provide an important and robust resource to the research community.

R2-7: It is unclear whether the "site" or source of the brain samples was included in the analysis since it is a significant variable, as reported in S1T.

We appreciate the reviewer’s comment, and agree that “site” or “source” is indeed a significant variable. To address this issue, we adjusted for the source of the sample using the “combat” function provided by the package “sva”. As shown in response R2-4, the source-corrected CQN values, which were used for the PSP vs control DEG analysis, are not significantly associated with the sample source. We also updated our text for clarification. In **Methods section: Bulk Human Brain RNAseq Analyses and Differential Gene Expression**, we state:

Batch effect due to the source of the samples (the brain bank from which the sample was obtained) was corrected using the combat function from R package sva¹³.

R2-8: Reporting the surrogate variable table while using the sva package will be advisable, which could give a precise view of how the batch effect and other latent sources of variation might have affected the data.

We would like to clarify that we did not include any surrogate variable in our model, as we used the “combat()” function¹⁴ in the “sva” package¹³ to adjust for a known technical variable (tissue source). The “combat()” function implements an empirical Bayesian (EB) framework to estimate the effect of technical variables (batch) on the gene expression data and subsequently remove such effects. Since we did not include any surrogate variable analysis, we do not have a surrogate variable table.

R2-9: In the snRNA-Seq data, individual samples dominating the number/proportion of nuclei per cluster in FS8, were they included in the analyses? They are clearly different than the rest of the samples. It is worrisome that CL27 (Microglia) is dominated by one sample 1933., CL25 (oligodendrocytes) is mostly from sample 11488, and CL24 (Neu) is from 1964.

We acknowledge that certain clusters have an over-representation of individuals/samples, however, most of these are small clusters, meaning the total nuclei account for less than 1% of the total nuclei of the dataset. In fact, all three clusters mentioned (CL24, CL25, and CL27) were such clusters. Indeed, we already stated in the old **Figure S8** (new number is **Figure S10**) that “Eight clusters had statistically significant enrichment of nuclei from a small number of samples, depicted with sample numbers on the clusters. All but one of these 8 clusters (CL12) constitute <1% of total nuclei.” We also now include UMAPs by multiple variables (new **Figure S9**) including sample IDs, as described in R1-1. This supports the conclusion that most of the clusters have contributions from all samples. We note that we analyzed these small clusters for completeness, but they do not harbor many differentially expressed genes (**Table S5**). Importantly, none of our conclusions were based on those clusters.

R2-10: The enrichment of the DEG in modules from bulk RNA seq in glia needs to be clarified.

We thank the reviewers for their comments and would like to also refer them to R1-4. We calculate the enrichment of snRNAseq DEG in each WGCNA module in a similar fashion as how we calculated the enrichment of cell type marker genes in each module. We used a one-sided Fisher’s exact test to check the probability of the number of overlapping genes being more than expected. In this case, we only kept snDEGs that are also detected in bulk RNAseq. We chose the entire expressed genes as the background.

We modified the method text accordingly to be more clear about our approach. In **Methods section: Human Brain Single Nucleus RNAseq (snRNAseq) Validations**, we now state:

... from the 16 WGCNA modules using the AddModuleScore() function implemented in Seurat. We assessed the overlap between module genes and PSP vs Control DEG in all clusters. Significance was calculated using a one-sided Fisher’s exact test. Only single-nucleus DEGs that are also detected in bulk RNAseq are used in the analysis.

R2-11: The astrocyte and microglia clusters from snRNA-Seq data have an over-representation from individual samples. They should have been excluded from the analyses. Therefore, the analyses of the human data could be presented in a better and more transparent way.

We refer the reviewer to R1-1 and R2-9. We already clearly stated in the old **Figure S8** (new number is **Figure S10**) that “Eight clusters had statistically significant enrichment of nuclei from a small number of samples, depicted with sample numbers on the clusters. All but one of these 8 clusters (CL12) constitute <1% of total nuclei.” Based on old **Figure S8** (new **Figure 10**), it is true that microglial cluster CL27 has an over-representation

of sample 1933. However, we did not use the results from this cluster for the prioritization of target genes because it has a low number of nuclei and accounts for less than 1% of the total nuclei in the study, as we have also explained in R2-9. For the snRNAseq clusters that we did utilize in our prioritization analyses, there was no significant over-representation of individual samples.

As for transparency of presenting findings, we could not agree more. As stated in our manuscript, “We share our findings and data via our new interactive application tool PSP RNAseq Atlas (https://rtools.mayo.edu/PSP_RNAseq_Atlas/).” This tool harbors all of our results in an easily accessible format. Furthermore, we now include 5 new Supplementary Data for both our bulk and snRNAseq data in our submission (please see R1-3 and below table). The new Supplementary Data file names, content, relatedness to manuscript figures and presence in our PSP RNAseq Atlas are depicted in the table below. As can be seen, all relevant results from the Supplementary Data are already available on our new interactive application tool PSP RNAseq Atlas (shiny app), and the results from 4 out of 5 new Extended Data tables can be directly exported as csv-formatted tables. Nonetheless, we wanted to ensure that our reviewers and readers have access to all of our data in an entirely transparent manner and expect that our mode of data sharing will enable this.

Filename	Description	Related to	In web app?
Supplementary Data 1.xlsx	Differential gene expression analysis results using the bulk RNAseq data	Figure 1	Yes
Supplementary Data 2.xlsx	WGCNA module eigengenes, module membership, enrichment of cell type marker genes, and correlation with different phenotypes	Figure 2	Yes
Supplementary Data 3.xlsx	Enriched gene ontology terms in each WGCNA module	Figure 2	Yes
Supplementary Data 4.xlsx	Significantly differentially expressed genes between PSP and control nuclei in each cluster from the snRNAseq analysis	Figure 3	Yes*
Supplementary Data 5.xlsx	Small molecules and gene interaction status, retrieved from DGIdb	Figures 4/5	Yes

*: data available in Web App figures only

R2-12: Adding more details (metrics) on the QC process of the snRNA-Seq could improve the reproducibility of this study.

We appreciate the reviewer’s comment. We included all the code used to perform QC, which would allow others to rerun our analysis. Nevertheless, we realize that it will make it easier if we also describe the QC parameters in the methods section for the readers. Therefore, we added additional QC information in the manuscript. We now have a new **Figure S8** which we present in R3-10. We also modified the **Methods Section: Human Brain Single Nucleus RNAseq (snRNAseq) Validations**, as follows:

Raw snRNAseq data were processed and aligned using Cell Range version 4.0 (10X Genomics). Raw reads were mapped to the human reference genome hg38 using the STAR aligner. We obtained an average of 924 nuclei (standard deviation: 469, N = 36) per sample and performed quality control for each individual (Figure S8). Nuclei with more than 10% mitochondrial genes were excluded. Additionally, nuclei with an extreme number of detected genes (lower bound 500, upper bound 98th percentile $\geq 9,648.8$) or mapped UMIs (lower bound 1,000, upper bound 98th percentile, or $\geq 85,906.5$) were also removed. After removing low-quality nuclei, genes that are only expressed (defined as having a count of greater than 0) in less than 6 nuclei are excluded. The recorded sex for each individual was compared against that of the median expressions of 4 chromosome Y genes

RPS4Y1, EIF1AY, DDX3Y, and KDM5D¹⁵ [PMID: 23829492] to identify any potential mislabeled samples.

Two PSP samples were removed because of low nuclei count, after which 18 PSP and 16 control samples remained for analysis.

R-Minor-2-1: A better rationale for choosing the temporal cortex in PSP cases where there is not much tau pathology.

We appreciate the reviewer's comment and would like to clarify that the temporal cortex (TCX) is especially suitable for this study precisely because it does not have abundant pathology. TCX has less tau pathology and neuronal loss in PSP than other brain regions¹⁶, making the expression changes less likely to be confounded by the cell proportion changes. Despite this, we have shown robust transcriptional changes in the TCX in PSP brains^{11,17,18}, demonstrating the utility of this brain region in the discovery of gene expression changes that are less likely due to gross cellular proportion changes. TCX is also a more easily accessible region, where there is typically more sample availability and less susceptibility to variations in acquisition of gray matter. We added text to further clarify our rationale for use of TCX in the manuscript, **Results section: PSP brains have vast and replicable transcriptome perturbations in glia-enriched genes** as follows:

Due to the relatively low degree of gross tau pathology in TCX^{11,12}, we hypothesized that this region would be less susceptible to confounding factors associated with downstream consequences of the disease, such as neuronal loss⁶, making the expression changes less likely to be secondary to these confounds. TCX is also a more easily accessible region, where there is typically more sample availability and less susceptibility to variations in acquisition of gray matter.

R-Minor-2-2: How many samples failed QC of the bulk RNA Seq of study 1?

We reported the post-QC demographic information in **Supplementary Table S1**, which consists of 257 individuals (199 PSP, 58 control) in study 1. We took a similar approach as we previously published for study 2 data¹². Specifically, we sequenced 265 brain tissue samples from PSP and control individuals in study 1. We removed 4 samples (2 PSP, 2 control) due to mismatches between the recorded sex and the expected expression of all chromosome Y genes. Additionally, two samples (1 PSP, 1 control) have a high gene body coverage ratio (>3), measured as the ratio between read number values at the 80th and 20th percentile using RSeQC. It suggests a potential 3' bias and was removed. Lastly, we removed two samples (2 control) that were not obtained from either Mayo Clinic Bank or the University of Kentucky.

R-Minor-2-3: From the description of the methods, one might think that there is not only a batch effect but that the bulk RNA-Seq pipeline used in study one (the current one) is different from the one used for samples from study 2.

We thank the reviewers for the comments. It is correct that within each study, there are "batch effects", which we refer to as technical factors that influence the sequencing results. A typical example is the source of the brain tissue (which brain bank did the sample come from). We addressed the "batch effect" using two methods: a correction of the CQN-normalized count using the combat function in the "sva" package¹³ (for the sample source) and adjusting for covariates in our models (for other technical and biological covariates).

The two bulk RNAseq datasets were indeed generated at different times and were analyzed by different versions of the analytic pipelines, as also discussed in R2-6. For clarity, we edited the **Methods Section: RNAseq of the bulk human brain**, as follows:

Generation of raw bulk RNAseq data in the study 2 cohort was previously described^{6,43}. Briefly, libraries were prepared from total RNA using the TruSeq RNA Sample Prep Kit (Illumina, San Diego, CA) and sequenced on Illumina HiSeq 2000 instruments.

R-Minor-2-4: A methodological consideration and limitation of the snRNA-Seq are using FANS, which has been demonstrated to induce artifacts in microglia transcriptome, which is not discussed in this manuscript.

We acknowledge the limitations of single-nucleus RNAseq including the assessment of only nuclear transcripts, the potential to miss microglial subtypes compared to scRNAseq¹⁹ and the potential for inducing artifacts. We would like to clarify, as we stated in R1-1, R1-4, and R1-5 that the snRNAseq dataset is used for the prioritization and subsequent validations (**Figure 4**). Nevertheless, we added the limitation to our discussion section:

Despite these strengths and insights, there are limitations to our study. We acknowledge that single-nucleus RNAseq captures only the transcriptional activity within the nucleus and might potentially miss biological changes relevant to PSP, which may be detectable by single cell RNAseq (scRNAseq). Nevertheless, we were still able to validate in snRNAseq the bulk RNAseq expression changes in the corresponding nuclei type that led to the discovery of *DDR2*, *KANK2*, and *STOM* as potential therapeutic targets for PSP. Moreover, snRNAseq allowed the use of frozen brain tissue, which made it possible to include as many PSP samples as possible. There are no PSP-specific mouse models....

R-Minor-2-5: Fig 3, panel D, Module 6 shows that endothelial cells and pericytes exhibit higher levels of expression of genes in that module.

We agree that the expression scores for M6 genes are high in endothelial cells and pericytes. We discussed this issue in R1-4 and refer our reviewer to this section for details. Briefly, based on the bulk RNAseq WGCNA results, M6 is defined as a mixed-glia module with enrichment of endothelial and microglial marker genes. Therefore, it is not surprising to observe high snRNAseq expression scores in endothelia for this module. In calculating the module scores, we found evidence from snRNAseq that M6 genes are also expressed in astrocytes and pericytes (**Figures 3D-E**). The module scores provide support that M6 genes are enriched with not only endothelial and microglial genes, but also astrocyte and pericyte genes. As we stated in the manuscript, the high module scores “underscore that they [cell marker genes] are enriched in but not exclusive to specific cell types”. We should acknowledge that because snRNAseq assesses gene expression at a higher resolution, it may capture aspects of the transcriptome (such as enrichment of a gene in a variety of cell types) that may have been masked in bulk RNAseq.

R-Minor-2-6: How was the enrichment of cell type-specific genes performed with the modules' DEG?

We understand from this question that the second reviewer is requesting clarifications on the calculation of enrichment of cell type-specific genes in each WGCNA module. We stated this in our **Method section: Bulk Human Brain RNAseq Co-expression Network Analyses** as follows:

Based on the module assignment, we tested if there were any significant enrichment of the top 1000 BRETIGEA²⁰ cell type marker genes in any of the modules using a one-sided version of Fisher's exact test and a Bonferroni adjusted p-value cut-off of 0.05.

Specifically, as stated in R1-4, we calculated the statistical significance of the overlap between each WGCNA module gene and the cell type marker genes using a one-sided Fisher's exact test. We chose all the 22,560 expressed genes in bulk RNAseq as the background for the tests. For any BRETIGEA marker gene that is not part of the background, we removed them from the calculation. Adjustment for multiple testing was done using the Bonferroni method.

R-Minor-2-7: From Table S4 is not possible to determine if the controls are from the same biobank or not. Again there is no mention of PMI, RIN, or additional variables.

We thank our reviewers for their suggestion which enabled us to modify this table. Please see our response in R1-2, where our new and edited **Table S4** is depicted. All 34 samples for the snRNAseq data came from the Mayo Clinic brain bank. We show RIN, sex, age and source of the samples in this table. As described in, R2-2, we did not have PMI on many samples, because Mayo Clinic serves as an archival brain bank, receiving post-mortem tissue from many other institutions, where this information is not always available.

However, the RNA integrity number (RIN) also reflects the quality of the sample and the degradation of RNA, especially in the context of RNA sequencing experiments. Our lab has been using RIN as a technical variable in prior transcriptome studies^{11,12}. In this manuscript, we included RIN as a covariate in bulk RNAseq DEG and WGCNA. We did not include RIN in the snRNAseq analysis because the RIN in PSP and control samples are not significantly different (Wilcoxon rank sum test, $p = 0.9724$).

Reviewer 3:

The third reviewer summarizes the manuscript as follows: “In this very interesting manuscript, Min and co-authors, present their work from cross-species systems biology study of PSP. In this project, the authors analyze a large scale human brain transcriptome data of bulk RNA-seq, and single nucleus of PSP and control donor, and follow up their results in the rTg4510 mouse model and performed RNAi experiments *Drosophila*. Overall, they present novel discoveries of robust glial transcriptomic perturbations and identified three potential therapeutic targets in PSP.”

We thank Reviewer 3 for their summary and positive comments. We provide a point-by-point response to each of their comments as follows:

R3-1: Results section “PSP brains have vast and replicable transcriptome perturbations in glia-enriched genes”:

- a. Are the differences identified in the magnitude detected PSP vs CT (N=2,513) and tau neuropathology (N=74 or less) related to power issue?

It is possible that power is a contributing factor to the reduced number of genes that are associated with neuropathology. In the analysis of the gene expression changes that are associated with neuropathology, only PSP cases (N=281) were used because control brains by definition did not have tau neuropathology at the level of these PSP donors. We also showed in the following figure that compared to the PSP vs CT DEG analysis (N=408), the power of a PSP case-only analysis is lower. Additionally, the expression differences (effect size) within PSP might also be smaller than that between PSP and control, making the analysis even less powered (both reduced numbers and lower effect size).

Response Figure 9

We also want to add that in running the power analysis, we discovered some minor typographical errors in **Figure 1A** relevant to the total number of genes that are associated with PSP diagnosis and the neuropathology phenotypes. We have made the relevant changes in the figure and the manuscript text

(highlighted in red), in **Results section: PSP brains have vast and replicable transcriptome perturbations in glia-enriched genes:**

Compared to controls, **2,528** genes were differentially expressed (DEGs) in PSP brains at an FDR-adjusted p-value of 0.05 (**Figure 1A**), suggesting extensive transcriptional dysregulations in PSP brains at the bulk tissue level, even in a brain region relatively spared from gross tau pathology. In terms of neuropathology, we detected the greatest number of associations with NFT (**134** genes), while **8** gene levels were associated with TauTh (**Figure 1B**).

Phenotype		Number of significant differentially expressed genes					
		FDR < 0.05		FDR < 0.10		Raw p value < 0.05	
Diagnosis		Higher in PSP	Lower in PSP	Higher in PSP	Lower in PSP	Higher in PSP	Lower in PSP
		PSP vs Ctrl	1,372	1,141	1,967	1,734	2,684
Neuropathology		Positive correlates	Negative correlates	Positive correlates	Negative correlates	Positive correlates	Negative correlates
	TA	0	0	0	0	937	934
	CB	0	0	0	0	349	361
	NFT	32	42	98	102	1,104	1,211
	TauTh	5	1	7	3	1,164	1,132
	Overall	0	0	2	0	770	850

Original figure 1A

Phenotype		Number of significant differentially expressed genes					
		FDR < 0.05		FDR < 0.10		Raw p value < 0.05	
Diagnosis		Higher in PSP	Lower in PSP	Higher in PSP	Lower in PSP	Higher in PSP	Lower in PSP
		PSP vs Ctrl	1,383	1,145	1,981	1,741	2,703
Neuropathology		Positive correlates	Negative correlates	Positive correlates	Negative correlates	Positive correlates	Negative correlates
	TA	0	0	0	0	941	938
	CB	0	0	0	0	351	361
	NFT	62	72	159	175	1,111	1,219
	TauTh	5	3	41	19	1,165	1,139
	Overall	0	0	2	0	773	854

Updated figure 1A

b. Are there any genes differentially expressed in some of the compression?

We understand from this inquiry that the reviewer is asking whether any of the 2,528 PSP vs control DEG are also significantly associated with one of the PSP neuropathology phenotypes. In the following upset plot, we show that there are 27 such genes. We noticed that there is one gene (*TRIP10*) whose expression is significantly associated with PSP diagnosis (logFC = 0.52, FDR = 0.009), NFT (beta = 0.23, FDR = 0.04), and

TauTh (beta = 0.31, FDR = 0.005) pathology. Additionally, the associations are concordant, as *TRIP10* expression is higher in PSP samples compared to controls and positively correlates with the severity of the pathology. We should note that both our interactive web application *PSP RNAseq Atlas* (https://rtools.mayo.edu/PSP_RNAseq_Atlas/) and the new **Supplementary Data 1** should enable the research community to query our findings for such overlapping associations.

Response Figure 10

c. Are these results driven from one of the two studies employed?

We understand that our reviewer is asking “are the transcriptional changes between PSP and control driven by one of the two studies”. It is unlikely that the direction of change (upregulation vs downregulation) is driven by one study. We defined significant DEG based on the meta-analyzed FDR-adjusted values. If a gene is perturbed in one study while having an opposite effect in the other, the meta-analyzed results will be less significant, thus this gene will not be considered as a DEG. In fact, there is a very strong correlation between the logFC in study 1 and study 2 for the significant DEGs (Spearman’s rho = 0.93). In terms of the significance of the DEG, study 1 has more samples than study 2, making it more powered than the other. Therefore, it is certainly possible that for some genes, the PSP vs control DEG values are more significant in one study than the other, leading to an overall significant DEG in the meta-analysis. However, our top 3 genes *DDR2*, *KANK2*, and *STOM* are not driven by one particular study as they all have comparable p values and consistent direction of effect in both studies (please see the following table, manuscript Figure 5A).

GeneName	study	PSP vs Control		
		logFC	pvalue	FDR
DDR2	study1	0.54	9.08e-04	2.31e-02
	study2	0.60	5.46e-03	7.12e-02
	meta	0.56	1.17e-05	1.03e-03
STOM	study1	0.43	1.02e-02	8.55e-02
	study2	0.51	1.49e-02	1.19e-01
	meta	0.46	3.64e-04	7.58e-03
KANK2	study1	0.52	1.04e-03	2.49e-02
	study2	0.42	9.37e-03	9.44e-02
	meta	0.47	2.52e-05	1.67e-03

- d. These results suggest that PSPS is associated with additional changes in the TCX not captured by tau neuropath variables. Are any of these changes observed in the TCX of other neuropath diseases, such as Alzheimer's? Are there any additional neuropath variable, or inferred variable (such as proportion of oligos, or other cell-type) that can be approximated using cell deconvolution, that can be explaining these differences, as suggested in lines 106-114? Additional analyses of sources of unwanted variation (e.g. SVA) might reveal additional interesting and important insights that can conciliate the differences in these analyses.

We thank the reviewer for the questions. Our group has shown that many transcriptomic changes between PSP and control brains in TCX were also detected in the analysis comparing the expression data of AD vs control brains¹¹. Some examples of such genes are *YAP1* and *CLU*, both upregulated in PSP and AD compared to control, respectively.

There are multiple factors that can explain the reduced number of genes that are significantly associated with neuropathology. As we discussed in R3-1a, power and the case-only nature of the neuropathology association can certainly contribute to detecting fewer significant genes. Besides these factors, we also want to highlight that neuropathology phenotypes are more specific than the overall case vs. control phenotypes, and the gene expression changes associated with the former might be subtle. For example, it is possible that not many genes are associated with the degree of tufted astrocyte (a continuous measure). In contrast, many more changes can be detected when comparing the gene expression between PSP and control (a binary measure).

We did not focus on the cell proportion in our manuscript because, as we discussed in response to reviewer 2 minor comment 1 (R-Minor-2-1), TCX is selected as it is relatively spared from PSP pathology. Nevertheless, we acknowledge that cell proportion changes can be a confounder in the analysis. To address the issue, we performed the PSP vs Control DEG analysis while adjusting for the cell proportions, estimated using the Digital Sorting Algorithm (DSA)²¹ and top 100 cell type marker genes from BRETIGEA²⁰. We showed in **Response Figure 11 Panel A** that there are no differences in cell proportion between PSP and control brains for microglia and endothelia but astrocyte and oligodendrocyte proportions do have a statistically significant difference in both and neurons in one of the two bulk RNAseq studies. Despite the proportion difference for these cell types, we showed in **Response Figure 11 Panel B** that there is still a high degree of correlation in PSP vs Control logFC between the simple model (reported in the manuscript) and the comprehensive model (adjusted for cell proportions as well). Lastly, we showed that the top three genes highlighted in our manuscript all have a consistent direction of change and remain statistically significant even after adjusting for the cell type **Response Figure 11 Panel C**. Taken together, we acknowledge the possibility of cell proportion changes in TCX of PSP brains, at least for some cell types, even though this brain

region is relatively spared of pathology. We, nevertheless, showed that the cell proportion changes do not significantly alter the PSP vs Control DEG.

We added text to the **Discussion** regarding potential reasons for the differences between numbers of DEGs for PSP vs. control and neuropathology phenotypes analyses:

Our collective findings demonstrate vast glial expression changes in PSP in a brain region relatively spared from gross neuropathological changes in this condition^{16,22}, suggesting these changes are less likely to be driven by neuropathology including neuronal loss or gliosis. We detected a smaller number of genes significantly associated with neuropathology than with PSP diagnosis. This is likely because the analysis was carried out only in PSP cases, which has lower power both owing to smaller sample size and smaller effect size. Furthermore, the neuropathology phenotypes represent specific cellular tau pathology in PSP, which may be associated with less transcriptional perturbations than the binary overall disease phenotype. Nevertheless, our cross-species multimodal systems biology approach prioritized three genes as potential therapeutic targets in PSP.

Response Figure 11

C

GeneName	Study	Simple			Comprehensive		
		logFC	pvalue	FDR	logFC	pvalue	FDR
DDR2	study 1	5.393e-01	9.082e-04	2.312e-02	2.077e-01	4.741e-05	2.930e-04
	study 2	5.991e-01	5.459e-03	7.123e-02	2.255e-01	2.236e-03	1.043e-02
	meta	5.611e-01	1.169e-05	1.032e-03	2.135e-01	2.184e-07	2.645e-06
KANK2	study 1	5.232e-01	1.035e-03	2.491e-02	3.407e-01	2.309e-07	3.420e-06
	study 2	4.157e-01	9.368e-03	9.435e-02	3.042e-01	1.044e-04	1.007e-03
	meta	4.695e-01	2.521e-05	1.668e-03	3.256e-01	2.854e-11	1.480e-09
STOM	study 1	4.313e-01	1.020e-02	8.548e-02	1.372e-01	8.083e-03	2.156e-02
	study 2	5.073e-01	1.491e-02	1.193e-01	1.238e-01	6.128e-02	1.312e-01
	meta	4.614e-01	3.643e-04	7.582e-03	1.321e-01	1.087e-03	3.596e-03

e. As currently stated, it is not clear for this reviewer what was tested in the section “top PSP vs. control brain expression changes defined as lowest 5% FDR and top 5% $|\logFC|$ ”.

We thank the reviewer for pointing out the lack of clarity in our wording. We are referring to a subset of bulk RNAseq PSP vs Control DEGs. We selected this subset of DEG based on two criteria: the gene is at the

bottom 5% FDR in terms of FDR value (lower FDR = more significant) and top 5% absolute value of logFC (higher |logFC| = more expression changes). We have now rephrased the sentence to better reflect our intention in **Results section: PSP brains have vast and replicable transcriptome perturbations in glia-enriched genes:**

We next focused on the top gene expression changes in PSP brains compared to controls. We defined top expression changes as a PSP vs control DEG with the bottom 5% FDR and top 5% |logFC| (**Figure 1B**). We observed biologically congruent gene expression associations with both diagnosis and neuropathology.

f. Data from the Cerebellum was also generated for some of these brain donors. How are the TCX changes compared to Cerebellum?

Indeed, for the donors from study 2, bulk RNAseq was also obtained from the cerebellum¹². Including the cerebellum data in this paper would make it lose its focus and make it too lengthy. Furthermore, we already evaluated TCX vs. cerebellum DEGs in PSP vs. control in another paper from our group¹¹. We showed that many gene expression changes between PSP and control brains are conserved between TCX and cerebellum. Further, we discovered that many DEGs are shared between PSP and another neurodegenerative disease Alzheimer's disease in both TCX and cerebellum, suggesting that these transcriptional changes are unlikely to be driven by gross pathology but may rather be common drivers of neurodegeneration.

R3-2: It would be important to communicate the readers the relational that guided the selection of the 18 PSP and 16 CT for single-cell experiments from the far larger number of brains with bulk. In addition, Is there any power analyses suggesting that single-cell from these would provide similar of more insights than larger cohorts.

We refer our reviewer to R1-2, where we now provide a modified Table S4, which depicts the demographics of the donors for the snRNAseq experiments and better delineates their metadata. We also discuss the validity of our snRNAseq results in R1-1, where we describe our approach in addressing sparsity, demonstrate high correlations of snRNAseq results from two different analytic approaches (MAST vs. pseudobulk) and show UMAPs of our snRNAseq data by different variables (now included in the manuscript as new **Figure S9**), which demonstrate no systematic enrichment of particular snRNAseq clusters by the different variables. Notwithstanding these strengths of our snRNAseq data, we acknowledge that the number of donors for this data is limited in comparison to that from the bulk brain transcriptome data from **two independent cohorts** (281 PSP and 127 control brains). This limitation is mainly due to the more arduous and far more costly nature of snRNAseq studies compared to bulk RNAseq. Despite this, to our knowledge, there is only one other study that profiled PSP brain transcriptome at the single-cell level² with 3 PSP and 3 controls. As such our snRNAseq data represents a valid, useful and necessary resource for the research community.

Finally, and as also described in R1-1, we utilize the brain snRNAseq data not as a discovery cohort per se but to “further replicate the glial gene expression changes from bulk brain RNAseq detected in our two independent studies” (Results section: **Single-nucleus RNAseq captures glial expression changes in PSP brains**). Indeed, our snRNAseq data “corroborates the cell-enrichment annotations for the PSP-associated bulk RNAseq modules M3, M4, and M6”. Further, snRNAseq data consistent with that in bulk transcriptome helped prioritize the high-confidence genes for cross-species validations (**Figure 4**).

R3-3: Sentence in lines 166-8 should be edited. It is not clear what contributed to <1% of the nuclei.

We appreciate the reviewer for pointing out this lack of clarity. We intend to explain that among the 8 clusters that had an over-representation of samples, 7 clusters were small and had low nuclei numbers. In other words, the number of nuclei in each of cluster is fewer than 1% of the total nuclei in the snRNAseq dataset. We have reworded the paragraph in **Results section: Single-nucleus RNAseq captures glial expression changes in PSP brains, as follows:**

Although a few samples were statistically over-represented in 8 clusters, 7 of these clusters were small, and each contributed to <1% of the total nuclei in the snRNAseq dataset, indicating homogeneity for most of the nuclear clusters with respect to sex, diagnosis, and samples.

R3-4: Regardless specific association to any of the 28 clusters, did the authors identify any of the changes in the cellular population structure identified using bulk RNA-seq in the single-cell data? I would suggest to perform additional analyses that can help determine this question, which I think is central to this work. Does pseudo-bulk analyses of single-cell data provides similar results to the bulk RNA-seq? Is it a power issue? More importantly, nowadays several authors isolate each of the cell-types from the data set (digital sorting) and recluster the data to identify specific changes. I think this approach can be highly informative for this data.

We appreciate the reviewers for their suggestions. We address the issue of cell proportions in R3-1d, where we did not identify any changes in PSP vs. control for microglia and endothelia proportions. Although there were statistically significant cell proportion changes for the other cell types in one or both of the two bulk RNAseq datasets, adjustment for cell type proportions did not influence PSP-associated DEGs either globally (**Response Figure 11 Panel B**) or for the top 3 highlighted genes (**Response Figure 11 Panel C**).

To address the current comment, we also calculated the proportion of the astrocyte, endothelia, microglia, neuronal, and oligodendrocyte nuclei per individual in our snRNAseq data (**Response Figure 12**).

Response Figure 12

Similar to the bulk deconvolution results we showed in R3-1d, there are no significant changes in microglia proportions. Although endothelial cells have a lower proportion in PSP brains, we should reiterate that the endothelial cells account for less than 1% of the total nuclei in the snRNAseq dataset. Therefore, this finding should be re-evaluated in future larger snRNAseq datasets from PSP and control brains. Finally, unlike the bulk RNAseq deconvolution, we did not find significant cell proportion differences in astrocytes, oligodendrocytes or neurons. This may either be because of the smaller number of donors in the snRNAseq data (i.e. false negative) or that these cells are indeed not very different proportionately, and deconvolution of bulk RNAseq is less accurate (i.e. true negative). Either way, adjusting for cell proportions does not make a difference in the bulk RNAseq DEG results as shown in detail in R3-1d.

We also performed pseudobulk analysis with our snRNAseq data and compared the PSP vs control samples. In R1-1, we show the comparison of pseudobulk vs. MAST analyses based snRNAseq DEGs and demonstrate their high correlations (**Response Figure 1**). In **Response Figure 13**, we depict the comparison of PSP vs. control DEG effect sizes (log fold change=logFC) between pseudobulk snRNAseq (n=34) and bulk RNAseq (n=401) DEGs. There is a highly significant correlation between these DEG effect sizes (Spearman's rho = 0.25), which is even stronger among the genes that are significantly associated with PSP in bulk RNAseq data (red dots, Spearman's rho = 0.48). These findings further demonstrate the replicability of the bulk RNAseq data with snRNAseq, which brings an additional level of validation and prioritization to the DEG findings, as detailed in our manuscript.

Response Figure 13

Following the reviewer's suggestion, we performed subcluster analysis of the astrocytic nuclei from our dataset. We re-analyzed them following the same strategy we used for the entire dataset, resulting in 3 astrocyte subclusters (**Response Figure 14, Panel A**). The three clusters had different expression marker genes (**Panel B**). For example, Ast_CL0 has high expression of genes involved in the synapse (*GRM3*, *NLGN4X*, *LRRRC4C*, *EGFR*, *EPHB1*). Ast_CL1 has a high expression of genes involved in cell cytoskeleton and extracellular matrix, such as *KAZN*, *VCAN*, and *L3MBTL4*. Ast_CL2 appears to be related to inflammation, as genes such as *CHI3L1*, *SERPINA3*, and *OSMR* are highly expressed^{23,24}. We also showed that our clusters are not confounded by the known variables after integration with Harmony²⁵ (**Panel C**), indicating that each cluster is likely to capture the true biological differences.

We compared the gene expression profile between PSP and control nuclei in each cluster (**Panel D**). Interestingly, there are many overlaps between the DEGs in Ast_CL0 and Ast_CL1 in both directions. This might indicate that those two clusters have similar transcriptional changes in PSP, though we note that Ast_CL2 is the smallest of the clusters where power to detect DEGs may be more limited. Lastly, we checked the DE statistics for the top three genes highlighted in our manuscript (**Panel E**). Both *DDR2* and *KANK2* are significantly upregulated in PSP in Ast_CL1 which is implicated in cell remodeling, consistent with their gene function. *KANK2* is also significantly higher in Ast_CL0. Although the FDR value for *STOM* did not reach statistical significance, it nevertheless has higher expression in PSP than in the control in all three astrocytic subclusters, suggesting that

STOM is perturbed universally in all astrocytes. We agree with our reviewer that investigating single nucleus subcluster analyses will be a very interesting area of research in PSP. We opted not to include these results in the manuscript, given constraints in both space and scope. Nevertheless, our data, scripts and detailed results that we make available to the research community through our new interactive application tool **PSP RNAseq Atlas** (https://rtools.mayo.edu/PSP_RNAseq_Atlas/), our 5 new **Supplementary Data** (see also R2-11) and the AD Knowledge Portal (<https://adknowledgeportal.synapse.org>) will enable the research community to pursue further investigations and analyses of our findings and data.

Response Figure 14

R3-5: I would suggest the authors to try to refine/extend the gene networks identified using RNA-seq using the single-cell data. In addition, gene regulatory network analyses (beyond co-expression) can be highly informative for these key modules identified in bulk and followed up in single-cell,

We have assessed the preservation of the bulk-RNAseq-derived WGCNA network in the snRNAseq data in R1-5. In summary, there is a high degree of consistency between the enriched cell marker genes in the modules and the snRNAseq expression data from the corresponding cell type for bulk RNAseq modules M3 (oligodendrocytes) and M4 (astrocytes). Module M6, which has enrichment of multiple cell type marker genes, does not have high preservation (preservation score >10) in any cell type. These findings are aligned with **Figure 3D** in our manuscript.

We agree with Reviewer 3 that gene regulatory networks can be informative. Using bulk transcriptome data from PSP brains, our group has previously published transcriptional regulatory network analysis results¹⁷. We found multiple transcription factors (TFs) that are highly connected and central in the modules that are associated with PSP. Some of the TFs also have nominally significant cis-eQTL, suggesting their potential co-regulatory effects of the module genes. Again, it will be interesting to pursue this type of analyses in future studies, which we intend to do. Other researchers will also be able to do this using the data, metadata and scripts that we provide as outlined in R3-4.

R3-6: Did the authors consider using additional data (e.g. Glasauer et al <https://doi.org/10.1016/j.stemcr.2022.07.011>) to replicate their findings in oligodendrocytes not captured in the rTg4510 model?

We appreciate the reviewer for sharing the excellent paper and dataset by Glasauer et al²⁶. We were reassured to find that *DDR2*, one of our top genes, is upregulated (logFC: 0.0701, FDR: 2.28E-02) in homozygous *MAPT* mutants compared to the isogenic control organoid astrocytes. We now include this paper in our references and report this finding in our discussion as well. We also fully intend to expand our findings to other model systems including iPSC-derived models and additional mouse models that will give us the opportunity to investigate the results of other modules, including those of oligodendrocytes.

... *DDR2* has also been implicated in neurodegeneration²⁷. Consistent with our findings, it was shown that *DDR2* levels are upregulated in the astrocytes from *MAPT* mutant organoids²⁶. *DDR2* knockdown *in vitro*....

R3-7: Additional data, statistical test significance should be provided for drosophila RNAi experiments. Did the authors evaluate and capture any additional phenotypic (expected/unexpected) changes beyond eye morphology?

We thank the reviewer for the opportunity to expand upon our *Drosophila* experiments. In response to Reviewer 3, we have now repeated the RNAi experiments for the top 3 genes, have two independent examiners score them in a blinded fashion and provide statistical significance for the findings. We used the same GMR-Gal4;UAS-Tau flies as described in the method section of our manuscript. We now added the following text to the **Methods Section: Validation with *Drosophila* Tau Model Experiments:**

To assess the impact of the three top tau-toxicity suppressor genes, we aged the progenies for 5 days at 25°C before pictures of the fly left eyes were taken. The severity of the tau-induced eye degeneration

was assessed blindly based on the following categories: loss of bristle (0-1), size (0-1), color (0-2), the presence of necrotic patterns (0-2), the collapse of the eye (0-2), and the loss of ommatidia (0-2), where a higher score indicates a more severe phenotype. Scores were provided by two independent evaluators separately, and the average scores were used for analysis.

We also added new text and figures summarizing the results of this additional validation in the *Drosophila* tau model in the **Results section: Validation of top tau toxicity suppressing PSP glial perturbed genes**. To better incorporate the new results into the manuscript, we edited text in both this aforementioned section, as well as in the preceding **Results section: Cross-species prioritization and screening of glial gene expression changes in PSP**. To conserve space in this rebuttal, we only show the text pertinent to the new *Drosophila* tau results, and refer our reviewer to the changes-tracked version of our Main manuscript for a full review of all the changes, which were done to address their comments. Our new *Drosophila* tau results are strongly supportive of those from the original submission, as follows:

To further validate and confirm these *Drosophila* screening results, we repeated the RNAi experiments for these three genes, and quantified the severity of tau-mediated toxicity using two blinded independent evaluators (Figures 5C-D). As expected, the expression of tau under the GMR-Gal4 driver led to significant *Drosophila* eye degeneration compared to wild type (WT) flies (two-sided Wilcoxon rank sum test, $p=6.16E-5$). Importantly, expression of RNAi against *DDR2*, *KANK2*, or *STOM* in tau *Drosophila* significantly reduced the tau-mediated cell toxicity in fly eyes.

Below figures correspond to the new Figures 5C and D

We did not assess other phenotypes in the GMR-Gal4 fly because the eye phenotype is the most robust outcome. Additional experiments with glial-specific drivers such as *repo*-Gal4⁹ can be planned, but such model systems will require additional time for optimization. We acknowledge such additional models in our **Discussion** as follows and also as mentioned in R1-6:

Additionally, unlike the *repo*-Gal4 driver⁹, the *Drosophila* GMR-Gal4 driver is not glia specific as it drives the expression of tau in all cell types.

R3-8: Are any of the compounds associated with the networks identified that can also ameliorate neurodegeneration in the drosophila models?

We thank the reviewer for the suggestions. Indeed by querying the Drug Gene Interaction database²⁸ we identified multiple small molecule inhibitors for *DDR2* expression, including Nilotinib and Dasatinib which had the highest scores among the inhibitors (**Supplementary Data 5**). To address our reviewer's comment, using the GMR-Gal4;UAS-Tau system, we assessed the effect of Dasatinib treatment on the eye phenotype. Briefly, crosses were set up with regular fly food supplemented with either Dasatinib (final concentrations of 1 uM or 10uM) or DMSO (0.5%). Progenies with the correct genotypes were transferred to fresh drug- or DMSO-containing food after eclosing, and aged for 5 days, changing to fresh drug- or DMSO-containing food every 2 days. The fly eye degeneration was assessed with blinded evaluators, as we described in R3-7. Compared to DMSO treatment, the treatment of Dasatinib led to a statistically significant, dosage-dependent reduction in tau-mediated eye pathology in the flies that express tau. The results suggested that *DDR2* is a robust and promising target in mediating PSP and other tauopathies.

Response Figure 15

We present these very encouraging results here for the reviewer but consider them to be preliminary, as they should be repeated and also compared to the Nilotinib treatment. We attempted Nilotinib treatment as well. However, as suggested by previous literature^{27,29}, a high concentration of Nilotinib is needed for such treatment. Combined with the fact that Nilotinib has low solubility, we had to increase the DMSO concentration from 0.5% to 1%, which showed toxicity against the fly pupas. For these reasons, we are in the process of further optimizing these experiments and fully intend to follow up on this in a future publication.

R3-9: Are the cells in clusters (CL23, CL24, CL26) undifferentiated/precursor cells? Additional investigation should be done to determine if these are artifacts and discard these in that case.

We appreciate the reviewer for asking the question. We want to clarify that clusters CL23, 24, and 26 are all very small clusters in terms of size, each accounting for less than 1% total nuclei. Due to their low nuclei count, assigning reliable cell types and inferring their biology is challenging. Further, those clusters do not show glial signatures gene expression (**Figure 3B, S11**) and are not used in our target nomination scheme (**Figure 4A**). We wish to clarify that for completeness and transparency, we depict the results in these clusters, but they do not have any impact on the findings in our manuscript.

R3-10: Please provide the QC parameters and add some plots/information for the snRNAseq QC.

We thank the reviewer for the suggestions, which we address in our revised manuscript. We provide the QC scripts for our snRNAseq analysis. The parameters used in the QC have also been described in R2-12. Additionally, we provided the distributions of the UMI (A), gene count (B), and percent mitochondrial genes (C) in each nuclei. At a gene level, we also show the number of nuclei expressing the gene (UMI > 5, D). Lastly, we provide figures to show that no sample has mismatched sex (E). We now include a new **Figure S8** in our revised manuscript to address this comment. This figure is also shown below.

Figure S8. Quality-Control metrics for the snRNAseq analysis.

We also updated the methods section, please see the updated text in R2-12.

R3-11: I suggest the authors to provide additional descriptions of the large variety of data and analyses shared on their website. As currently stated, readers might miss many of these great resources.

We appreciate the reviewer's positive comments. We modified the **Results section: Validation of top tau toxicity suppressing PSP glial perturbed genes** with the following text regarding our interactive web tool.

We built an interactive web application tool PSP RNAseq Atlas ([https://rtools.mayo.edu/PSP RNAseq Atlas/](https://rtools.mayo.edu/PSP_RNAseq_Atlas/)), which houses human bulk and snRNASeq transcriptome, mouse and Drosophila results. Our application can quickly report the key statistics for queried genes and is built to facilitate the dissemination of our results to the broader research community.

We also modified the **Discussion** section to better reflect the utility of our web application:

Our findings and data we make available in the interactive web application PSP RNAseq Atlas ([https://rtools.mayo.edu/PSP RNAseq Atlas/](https://rtools.mayo.edu/PSP_RNAseq_Atlas/)) highlight the complex pathophysiology of PSP. Our web application, data and results are also expected to serve as a resource for the research community to apply their own paradigms for therapeutic target or biomarker prioritizations in PSP or other neurodegenerative diseases.

Furthermore, we now include 5 new **Supplementary Data** for both our bulk and snRNAseq data. We refer our reviewer to sections R1-3 and R2-11 regarding these resources.

R3-12: In this manuscript author mentioned the three potential targets (DDR2, STOM and KANK2) but they have focused on DDR2 inhibitor. It would be more informative if they mentioned the list of drugs/inhibitor that can be used against other targets.

We agree with the reviewer though note that the other two genes did not have a list of inhibitors in the Drug Gene Interaction database (DGIdb). We should also note that we included all potential drugs against our genes of interest in our web application, **PSP RNAseq Atlas**. To make it even easier for the reader to locate this information, we also now provide the list of DGIdb drugs against our target genes in a new **Supplementary Data 5**. We edited the **Discussion** as follows:

Our findings suggest that nilotinib or other small molecules inhibiting DDR2 may also be viable therapeutic candidates in PSP. We provide a list of small molecule inhibitors retrieved from the Drug Gene Interaction database (DGIdb) in **Supplementary Data 5**.

References

- 1 Mathys, H. *et al.* Single-cell transcriptomic analysis of Alzheimer's disease. *Nature* **570**, 332-337 (2019). <https://doi.org:10.1038/s41586-019-1195-2>
- 2 Sharma, A. *et al.* Single-cell atlas of progressive supranuclear palsy reveals a distinct hybrid glial cell population. *bioRxiv*, 2021.2004.2011.439393 (2021). <https://doi.org:10.1101/2021.04.11.439393>
- 3 Finak, G. *et al.* MAST: a flexible statistical framework for assessing transcriptional changes and characterizing heterogeneity in single-cell RNA sequencing data. *Genome Biol* **16**, 278 (2015). <https://doi.org:10.1186/s13059-015-0844-5>
- 4 Squair, J. W. *et al.* Confronting false discoveries in single-cell differential expression. *Nature Communications* **12**, 5692 (2021). <https://doi.org:10.1038/s41467-021-25960-2>
- 5 Patel, T. *et al.* Transcriptional landscape of human microglia implicates age, sex, and APOE-related immunometabolic pathway perturbations. *Aging Cell*, e13606 (2022). <https://doi.org:10.1111/ace1.13606>
- 6 Habib, N. *et al.* Disease-associated astrocytes in Alzheimer's disease and aging. *Nat Neurosci* **23**, 701-+ (2020). <https://doi.org:10.1038/s41593-020-0624-8>
- 7 Sadick, J. S. *et al.* Astrocytes and oligodendrocytes undergo subtype-specific transcriptional changes in Alzheimer's disease. *Neuron* (2022). <https://doi.org:10.1016/j.neuron.2022.03.008>
- 8 Wittmann, C. W. *et al.* Tauopathy in Drosophila: neurodegeneration without neurofibrillary tangles. *Science* **293**, 711-714 (2001). <https://doi.org:10.1126/science.1062382>
- 9 Sepp, K. J., Schulte, J. & Auld, V. J. Peripheral glia direct axon guidance across the CNS/PNS transition zone. *Dev Biol* **238**, 47-63 (2001). <https://doi.org:10.1006/dbio.2001.0411>
- 10 Lisi, V., Luna, G., Apostolaki, A., Giroux, M. & Kosik, K. S. Cell Population Effects in a Mouse Tauopathy Model Identified by Single Cell Sequencing. *bioRxiv*, 771501 (2019). <https://doi.org:10.1101/771501>
- 11 Wang, X. *et al.* Alzheimer's disease and progressive supranuclear palsy share similar transcriptomic changes in distinct brain regions. *J Clin Invest* **132** (2022). <https://doi.org:10.1172/JCI149904>
- 12 Allen, M. *et al.* Human whole genome genotype and transcriptome data for Alzheimer's and other neurodegenerative diseases. *Sci Data* **3** (2016). <https://doi.org:ARTN 160089>
10.1038/sdata.2016.89
- 13 Leek, J. T., Johnson, W. E., Parker, H. S., Jaffe, A. E. & Storey, J. D. The sva package for removing batch effects and other unwanted variation in high-throughput experiments. *Bioinformatics* **28**, 882-883 (2012). <https://doi.org:10.1093/bioinformatics/bts034>
- 14 Johnson, W. E., Li, C. & Rabinovic, A. Adjusting batch effects in microarray expression data using empirical Bayes methods. *Biostatistics* **8**, 118-127 (2007). <https://doi.org:10.1093/biostatistics/kxj037>
- 15 Staedtler, F. *et al.* Robust and tissue-independent gender-specific transcript biomarkers. *Biomarkers* **18**, 436-445 (2013). <https://doi.org:10.3109/1354750X.2013.811538>
- 16 Hauw, J. J. *et al.* Preliminary NINDS neuropathologic criteria for Steele-Richardson-Olszewski syndrome (progressive supranuclear palsy). *Neurology* **44**, 2015-2019 (1994). <https://doi.org:10.1212/wnl.44.11.2015>
- 17 Allen, M. *et al.* Divergent brain gene expression patterns associate with distinct cell-specific tau neuropathology traits in progressive supranuclear palsy. *Acta Neuropathol* **136**, 709-727 (2018). <https://doi.org:10.1007/s00401-018-1900-5>
- 18 Allen, M. *et al.* Gene expression, methylation and neuropathology correlations at progressive supranuclear palsy risk loci. *Acta Neuropathol* **132**, 197-211 (2016). <https://doi.org:10.1007/s00401-016-1576-7>
- 19 Thrupp, N. *et al.* Single-Nucleus RNA-Seq Is Not Suitable for Detection of Microglial Activation Genes in Humans. *Cell Rep* **32**, 108189 (2020). <https://doi.org:10.1016/j.celrep.2020.108189>

- 20 McKenzie, A. T. *et al.* Brain Cell Type Specific Gene Expression and Co-expression Network Architectures. *Sci Rep-Uk* **8** (2018). <https://doi.org:ARTN> 8868
10.1038/s41598-018-27293-5
- 21 Zhong, Y., Wan, Y. W., Pang, K. F., Chow, L. M. L. & Liu, Z. D. Digital sorting of complex tissues for cell type-specific gene expression profiles. *Bmc Bioinformatics* **14** (2013). <https://doi.org:Artn> 89
10.1186/1471-2105-14-89
- 22 Roemer, S. F. *et al.* Rainwater Charitable Foundation criteria for the neuropathologic diagnosis of progressive supranuclear palsy. *Acta Neuropathol* **144**, 603-614 (2022). <https://doi.org:10.1007/s00401-022-02479-4>
- 23 Liddelw, S. A. *et al.* Neurotoxic reactive astrocytes are induced by activated microglia. *Nature* **541**, 481-487 (2017). <https://doi.org:10.1038/nature21029>
- 24 Barbar, L. *et al.* CD49f Is a Novel Marker of Functional and Reactive Human iPSC-Derived Astrocytes. *Neuron* **107**, 436-453 e412 (2020). <https://doi.org:10.1016/j.neuron.2020.05.014>
- 25 Korsunsky, I. *et al.* Fast, sensitive and accurate integration of single-cell data with Harmony. *Nat Methods* **16**, 1289-1296 (2019). <https://doi.org:10.1038/s41592-019-0619-0>
- 26 Glasauer, S. M. K. *et al.* Human tau mutations in cerebral organoids induce a progressive dyshomeostasis of cholesterol. *Stem Cell Reports* **17**, 2127-2140 (2022). <https://doi.org:10.1016/j.stemcr.2022.07.011>
- 27 Hebron, M. *et al.* Discoidin domain receptor inhibition reduces neuropathology and attenuates inflammation in neurodegeneration models. *J Neuroimmunol* **311**, 1-9 (2017). <https://doi.org:10.1016/j.jneuroim.2017.07.009>
- 28 Freshour, S. L. *et al.* Integration of the Drug-Gene Interaction Database (DGIdb 4.0) with open crowdsource efforts. *Nucleic Acids Res* **49**, D1144-D1151 (2021). <https://doi.org:10.1093/nar/gkaa1084>
- 29 Sterne, G. R., Kim, J. H. & Ye, B. Dysregulated Dscam levels act through Abelson tyrosine kinase to enlarge presynaptic arbors. *Elife* **4** (2015). <https://doi.org:10.7554/eLife.05196>

REVIEWER COMMENTS

Reviewer #1 (Remarks to the Author):

The review of the article on the discovery of high-confidence expression changes in Progressive supranuclear palsy (PSP) has been comprehensive and rigorous. The study combined bulk tissue and single nucleus RNAseq data from PSP and control brains with transcriptome data from a mouse tauopathy and experimental validations in *Drosophila* tau models for systematic discovery of expression changes in PSP with therapeutic potential. The findings reveal robust glial transcriptome changes in PSP and provide a cross-species systems biology approach, as well as a novel tool for therapeutic target discoveries in PSP with potential application in other neurodegenerative diseases. Based on the thorough analysis and the promising results, I recommend that the article be accepted for publication.

The authors should upload the codes used for analysis separated by each figure on GitHub.

Reviewer #2 (Remarks to the Author):

In the revised version, the authors added several analyses but still need to catch up on a few critical aspects of data quality and limited transcriptomic analysis. In this regard, this reviewer proposes these revisions:

- Sample level QC violin plots, showing UMI, number of genes, and percent mt-reads. Also same QC post-filtering for each cluster

- It is clear from the point-to-point responses that the snRNA-Seq data was only used in the function of the bulk-RNAseq results. This is a missed opportunity to properly analyze the snRNA-Seq data from PSP vs. controls, which is very important for the field. It is unclear why the revised version of the manuscript did not include the analysis results performed in response to reviewer #1 about genes differentially expressed between different groups, what is shared between them, and what is common. This manuscript is a resource of single-cell transcriptomics, but that data currently needs to be included.

- The snRNA-Seq expression analysis is biased toward a specific set of genes from the modules M3, M4, and M6. The authors should undertake unbiased comparison analyses with proper controls.

- It is also clear that some of the key findings of their bulk-RnaSeq data were already published previously, which raises the question of novelty. Additional analysis could be gene-regulatory network analysis using SCENIC, for instance.

- is there any genetic information on the mutation status of the PSP patients? Since both mouse and fly models are based on the transgenic expression of MAPT mutants, the conclusions must be in the context of tau-induced neurodegeneration

Reviewer #3 (Remarks to the Author):

I appreciate the efforts of the authors to address previous revisions. Many of my comments, suggestions, and critiques have been addressed in this new version of the manuscript. However, there are still a few comments and suggestions pending to be addressed:

- 1) I recognize the authors have performed quite an effort to try to address the large number of hits using disease status compared to NFT. Still, there are some key experiments that remain to be done to give us a better understanding of these huge difference. By performing CT vs PSP DE among the subset of donors with neuropathology (regardless NFT-positive or -negative), the authors should be able to obtain a straightforward comparison of #DE, which can help to conciliate the differences between neuropath and diagnosis.
- 2) Response Fig 10. suggests that results from transcriptomics analyses differ greatly for diagnosis compared to neuropathology. It should be analyzed and discussed in detail.
- 3) The authors have proved that there are additional changes in the TCX not captured by tau neuropathology and indeed identified significant changes in the cell proportions. These are key results from their work and should be clearly communicated. A detailed description of the comprehensive model should be provided and used as the primary model. Furthermore, by adding these as covariates into their models, they are finding stronger associations. The authors might want to communicate any additional results without corrections in the supplementary section.
- 4) Cerebellum analyses are critical for PSP, and results from TCX should be analyzed in the light of unique and common hits.
- 5) The authors have access to a very large pool of PSP brains, and in previous studies, they have sequenced a huge number of brains. Plenty of reasons, including the high cost of single-cell data generation, restrict the generation of single-cell data for all of these brains.
 - a. It would be really informative to indicate why/how the brains were selected for single-cell studies
 - b. Discuss any possible limitation or bias of the selection of brains can impact in the results presented.
- 6) Web browser: I anticipate the server will be highly used, and at the present time I found it somehow slow. This brought me concerns about its performance at the time the manuscript is published. I suggest deploying it in a more performant server / cloud to scale up.
- 7) Does any of the astrocyte subclusters show the expression of genes of reactive astrocytes? These studies are also key to supporting authors' findings and should be comprehensively done and reported.
- 8) Raw p-value is often referred as nominal p-value. I would suggest using the most accepted nomenclature.

Cross Species Systems Biology Discovers Glial *DDR2*, *STOM*, and *KANK2* as Therapeutic Targets in Progressive Supranuclear Palsy

NCOMMS-22-50578A

Response to reviewer comments

R1-1	2
R2-1	3
R2-2	5
R2-3	6
R2-4	7
R2-5	8
R3-1	9
R3-2	10
R3-3	10
R3-4	12
R3-5a	14
R3-5b	14
R3-6	15
R3-7	15
R3-8	18

Reviewer 1:

The first reviewer summarized the manuscript as follow: “The review of the article on the discovery of high-confidence expression changes in Progressive supranuclear palsy (PSP) has been comprehensive and rigorous. The study combined bulk tissue and single nucleus RNAseq data from PSP and control brains with transcriptome data from a mouse tauopathy and experimental validations in Drosophila tau models for systematic discovery of expression changes in PSP with therapeutic potential. The findings reveal robust glial transcriptome changes in PSP and provide a cross-species systems biology approach, as well as a novel tool for therapeutic target discoveries in PSP with potential application in other neurodegenerative diseases. Based on the thorough analysis and the promising results, I recommend that the article be accepted for publication.”

We thank Reviewer 1 for their very positive comments and their recommendation of acceptance of our manuscript. Their one comment and our response are below:

R1-1: The authors should upload the codes used for analysis separated by each figure on GitHub.

We thank the reviewer for this comment. All analyses and visualization codes will be hosted on synapse.org and made broadly available as we stated in the **Code availability** section:

The analysis and visualization codes are available via the AD Knowledge Portal. The source code powering the web application is also hosted on the AD Knowledge Portal.

Reviewer 2:

The second reviewer commented: “In the revised version, the authors added several analyses but still need to catch up on a few critical aspects of data quality and limited transcriptomic analysis”.

We thank Reviewer 2 for the summary and provide a point-by-point response to each of their comments as follows:

R2-1: Sample level QC violin plots, showing UMI, number of genes, and percent mt-reads. Also same QC post-filtering for each cluster.

We thank the reviewer for the suggestion. To better present and explain the QC procedures. We modified **Figure S9** and the **Methods section: Human Brain Single Nucleus RNAseq (snRNAseq) Validations** to report the relevant QC metrics.

We obtained an average of 924 nuclei (standard deviation: 469, N = 36) per sample and performed quality control for each individual (**Figure S9**). First, nuclei with an extreme number of mapped UMIs (lower bound 1,000, upper bound 98th percentile, or $\geq 85,906.5$, Figures S9A-B) or detected genes (lower bound 500, upper bound 98th percentile $\geq 9,648.8$, Figures S9C-D) were removed. Nuclei with more than 10% mitochondrial genes were also excluded (Figure S9E-F). At a sample level, PSP sample 12117 was removed from the analysis because it has an abnormal distribution in terms of the number of mapped UMIs and the number of detected genes (Figure S9G). PSP sample 12051 was removed due to the low number of nuclei (N = 117, Figure S9H). Genes that are only expressed (defined as having a count of greater than 0) in less than 6 nuclei are excluded (Figure S9I). The recorded sex for each individual was compared against that of the median expressions of 4 chromosome Y genes *RPS4Y1*, *EIF1AY*, *DDX3Y*, and *KDM5D* to identify any potential mislabeled samples (Figure S9J).

Legend for New Supplementary Figure 9. Quality-Control metrics for the snRNAseq analysis. **A-B:** Distribution of the number of mapped UMIs from all nuclei and in each sample. **C-D:** Distribution of the number of detected genes from all nuclei and in each sample. **E-F:** Distribution of the percent of mitochondrial genes from all nuclei and in each sample. **G:** Distribution of the number of mapped UMI vs the number of detected genes in each sample. **H:** Number of nuclei in each sample. **I:** Distribution of the number of nuclei that are expressing each gene. **J:** Expression of chromosome Y genes *RPS4Y1*, *EIF1AY*, *DDX3Y*, and *KDM5D*.

Additionally, we generated and present the post-QC statistics for each cluster following the reviewer's suggestions.

Response Figure 1

R2-2: It is clear from the point-to-point responses that the snRNA-Seq data was only used in the function of the bulk-RNAseq results. This is a missed opportunity to properly analyze the snRNA-Seq data from PSP vs. controls, which is very important for the field. It is unclear why the revised version of the manuscript did not include the analysis results performed in response to reviewer #1 about genes differentially expressed between different groups, what is shared between them, and what is common. This manuscript is a resource of single-cell transcriptomics, but that data currently needs to be included.

We appreciate the reviewer for the suggestions. We agree that our PSP transcriptomics datasets are great resources for the research community, where many other research questions can be addressed. We had already provided the MAST differential expression (DE) results from the snRNAseq in the manuscript in **Supplementary Data 4**, in addition to our new interactive application tool **PSP RNAseq Atlas** (https://rtools.mayo.edu/PSP_RNAseq_Atlas/), which is built to facilitate the dissemination of our results to the broader research community. To make the manuscript an even better resource for the research community, we addressed the reviewer's comment by also adding the pseudobulk DE results to the existing MAST DE results in **Supplementary Data 4** as well as to our **PSP RNAseq Atlas**.

In the **Methods section: Human Brain Single Nucleus RNAseq (snRNAseq) Validations**, we describe how the pseudobulk DE analysis was carried out:

Pseudobulk DEG comparing the expression between PSP and control participants was performed for each cell type separately (Supplementary Data 4, Table S9). Briefly, the pseudo-count matrix was obtained by adding the read count from all the nuclei of the same individual. The pseudo count between PSP and control individuals was compared using a negative binomial generalized linear model implemented in R package edgeR, with adjustment of sex and age.

We now also include the response to reviewer 1 as **Supplementary Table 8** and in the **Results section: Validation of top tau toxicity suppressing PSP glial perturbed genes**, we added:

The up-regulation in bulk tissue is also consistent across both studies, suggesting the results are not driven by a particular cohort. In snRNAseq (**Figure 5B**), all three genes are significantly upregulated in astrocytes. Pseudobulk DEG analysis using the snRNAseq datasets also supported the up-regulation of the three genes in astrocytes (Table S8). Interestingly, the subclustering analysis (Figure S17) indicated *DDR2* and *KANK2* are up-regulated in specific populations of astrocytes, whereas the up-regulation of *STOM* is universal to all astrocytes. Our transcriptome data demonstrate that *STOM*, *KANK2*, and *DDR2* are up-regulated in glial cells in PSP.

New Supplementary Table 8: PSP vs Control DE results using MAST in each cluster and pseudobulk in each cell type for the top nominated genes.

Gene	Cell	Cluster	MAST			Pseudobulk		
			logFC	pvalue	FDR	logFC	pvalue	FDR
DDR2	ast	CL2	0.139	6e-06	6e-05	0.361	0.089	0.498
	mic	CL3	NA	NA	NA	-0.070	0.919	1.000
	oli	CL0	NA	NA	NA	0.111	0.763	0.936
	opc	CL7	NA	NA	NA	-0.136	0.749	0.970
	per	CL12	0.018	0.774	0.968	-0.077	0.743	0.969
		CL17	-0.077	0.409	1.000			
KANK2	ast	CL2	0.047	1e-06	1e-05	0.726	0.013	0.236
	mic	CL3	NA	NA	NA	0.023	0.962	1.000
	oli	CL0	NA	NA	NA	1.159	0.007	0.160
	opc	CL7	NA	NA	NA	0.371	0.497	0.912
	per	CL12	0.033	0.855	0.984	-0.241	0.333	0.864
		CL17	0.036	0.090	0.886			
STOM	ast	CL2	0.087	1e-04	7e-04	0.435	0.010	0.208
	mic	CL3	0.084	0.081	0.201	0.509	0.018	0.497
	oli	CL0	NA	NA	NA	1.221	0.003	0.107
	opc	CL7	NA	NA	NA	0.194	0.691	0.963
	per	CL12	0.081	0.017	0.302	0.538	0.033	0.634
		CL17	0.028	0.757	1.000			

R2-3: The snRNA-Seq expression analysis is biased toward a specific set of genes from the modules M3, M4, and M6. The authors should undertake unbiased comparison analyses with proper controls.

We thank the reviewer for the insightful comment. In our manuscript, we purposefully chose to use the snRNAseq dataset to validate and annotate the WGCNA results for the nomination of therapeutic targets. Therefore, the analysis has an emphasis on the modules M3, M4, and M6 genes. We agree that an unbiased analysis of the snRNAseq dataset is valuable and informative. However, such analysis is beyond the scope of the current manuscript. Nevertheless, we provided both the MAST DEG and pseudobulk DEG results in **Supplementary Data 4** and in the **PSP RNAseq Atlas**. In **Supplementary Data 4**, we include genes that are significantly differentially expressed in either MAST or pseudobulk or both analyses. Our manuscript, its results, detailed tables and our new interactive application tool **PSP RNAseq Atlas** (https://rtools.mayo.edu/PSP_RNAseq_Atlas/) will be highly useful resources to the research community beyond the novel discoveries that we provide in this work.

R2-4: It is also clear that some of the key findings of their bulk-RnaSeq data were already published previously, which raises the question of novelty. Additional analysis could be gene-regulatory network analysis using SCENIC, for instance.

We appreciate the reviewer for this comment which helps us clarify how our current manuscript is clearly novel and distinct from our previously published work. Although bulk RNAseq from study 2 was previously published, these data were used in a limited fashion either to replicate or validate findings from other datasets of microarray-based expression data (Allen et al., 2017, *Alzheimer's and Dementia*; Allen et al., 2018, *Acta Neuropathologica*) or in comparative differential gene expression analyses with Alzheimer's Disease datasets (Wang et al., 2022, *The Journal of Clinical Investigation*). We already mention these in the **Introduction** and discuss gaps in the literature that our current manuscript fills with novel findings. To further clarify these gaps and differences, we edited the **Introduction**, as follows:

While these findings nominate brain perturbations in gene expression and transcriptional networks as potential culprits in PSP pathophysiology, they are primarily based solely on microarray expression measures of bulk brain tissue from human cohorts or do not systematically investigate PSP brain RNAseq changes in independent cohorts. Additionally, Complementary data from single cell/single nucleus RNAseq (sc/snRNAseq) is necessary to assess cell-type-specific brain gene expression changes in PSP. Further, although investigating these transcriptome changes in model systems across different species can increase confidence in the findings and enable experimental validations, such studies are lacking in neurodegenerative diseases with few exceptions^{9,10}. Consequently, the exact disease mechanism(s) and key molecular players that underlie PSP pathogenesis remain unclear, creating a barrier to nominating effective targets for disease-modifying treatment.

Our current study is clearly unique in its design, additional datasets and key findings. First, in addition to study 2 bulk RNAseq cohort (82 PSP, 69 controls), we include a new, independent study 1 bulk RNAseq cohort (199 PSP, 58 controls) that increases the total bulk RNAseq sample size by 2.7x (**Supplementary Table S1**). To our knowledge, this is the largest PSP brain transcriptome study to date. Second, using this bulk RNAseq data, we find thousands of perturbed transcripts, define co-expression modules with robust replication across the two bulk RNAseq datasets and annotate these modules for their enriched biological terms and cell types. That we did replicate some of the prior findings from our group and others speaks to the rigor and reproducibility of our work. This does not take away from the novelty of the present study, as the key findings from the present work have not previously been published. Third, we include new snRNAseq data in this work, which are utilized in an innovative manner to validate the findings from bulk RNAseq. We also provide detailed gene expression perturbation information at bulk, single-cell, and pseudo-bulk levels (**Supplementary Data 1, Supplementary Data 4, online web application**), which to our knowledge, has not been reported before. Fourth, we perform additional and systematic validations using a mouse model of tauopathy. Fifth, using RNAi screening in a tau

Drosophila model, we provide functional validations for the top tau-mediated toxicity suppressing PSP genes and nominate *DDR2*, *KANK2*, and *STOM* as novel therapeutic targets for PSP. Sixth, we generated a new interactive application tool **PSP RNAseq Atlas** (https://rtools.mayo.edu/PSP_RNAseq_Atlas/) as a facile way to broadly share our data and findings. Seventh, we provide a novel, cross species systems biology approach to discover, prioritize and validate potential therapeutic targets for complex diseases, including but not limited to PSP.

In summary, our work is novel, robust and replicable. We appreciate the suggestion to include additional analyses such as gene regulatory analyses. We agree that many other analyses can be applied to these datasets to infer additional new insights about PSP and neurodegenerative diseases. These are beyond the scope of our existing work but are expected to be conducted by our group and others using these datasets that we make broadly available.

R2-5: Is there any genetic information on the mutation status of the PSP patients? Since both mouse and fly models are based on the transgenic expression of *MAPT* mutants, the conclusions must be in the context of tau-induced neurodegeneration.

We thank the reviewer for this comment. Among the PSP donors in this study, there is only one patient harboring the A152T risk variant in the *MAPT* gene. This variant is associated with a higher risk for tauopathies, rather than being a fully penetrant pathogenic *MAPT* mutation^{1,2}. We added the following text to **Methods: Sample Information** to clarify this issue and address this comment:

Amongst brain donors with PSP, there was only one patient with a *MAPT* mutation, namely the A152T variant which is considered a rare risk factor for tauopathies and not a fully penetrant pathogenic mutation^{42,43}.

Reviewer 3:

The third reviewer commented: “I appreciate the efforts of the authors to address previous revisions. Many of my comments, suggestions, and critiques have been addressed in this new version of the manuscript. However, there are still a few comments and suggestions pending to be addressed”.

We thank Reviewer 3 for the positive feedback, and we provide a point-by-point response to each of their comments as follows:

R3-1: I recognize the authors have performed quite an effort to try to address the large number of hits using disease status compared to NFT. Still, there are some key experiments that remain to be done to give us a better understanding of these huge difference. By performing CT vs PSP DE among the subset of donors with neuropathology (regardless NFT-positive or -negative), the authors should be able to obtain a straightforward comparison of #DE, which can help to conciliate the differences between neuropath and diagnosis.

We appreciate the reviewer for the suggestions. We wish to clarify that the neuropathology scores are only available in the PSP cases. In other words, none of the control brains has the tau neuropathology scores for the detailed tau measures available, nor could these tau scores be obtained from controls for the following reasons: These neuropathology scores are based on tau measurements for four tau lesions either specifically observed in PSP (i.e. oligodendroglial coiled bodies (CB), tufted astrocytes (TA), and tau neuropil threads (TAUTH)) or in tauopathies (i.e. tau neurofibrillary tangles (NFT)). By definition, controls do not have tau lesions that would be seen in PSP, as this would disqualify them from being a control.

Furthermore, these detailed PSP-specific tau scores are obtained using very detailed multi-region (19 anatomical structures) measurements that were specifically conducted in the PSP brains, followed by the generation of continuous scores for the degree of pathology using latent variable modeling that is based on the semi-quantitative tau scores from the 19 brain regions³. These 19 brain regions are specifically selected based on those affected in PSP, namely: basal nucleus, caudate putamen, globus pallidus, hypothalamus, motor cortex, subthalamic nucleus, thalamic fasciculus, ventral thalamus, cerebellar white matter, dentate nucleus, inferior olive, locus coeruleus, medullary tegmentum, midbrain tectum, oculomotor complex, pontine base, pontine tegmentum, red nucleus, and substantia nigra. The neuropathology measures are each on a 0–3 scale and were used to create continuous scores for the degree of pathology, for each of the four pathological lesions (NFT, CB, TA, TAUTH), based on a latent trait approach. An overall latent variable was also calculated by using the semi-quantitative scores for all four lesion types in all regions. These scores are an estimate of an assumed underlying level of pathology severity that all individual scores are dependent on, or correlated with. Notably, a large number of PSP cases (>800) with semi-quantitative tau scores were used to create these latent tau neuropathology variables that are then used in genomic analyses^{4,5}.

In sum, given both the arduous nature of collection of the tau neuropathology scores and their specificity to PSP, these scores are only available from the PSP brains. As such, DEGs between PSP and controls reflect the transcriptional differences between this disease and control (many such transcripts), whereas DEGs associated with tau neuropathology within PSP brains reflect transcriptional differences associated with varying levels of pathology in disease-only brains (fewer number of transcripts).

To avoid confusion to the readers, we updated the **Results sections PSP brains have vast and replicable transcriptome perturbations in glia-enriched genes.**

... even in a brain region relatively spared from gross tau pathology. ~~In terms of neuropathology, we detected~~ Neuropathology association among the PSP cases indicated the greatest number of associations with NFT (134 genes), while 8 gene levels were associated with TauTh (**Figure 1B**).

R3-2: Response Fig 10. suggests that results from transcriptomics analyses differ greatly for diagnosis compared to neuropathology. It should be analyzed and discussed in detail

We thank the reviewer for the comment. In the original Response Fig 10, we showed that there are more genes that are significantly associated with PSP diagnosis than neuropathology. As we explained in the initial response R3-1a, R3-1d, and the **Discussion** section of the manuscript, the neuropathology measures are only available in the PSP cases. Therefore, the neuropathology association “has lower power both owing to smaller sample size and smaller effect size”. Additionally, the smaller number of significant associations between gene expression and neuropathology is likely due to a more homogenous transcriptional state and “fewer transcriptional perturbations” within PSP cases. We also refer the reviewer to our preceding response R3-1 in this document.

R3-3: The authors have proved that there are additional changes in the TCX not captured by tau neuropathology and indeed identified significant changes in the cell proportions. These are key results from their work and should be clearly communicated. A detailed description of the comprehensive model should be provided and used as the primary model. Furthermore, by adding these as covariates into their models, they are finding stronger associations. The authors might want to communicate any additional results without corrections in the supplementary section.

We thank the reviewer for the feedback and include the results from the Comprehensive model (adjusting for cell proportion) in the Supplementary Section as detailed below. We should note that our analytic paradigm that starts with bulk RNAseq discovery, utilizes snRNAseq followed by model system validations and prioritizations. Given that the PSP vs Control DEG at single-cell levels (**Supplementary Data 4**) are not influenced by any cell proportion changes, our results and conclusions are not driven by any changes in cell proportions. Further, although by using the Digital Sorting Algorithm (DSA) package⁶, we detected differences in astrocyte and oligodendrocyte proportions between PSP and control brains in the bulk expression data, the Simple model (not adjusting for cell proportions) and the Comprehensive model (adjusting for cell proportions) are very similar as indicated by the comparable logFC (see **New Supplementary Figure S2B**). Additionally, our top 3 genes, *DDR2*, *STOM* and *KANK2*, are significant in both Simple and Comprehensive model analyses with congruent direction of change (see **New Supplementary Table 7**). Lastly, we should emphasize that the cell proportion estimation is dependent on the deconvolution algorithms and the cell marker genes selected. Put differently, an estimated cell proportion difference may be driven by actual cell proportion changes or by changes in transcription levels. For this reason, Comprehensive model may result in an over-correction for actual transcriptional differences one seeks to capture. Our analytic paradigm which detects transcriptional changes at bulk level using the Simple model, and then filters them using snRNAseq (not affected by cell proportion changes) provides the field with a robust approach aimed at minimizing both false positives (potentially due to cell proportion changes) and false negatives (potentially due to over-correcting). By providing results from the Comprehensive model, we enable the reader to also appreciate the robust nature of our findings.

We report the findings from the Comprehensive model in **Supplementary Figure 2** in the **Results section: PSP brains have vast and replicable transcriptome perturbations in glia-enriched genes.**

Compared to controls, 2,528 genes were differentially expressed (DEGs) in PSP brains at an FDR-adjusted p-value of 0.05 (Figure 1A), suggesting extensive transcriptional dysregulations in PSP brains

at the bulk tissue level, even in a brain region relatively spared from gross tau pathology. **Importantly, a secondary model that adjusted for cell proportions (Figure S2) showed congruent expression dysregulation, indicating that the expression perturbations are not confounded by neuropathology-induced differences in cell populations.**

New Supplementary Figure S2: Secondary analysis of the bulk RNAseq data to assess cell proportion changes in PSP. (A) Estimated cell proportions in PSP and control participants using the Digital Sorting Algorithm. (B) Correlation between the PSP vs Control logFC derived from the Simple model (not adjusting for cell proportion) and the Comprehensive model (adjusting for cell proportion).

We also reported the results from our top three candidate genes in **Supplementary Table 7** and **Results section: Validation of top tau toxicity suppressing PSP glial perturbed genes.**

In alignment with the prioritization paradigm, all three genes have higher expression levels (FDR < 0.05) in PSP brains based on bulk RNAseq (**Figure 5A**). The up-regulation in bulk tissue is also consistent across both studies, suggesting the results are not driven by a particular cohort. **The three genes are up-regulated even after adjusting for differences in cell proportions (Table S7), indicating that expression perturbations are not driven by this but are likely associated with disease pathogenesis.**

New Supplementary Table 7: PSP vs Control DEG statistics for the top 3 candidate genes in the Simple model (not adjusting for cell proportion) and the Comprehensive model (adjusting for cell proportion).

Gene	Study	Simple			Comprehensive		
		logFC	pvalue	FDR	logFC	pvalue	FDR
DDR2	study 1	0.539	9e-04	0.023	0.208	5e-05	3e-04
	study 2	0.599	0.005	0.071	0.226	0.002	0.010
	meta	0.561	1e-05	0.001	0.213	2e-07	3e-06
STOM	study 1	0.431	0.010	0.085	0.137	0.008	0.022
	study 2	0.507	0.015	0.119	0.124	0.061	0.131
	meta	0.461	4e-04	0.008	0.132	0.001	0.004
KANK2	study 1	0.523	0.001	0.025	0.341	2e-07	3e-06
	study 2	0.416	0.009	0.094	0.304	1e-04	0.001
	meta	0.469	3e-05	0.002	0.326	3e-11	1e-09

We also added relevant methods for the cell proportion estimation and comprehensive model in the **Methods section: Bulk Human Brain RNAseq Analyses and Differential Gene Expression**.

Cell proportions were estimated based on the expression values of the top 100 BRETIGEA cell type marker genes for astrocyte, endothelia, microglia, neuron, and oligodendrocyte using the Digital Sorting Algorithm (DSA). The association between PSP vs Control diagnosis and gene expression was assessed in a comprehensive model that adjusted for key covariates (sex, age at death, RIN, sequencing flowcell) and the estimated cell proportions in each study. Meta-analysis was carried out using the same strategy outlined for the main model.

R3-4: Cerebellum analyses are critical for PSP, and results from TCX should be analyzed in the light of unique and common hits.

We thank the reviewer for the suggestions. We completely agree with the reviewer and have already published on transcriptome changes in TCX vs cerebellum (CER) in PSP and in AD⁷. Based on our published work and as discussed in the original response R3-1f, there are conserved gene expression patterns between TCX and cerebellum (CER)⁷. We confirmed that such conservation holds in our data in this manuscript by comparing the PSP vs Control logFC from the meta-analyzed TCX datasets and the CER datasets (**Response Figure 2**). Gene expression levels are perturbed in PSP in a congruent fashion between the two brain regions, indicated by a high degree of correlation of the logFC (Spearman rho = 0.51). Although the PSP vs Control DEG for the three candidate genes we identified are not significant at an FDR cut-off of 0.05, possibly due to the smaller sample size of the CER dataset (N=144), all three are all concordantly up-regulated in both brain regions, again highlighting that these transcriptional changes are unlikely to be driven by gross pathology but may rather be common drivers of neurodegeneration *per se*.

Response Figure 2

A Correlations of PSP vs Ctrl logFC in different brain regions

B

Gene	TCX				CER			
	logFC	pvalue	FDR	N	logFC	pvalue	FDR	N
DDR2	0.561	1e-05	0.001	408	0.399	0.012	0.163	144
STOM	0.461	4e-04	0.008	408	0.508	0.006	0.122	144
KANK2	0.469	3e-05	0.002	408	0.344	0.014	0.176	144

R3-5a: The authors have access to a very large pool of PSP brains, and in previous studies, they have sequenced a huge number of brains. Plenty of reasons, including the high cost of single-cell data generation, restrict the generation of single-cell data for all of these brains.

- a. It would be really informative to indicate why/how the brains were selected for single-cell studies.

We appreciate the reviewer for the questions. Many factors were considered when selecting the samples for snRNAseq. All candidate samples were from individuals with a pathological diagnosis of PSP conducted by our neuropathologist (Dr. Dennis Dickson). We selected the samples for snRNAseq by applying the following selection rules. First, we made sure that the PSP and control samples were from individuals with comparable sex and age at death (**Table S4**). Additionally, we prioritized samples with available bulk tissue RNAseq and/or whole genome sequencing data and for those that had sufficient amount of available frozen tissue. Lastly, for the PSP samples, we selected samples that can best represent the quantitative neuropathology scores from the larger pool of available PSP samples.

We now include text in **Methods: Sample Information** to clarify these points as follows:

Samples for snRNAseq were selected to ensure that the PSP and control donors had comparable sex and age at death. Additionally, we prioritized samples with available bulk tissue RNAseq and/or whole genome sequencing data and for those that had sufficient amount of available frozen tissue. Lastly, for the PSP samples, we selected samples that represented the quantitative neuropathology score distribution of the larger pool of available PSP samples.

R3-5b: The authors have access to a very large pool of PSP brains, and in previous studies, they have sequenced a huge number of brains. Plenty of reasons, including the high cost of single-cell data generation, restrict the generation of single-cell data for all of these brains.

- b. Discuss any possible limitation or bias of the selection of brains can impact in the results presented.

As outlined in R3-5a, the selection of the PSP samples is guided by a combination of key covariates, the availability of tissue and other omics datasets, and the degree of the PSP-related neuropathology. As we showed in **Supplementary Table 4**, the PSP samples selected had comparable sex and age distributions as the controls. Therefore, we do not think the analysis we performed (PSP vs Control DEG) using the snRNAseq data is biased toward any particular biological or technical covariates.

We also note that the selected PSP cases reflect the overall neuropathologic characteristics of our larger PSP samples. We compared the distribution of the PSP-related neuropathology scores between the PSP cases in the snRNAseq dataset and all the available PSP brains with neuropathology scores in our records (**Response Figure 3**). We concluded that the distributions are similar, and the selected cases reflected the overall pathology of all the available PSP cases. Additionally, the selected PSP cases had a similar proportion of females as in the overall PSP population. We want to acknowledge that although the PSP cases selected for snRNAseq had comparable ages as the control samples (**Table S4**), those PSP cases are older compared to the average age of death for PSP patients in the population. This is due to the design of matching age with control samples, who have an older age of death (**Table S4**). However, the differences in PSP ages between the snRNAseq and larger PSP population is unlikely to create a bias in our analysis as we are comparing PSP samples with control samples. If anything, similarity of ages between cases and controls is desirable to reduce any age-related biases.

Response Figure 3

PSP-related pathology distribution of all available samples and those used for snRNAseq

R3-6: Web browser: I anticipate the server will be highly used, and at the present time I found it somehow slow. This brought me concerns about its performance at the time the manuscript is published. I suggest deploying it in a more performant server / cloud to scale up.

We appreciate the suggestions on the performance of the web application. The performance of the web app is limited by its underlying web framework (R shiny). We will continuously monitor the usage of the web app following its public release and make appropriate changes. At the same time, all the data used in the web app are also reported as extended data tables, which the readers can access.

R3-7: Does any of the astrocyte subclusters show the expression of genes of reactive astrocytes? These studies are also key to supporting authors' findings and should be comprehensively done and reported.

We thank the reviewer for the suggestion. In the initial responses to the reviewer (R3-4), we presented results of subclustering of astrocytes from both PSP and control samples. We identified three subclusters Ast_CL0, Ast_CL1, and Ast_CL2. In this analysis, we observed high expression of inflammation related genes such as *SERPINA3* and *OSMR* in Ast_CL2. The expression of reactive astrocyte genes is high in both Ast_CL1 and Ast_CL2 (**Response Figure 4**). Gene expression score indicates a statistically significant higher expression of reactive astrocyte genes in Ast_CL1 and Ast_CL2.

Response Figure 4

Following the reviewer's suggestions, we also included the results of the subclustering analysis in the **Results section: Validation of top tau toxicity suppressing PSP glial perturbed genes** and added it as **Supplementary Figure 17**.

The up-regulation in bulk tissue is also consistent across both studies, suggesting the results are not driven by a particular cohort. In snRNAseq (**Figure 5B**), all three genes are significantly upregulated in astrocytes. Pseudobulk DEG analysis using the snRNAseq datasets also supported the up-regulation of the three genes in astrocytes (**Table S10**). Interestingly, the subclustering analysis (Figure S17) indicated *DDR2* and *KANK2* are up-regulated in specific populations of astrocytes including reactive astrocytes, whereas the up-regulation of *STOM* is universal to all astrocytes. Our transcriptome data demonstrate that *STOM*, *KANK2*, and *DDR2* are up-regulated in glial cells in PSP.

New Supplementary Figure 17. Subclustering analysis of the astrocyte nuclei. **(A)** UMAP projection of the nuclei and the subcluster assignment. **(B)** Top 5 marker genes for each subcluster. **(C-E)** UMAP projection of the nuclei colored by key covariates. **(F-G)** The number of significant PSP vs Control DEGs in each cluster and their overlaps. Interestingly, there are many overlaps between the DEGs in Ast_CL0 and Ast_CL1 in both directions. **(H)** PSP vs Control DEG statistics for *DDR2*, *KANK2*, and *STOM*.

H

GeneName	subcluster	PSPvsCtrl			Control		PSP	
		logFC	pvalue	FDR	avg_exp	pct_exp	avg_exp	pct_exp
DDR2	Ast_CL0	0.0869	0.09324	0.2708	0.6963	58.73%	1.09721	59.12%
	Ast_CL1	0.2124	0.00021	0.0051	0.7960	46.34%	1.09811	60.42%
	Ast_CL2	-0.0262	0.46701	0.8701	0.8575	64.06%	0.84830	70.34%
KANK2	Ast_CL0	0.0343	0.00401	0.0277	0.2504	25.09%	0.32264	26.17%
	Ast_CL1	0.0281	0.00270	0.0330	0.2060	15.30%	0.30969	27.56%
	Ast_CL2	0.0771	0.46194	0.8683	0.2039	26.56%	0.26378	38.98%
STOM	Ast_CL0	0.0614	0.02658	0.1138	0.5195	55.27%	0.57767	48.23%
	Ast_CL1	0.0995	0.01504	0.1064	0.4368	35.34%	0.61523	54.42%
	Ast_CL2	0.2406	0.01463	0.2901	0.4465	53.12%	0.76158	75.42%

We also added relevant text in the **Methods section: Human Brain Single Nucleus RNAseq (snRNAseq)** **Validations** to describe how the analysis was performed.

To elucidate additional cell population, the astrocyte nuclei were subjected to subclustering analysis (Figure S17). Briefly, read counts of nuclei from astrocyte cluster CL2 were normalized, log-transformed, and harmonized using the same procedures for the full dataset. Clustering was carried out at a resolution of 0.3 with default parameters. Subcluster marker genes were identified using the Wilcoxon Rank Sum test. Differentially expressed genes between PSP and Control nuclei in each subcluster were identified with the same approach used for the full dataset.

R3-8: Raw p-value is often referred as nominal p-value. I would suggest using the most accepted nomenclature.

We appreciate the reviewer for the suggestions. We updated the nomenclature in **Figure 1A** and **Supplementary Data 1** from “raw p-value” to “unadjusted p-value”.

References

- 1 Coppola, G. *et al.* Evidence for a role of the rare p.A152T variant in MAPT in increasing the risk for FTD-spectrum and Alzheimer's diseases. *Hum Mol Genet* **21**, 3500-3512 (2012). <https://doi.org:10.1093/hmg/dds161>
- 2 Kara, E. *et al.* The MAPT p.A152T variant is a risk factor associated with tauopathies with atypical clinical and neuropathological features. *Neurobiol Aging* **33**, 2231 e2237-2231 e2214 (2012). <https://doi.org:10.1016/j.neurobiolaging.2012.04.006>
- 3 Bollen, K. A. Latent variables in psychology and the social sciences. *Annu Rev Psychol* **53**, 605-634 (2002). <https://doi.org:10.1146/annurev.psych.53.100901.135239>
- 4 Allen, M. *et al.* Gene expression, methylation and neuropathology correlations at progressive supranuclear palsy risk loci. *Acta Neuropathol* **132**, 197-211 (2016). <https://doi.org:10.1007/s00401-016-1576-7>
- 5 Allen, M. *et al.* Divergent brain gene expression patterns associate with distinct cell-specific tau neuropathology traits in progressive supranuclear palsy. *Acta Neuropathol* **136**, 709-727 (2018). <https://doi.org:10.1007/s00401-018-1900-5>
- 6 Zhong, Y., Wan, Y. W., Pang, K. F., Chow, L. M. L. & Liu, Z. D. Digital sorting of complex tissues for cell type-specific gene expression profiles. *Bmc Bioinformatics* **14** (2013). <https://doi.org:Artn> 89 10.1186/1471-2105-14-89
- 7 Wang, X. *et al.* Alzheimer's disease and progressive supranuclear palsy share similar transcriptomic changes in distinct brain regions. *J Clin Invest* **132** (2022). <https://doi.org:10.1172/JCI149904>

REVIEWERS' COMMENTS

Reviewer #2 (Remarks to the Author):

After reviewing the revised submission, the authors have successfully addressed my previous comments and concerns regarding their original submission. I believe that this paper makes a significant contribution to its field and meets the necessary requirements for publication in Nature Communications. Therefore, I highly recommend its publication.

Cross Species Systems Biology Discovers Glial *DDR2*, *STOM*, and *KANK2* as Therapeutic Targets in Progressive Supranuclear Palsy

NCOMMS-22-50578B

Response to reviewer comments

Reviewer 2.....1

Reviewer 2:

The second reviewer commented: “After reviewing the revised submission, the authors have successfully addressed my previous comments and concerns regarding their original submission. I believe that this paper makes a significant contribution to its field and meets the necessary requirements for publication in Nature Communications. Therefore, I highly recommend its publication.”

We thank Reviewer 2 for their time in reviewing our manuscript. We appreciate the positive feedback and the recommendation for publication.